# PRINCIPLED OUT-OF-DISTRIBUTION GENERALIZATION VIA SIMPLICITY

## ABSTRACT

Modern foundation models exhibit remarkable out-of-distribution (OOD) generalization, solving tasks far beyond the support of their training data. However, the theoretical principles underpinning this phenomenon remain elusive. This paper investigates this problem by examining the compositional generalization abilities of diffusion models in image generation. Our analysis reveals that while neural network architectures are expressive enough to represent a wide range of models—including many with undesirable behavior on OOD inputs—the true, generalizable model that aligns with human expectations typically corresponds to the simplest among those consistent with the training data.

Motivated by this observation, we develop a theoretical framework for OOD generalization via simplicity, quantified using a predefined simplicity metric. We analyze two key regimes: (1) the *constant-gap* setting, where the true model is strictly simpler than all spurious alternatives by a fixed gap, and (2) the *vanishing-gap* setting, where the fixed gap is replaced by a smoothness condition ensuring that models close in simplicity to the true model yield similar predictions. For both regimes, we study the regularized maximum likelihood estimator and establish the first sharp sample complexity guarantees for learning the true, generalizable, simple model.

## 1 INTRODUCTION

Modern foundation models have demonstrated impressive capabilities to generalize to tasks well beyond their training distribution. For instance, diffusion models can generate realistic images from novel combinations of attributes never explicitly observed during training (Dhariwal & Nichol, 2021b; Ho et al., 2020a; Ho & Salimans, 2022; Nichol & Dhariwal, 2021; Ramesh et al., 2021; 2022; Saharia et al., 2022), and large language models routinely produce coherent text that extends beyond explicitly learned patterns (Wei et al., 2021; Chowdhery et al., 2023; Touvron et al., 2023; Bubeck et al., 2023; Achiam et al., 2023; Team et al., 2024; Bai et al., 2023). Despite these compelling successes, the theoretical underpinnings of such out-of-distribution (OOD) generalization remain poorly understood. A fundamental puzzle arises: how do models with extremely high expressive capacity—models known to even memorize random noise (Zhang et al., 2016)—manage to generalize in ways consistent with human expectations?

To shed light on this phenomenon, we begin by closely examining the empirical behavior of diffusion models, particularly their ability to generate coherent images featuring attribute combinations unseen during training (Okawa et al., 2023). Inspired by this observation, we construct a simplified conceptual framework that abstracts key aspects of compositional generalization. Within this abstraction, multiple solutions perfectly fit the source domain, yet exhibit widely divergent predictions when tested on unseen target domain. Crucially, we observe that models failing to generalize tend to exhibit significantly higher structural complexity compared to the model that aligns with human intuition.

Motivated by this insight, we propose that simplicity—quantified by a predefined complexity metric $R(\cdot)$—acts as the key principle guiding successful OOD generalization. Specifically, among all models that fit the training data, the one that generalizes is typically the simplest according to this metric. We formalize this idea in a parametric setting, where the model is parameterized by $\beta \in \mathbb{R}^d$. We assume that the only generalizable parameter (i.e., the ground truth), denoted $\beta^\star$,

satisfies $\beta^\star = \arg\min_{\beta \in \mathcal{B}_S} R(\beta)$, where $\mathcal{B}_S$ denotes the set of all the minimizers on the source (i.e., training) domain. Building upon this simplicity hypothesis, we develop a rigorous theoretical framework for OOD generalization via a regularized maximum likelihood estimator (MLE). Within this framework, we analyze two distinct regimes: (1) the *constant-gap* regime, where the simplicity measure of the true model is strictly lower than that of all spurious alternatives by a fixed margin, i.e., $\Delta := \min_{\beta \in \mathcal{B}_S \setminus \{\beta^\star\}} \{R(\beta) - R(\beta^\star)\} > 0$, and (2) the *vanishing-gap* regime, in which the fixed simplicity margin is replaced by a smoothness condition requiring models close in simplicity to also be similar in their predictions, i.e., for all $\beta_0 \in \mathcal{B}_S$, we have $\|\beta_0 - \beta^\star\|_2 \leq (R(\beta_0) - R(\beta^\star))^\tau$ for some $\tau > 0$.

**Our contributions.** This paper makes two primary contributions toward understanding OOD generalization through the lens of simplicity:

1. **Identification of simplicity as a key driver for OOD generalization.** We propose and formalize the principle that simplicity—measured by a well-defined complexity metric—is a reliable indicator of a model's ability to generalize beyond the training domain. This insight is grounded in a carefully designed experiment, motivated by empirical observations from image generation tasks using diffusion models.

2. **Theoretical analysis providing sharp sample complexity guarantees.** We rigorously examine the regularized maximum likelihood estimator in both the constant-gap and vanishing-gap regimes:

   (a) In the *constant-gap* regime, the estimator recovers the true model at a rate of $\tilde{O}(1/n)$, where $n$ is the sample size.

   (b) In the *vanishing-gap* regime, the estimator recovers the true model at a rate of $\tilde{O}(1/n^{1-\frac{2}{3\tau}})$, which smoothly approaches $\tilde{O}(1/n)$ as a fixed simplicity gap corresponds to a smaller gap in the parameter space (i.e., $\tau \to \infty$).

Collectively, our results provide a principled explanation for how modern foundation models can perform robustly outside their training distribution, highlighting model simplicity as a key mechanism for reliable generalization.

## 1.1 RELATED WORK

**Compositional Generalization.** Recent work has demonstrated that modern foundation models possess remarkable capabilities for compositional generalization, i.e., solving novel tasks by recombining known components in ways not encountered during training. For example, Bubeck et al. (2023) found that an early version of GPT-4 could combine concepts and skills across modalities and domains to solve problems in reasoning, coding, and mathematics. Similar capabilities have been reported in a wide range of large language models (Touvron et al., 2023; Bai et al., 2023; Chowdhery et al., 2023; Team et al., 2024; Wei et al., 2023). These capabilities are closely related to zero-shot and few-shot generalization, which have been extensively explored in prior work (Brown et al., 2020; Wei et al., 2021; Kojima et al., 2023).

To better understand the mechanisms underlying compositional behavior, a line of research has investigated compositional generalization in controlled settings using smaller-scale models. For instance, Ramesh et al. (2024) and Peng et al. (2024) investigate the ability of autoregressive transformers to generalize through function composition. More recently, compositional generalization has also been studied in image generation tasks with conditional diffusion models. Several works (Okawa et al., 2023; Park et al., 2024; Yang et al., 2025) examine synthetic datasets to analyze generalization behavior, identify success and failure modes, and explore the dynamics of learning. These studies also draw connections between compositional generalization and emergent phenomena in generative models, as discussed in Arora & Goyal (2023). However, the primary focus of these studies is to characterize the empirical behaviors of diffusion models. In contrast, our work provides a theoretical framework to explain *why* generalization occurs—even when multiple models fit the training data equally well. We abstract a core aspect of compositional generalization—namely, the ability to correctly predict in unobserved regions of input space—and show that this ability can be explained by a simplicity principle.

**OOD generalization under covariate shift.** The primary focus of this paper is OOD generalization under covariate shift in the underparameterized regime. This line of study dates back to the work of

Shimodaira (2000), who showed that when the model is well-specified, vanilla MLE is asymptotically optimal among all weighted likelihood estimators. For non-asymptotic analysis, Cortes et al. (2010) and Agapiou et al. (2017) established risk bounds for importance weighting. More recent works have extended non-asymptotic guarantees to specific model classes, such as linear regression and one-hidden-layer neural networks (Mousavi Kalan et al., 2020; Lei et al., 2021; Zhang et al., 2022). Most notably, Ge et al. (2023) gave tight non-asymptotic guarantees for well-specified parametric models, showing that vanilla MLE achieves minimax-optimal excess risk without target data. However, their analysis assumes a unique global minimizer on the source domain. We relax this assumption by allowing multiple global minima, recovering their results as a special case within our more general framework.

There is also a growing body of work on covariate shift in the overparameterized regime (Kausik et al., 2023; Chen et al., 2024; Hao et al., 2024; Mallinar et al., 2024; Tsigler & Bartlett, 2023; Tang et al., 2024), as well as in nonparametric settings (Kpotufe & Martinet, 2021; Pathak et al., 2022; Ma et al., 2023; Wang, 2023). However, both of these settings lie outside the scope of our work.

**Regularized maximum likelihood estimation.** Regularized maximum likelihood estimators are a foundational tool in high-dimensional statistics and machine learning, particularly in settings where the number of parameters exceeds the number of samples. Theoretical guarantees for these estimators are typically categorized into two categories: *fast-rate* and *slow-rate* bounds.

Fast-rate bounds, typically of order $O(1/n)$ , are achievable under strong structural assumptions, such as sparsity or restricted conditions on the design matrix. These results are well studied in regression models (Bunea et al., 2007; Raskutti et al., 2019) and graphical models (Ravikumar et al., 2011), and are extensively covered in Bühlmann & Van De Geer (2011); Van de Geer et al. (2016). A canonical example is sparse linear regression, particularly the Lasso, where $\ell_1$-regularization is used to promote sparsity. In this setting, the excess risk is often bounded by $(s \log d)/n$, where $s$ is the sparsity level of the true regression vector, $d$ is the number of parameters, and $n$ is the number of samples. However, such guarantees typically rely on restricted eigenvalue-type conditions, which are challenging to verify and may not hold in practical scenarios.

In the absence of sparsity or restricted eigenvalue assumptions, slow-rate bounds, typically of order $O(1/\sqrt{n})$, can be established for both the linear and nonlinear settings (Greenshtein & Ritov, 2004; Rigollet & Tsybakov, 2011; Massart & Meynet, 2011; Koltchinskii et al., 2011; Huang & Zhang, 2012; Chatterjee, 2013; 2014; Bühlmann, 2013; Dalalyan et al., 2017). For instance, in the Lasso setting without restricted eigenvalue conditions, the prediction error is often bounded by $\sqrt{\log d/n}$.

In contrast to prior work, our setting differs in two key aspects: (1) we operate in the *low-dimensional* regime with a *nonconvex* loss function, and (2) we focus on *OOD generalization* under covariate shift rather than standard in-distribution prediction. As such, existing results do not directly apply, and our analysis develops new tools to handle model selection among multiple source minimizers via a simplicity-based regularization.

## 2 PRELIMINARIES

In this paper, we study covariate shift under a well-specified model. Specifically, we consider covariates $X \in \mathcal{X}$ and responses $Y \in \mathcal{Y}$, with the goal of predicting $Y$ given $X$. We assume two distinct domains: a source domain $S$, with data-generating distribution $P_S(X, Y)$, and a target domain $T$, with distribution $P_T(X, Y)$. Our training data consists of $n$ i.i.d. samples $\{(x_i, y_i)\}_{i=1}^n$ drawn from the source domain. The objective of OOD generalization is to learn a prediction rule from the source data that performs well on the target domain.

Achieving this requires structural assumptions. We focus on *covariate shift*, where the marginal distributions differ, $P_S(X) \neq P_T(X)$, but the conditional distribution remains invariant: $P_S(Y \mid X) = P_T(Y \mid X)$. To formalize this, we consider a parametric function class $\mathcal{F} = \{f(y \mid x; \beta) \mid \beta \in \mathbb{R}^d\}$ for modeling the conditional density $p(y \mid x)$ of $Y \mid X$. The model is *well-specified* if there exists a parameter $\beta^\star$ such that $p(y \mid x) = f(y \mid x; \beta^\star)$.

We use the negative log-likelihood as the loss function: $\ell(x, y, \beta) := -\log f(y \mid x; \beta)$. Given the dataset $\{(x_i, y_i)\}_{i=1}^n$, the empirical loss is defined as the average loss over the training samples: $\ell_n(\beta) := \frac{1}{n} \sum_{i=1}^n \ell(x_i, y_i, \beta)$. The standard maximum likelihood estimator (MLE) is then

defined as the parameter that minimizes this empirical loss, i.e., $\hat{\beta}_{\mathsf{MLE}} := \arg\min_\beta \ell_n(\beta)$. To evaluate generalization performance on the target domain, we define the *excess risk* at a parameter $\beta$ as $\mathcal{E}(\beta) := \mathbb{E}_T[\ell(x, y, \beta)] - \mathbb{E}_T[\ell(x, y, \beta^\star)]$, where $\mathbb{E}_T$ denotes expectation under the target distribution. The excess risk quantifies how much worse a model with parameter $\beta$ performs on the target domain compared to the true model $\beta^\star$. A small excess risk indicates that $\beta$ makes predictions nearly as accurate as the optimal model under the target distribution.

## 3 EMPIRICAL OBSERVATIONS ON OOD GENERALIZATION

In this section, we present empirical observations from two complementary settings. The first involves a text-conditioned diffusion model for image generation, where we observe strong OOD generalization. The second abstracts this setup into a simple model using a multilayer perceptron (MLP), enabling controlled comparisons between generalizable and non-generalizable solutions.

### 3.1 OOD GENERALIZATION IN DIFFUSION MODELS

We study OOD generalization in a diffusion model trained to generate images conditioned on text-based attribute combinations. Our dataset consists of $28 \times 28$ images of circles that vary along three binary attributes: background color (light/dark), foreground color (blue/red), and size (large/small). This results in $2^3 = 8$ unique classes, each represented by a 3-bit label: the first bit denotes background color, the second foreground color, and the third size. Figure 1a displays one representative image for each class.

We train a diffusion model on 200,000 images, sampling 50,000 examples from each of the four source classes: $S = \{(0, 0, 0), (0, 0, 1), (0, 1, 0), (1, 0, 0)\}$. Each training image is subject to minor attribute variations and small additive Gaussian noise. We then evaluate the model on the four held-out target classes: $T = \{(0, 1, 1), (1, 0, 1), (1, 1, 0), (1, 1, 1)\}$. Despite never seeing these combinations during training, the model generates high-quality, semantically accurate images for all target classes (Figure 1b). This indicates a strong degree of OOD generalization.

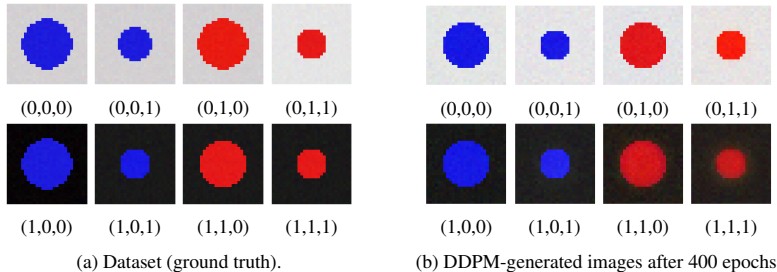

| (a) Dataset (ground truth). | (b) DDPM-generated images after 400 epochs. |

Figure 1: Diffusion Model Image Generation Setting.

### 3.2 A SIMPLIFIED SETTING FOR ANALYSIS

To better understand the generalization behavior observed in the diffusion model, we construct a simplified version of the task using a 2-layer multilayer perceptron (MLP). Instead of generating images, the model is trained to learn the identity function on $\mathbb{R}^3$. Each of the 3-bit labels from the original image generation task is now treated as a point in $\mathbb{R}^3$, and the MLP is trained to map input $x$ to output $x$. As before, we use the classes $S = \{(0, 0, 0), (0, 0, 1), (0, 1, 0), (1, 0, 0)\}$ and $T = \{(0, 1, 1), (1, 0, 1), (1, 1, 0), (1, 1, 1)\}$ as our source and target domains, respectively.

For each $s \in S$, we sample 100 input vectors $x_i$ from a multivariate Gaussian with mean $s$ and covariance $0.01 I_3$, and assign $y_i = x_i$, producing what we refer to as identity samples. This yields 400 training examples of the form $(x_i, x_i)$. For evaluation, we generate 20 test examples for each $t \in T$, sampling from a Gaussian with mean $t$ and covariance $0.001 I_3$. We find that a well-initialized and optimized MLP trained solely on the source domain reliably generalizes to the target domain. We refer to such a solution as the *generalizable model*.

**Non-generalizable Alternatives.** To contrast this behavior, we train additional models that match the identity function on the source domain but intentionally deviate from it on the target domain.

Each such model is trained on 400 identity samples from $S$, along with 400 modified samples from $T$ (100 per target class), where the outputs are systematically altered to break the identity mapping. We explore three distinct modification schemes:

- **Uniform Map:** For each $t \in T$, we uniformly sample a random vector $r_t \in [0, 2]^3$ and draw 100 input vectors $x_i$ from a Gaussian centered at $t$. The corresponding outputs are set to $y_i = x_i - t + r_t$, which centers the responses at $r_t$ rather than $t$. Note that $r_t$ can take non-integer values, introducing continuous distortions in the output space. We run 80 independent trials; in each trial, we independently resample a new shift $r_t$ for each $t \in T$, generating 400 modified samples. These are then combined with 400 newly sampled identity samples from $S$, and the model is trained on the full set of 800 samples.
- **Permutation Map:** Each $t \in T$ is randomly assigned to a different center $r_t$ chosen from $S \cup T$. We sample 100 inputs $x_i$ from a Gaussian centered at $t$, and define outputs as $y_i = x_i - t + r_t$. We conduct 20 such trials.
- **Flipped Map and Interpolations:** We define the flipped map by $x \mapsto (1, 1, 1) - x$ for inputs $x$ sampled near $T$. One trial uses this exact mapping. In 10 additional trials, we interpolate between the identity and flipped maps with $y_i = \alpha((1, 1, 1) - x_i) + (1 - \alpha)x_i$, for $\alpha = 0, 0.1, 0.2, \ldots, 0.9$.

We consider three types of non-generalizable maps, each designed to probe a different aspect of model behavior. The *Uniform Map* introduces high variability by randomly shifting target outputs to continuous locations in $[0, 2]^3$, allowing us to explore a wide range of spurious solutions that still perfectly fit the source data. The *Permutation Map* is more structured and realistic, as each target label is reassigned to another vertex in $S \cup T$; this better reflects failure modes observed in diffusion models, where the model might misassociate target combinations with incorrect but discrete concepts. Finally, the *Flipped Map and its interpolations* allow us to systematically study how gradual deviations from the identity mapping affect the simplicity and generalizability of the learned model.

**Comparing Simplicity.** While all models fit the source domain, those trained with non-generalizable target mappings fail to extend the identity function to $T$. These models consistently exhibit higher complexity, measured by the sum of squared Frobenius norms of all layer weights and biases (Figure 2). In contrast, the generalizable model has significantly smaller norms, suggesting that simplicity is a key factor in achieving successful OOD generalization.

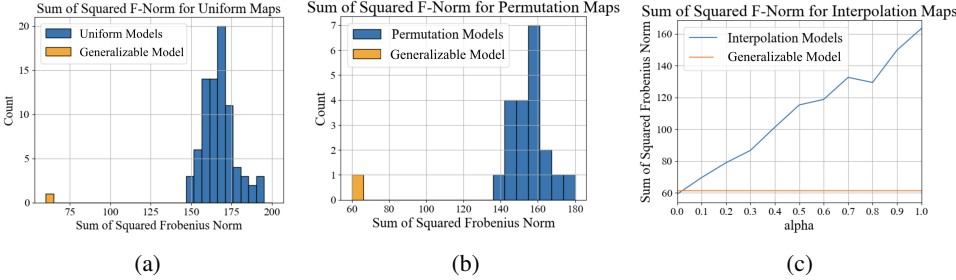

(a)                          (b)                          (c)

Figure 2: **generalizable vs. non-generalizable model weights.** (a) Sum of squared Frobenius norms of weights in models trained on uniform mappings. (b) Sum of squared Frobenius norms of weights in models trained on permutation mappings. (c) Sum of squared Frobenius norms for models trained on interpolations between the identity and flipped maps. Here, $\alpha = 0$ corresponds to the identity map and $\alpha = 1$ to the flipped map. In all three plots, the model trained solely on $S$ using the identity map is shown in orange.

## 4 MAIN RESULTS

In this section, we begin with a formal problem setup in Section 4.1. We then analyze the performance of the regularized MLE by deriving excess risk bounds. Specifically, Section 4.2 presents results for the constant-gap regime, while Section 4.3 addresses the vanishing-gap regime.

### 4.1 PROBLEM FORMULATION

Motivated by the observations in Section 3, we consider the setting where the population loss on the source domain, $\mathbb{E}_S[\ell(x, y, \beta)]$, admits multiple minimizes. This arises naturally because the

source data only partially constrains the prediction function; specifically, in regions of the covariate space that lie outside the support of the source domain, predictions can be defined arbitrarily without affecting performance on the source domain. However, among these multiple solutions, typically only one parameter—the true parameter $\beta^\star$—generalizes effectively, *i.e.*, $\beta^\star$ is the unique minimizer of the target-domain loss $\mathbb{E}_T[\ell(x, y, \beta)]$.

We posit that this true parameter corresponds to the "simplest" solution among all the source-domain minima, where "simplicity" is quantified by a measure denoted by $R(\beta)$. Formally, we assume:

$$\beta^\star = \arg\min_\beta R(\beta) \quad \text{s.t.} \quad \beta \in \arg\min_\beta \mathbb{E}_S[\ell(x, y, \beta)].$$

This perspective aligns with common observations in practice, where multiple parameter configurations yield identical performance on training data but differ significantly in their generalization capabilities. Typically, parameters with smaller norms or simpler representations often generalize better, a phenomenon widely leveraged in practice through regularization techniques such as weight decay. Accordingly, we consider the regularized maximum likelihood estimator (MLE) defined by

$$\hat{\beta}_\lambda := \arg\min_\beta \left\{ \ell_n(\beta) + \lambda R(\beta) \right\}, \tag{1}$$

where $\lambda > 0$ is a regularization parameter to be determined later. Note that the solution to (1) might not be unique; in the event of multiple solutions, $\hat{\beta}_\lambda$ denotes any solution from the solution set. For simplicity of notation, we define $\mathcal{B}_S := \arg\min_\beta \mathbb{E}_S[\ell(x, y, \beta)]$ and $B_S := \max_{\beta \in \mathcal{B}_S} \|\beta\|_2$.

To facilitate the forthcoming analysis, we invoke the concept of Fisher information. Formally, we define the Fisher information at $\beta$ for the source and target domains as: $\mathcal{I}_S(\beta) := \mathbb{E}_S[\nabla^2 \ell(x, y, \beta)]$ and $\mathcal{I}_T(\beta) := \mathbb{E}_T[\nabla^2 \ell(x, y, \beta)]$. In this paper, we consider two distinct scenarios based on the simplicity measure $R(\cdot)$ evaluated on the solution set $\mathcal{B}_S$ from the source domain:

1. **Constant gap scenario:** The simplicity measure $R(\cdot)$ has a strictly positive gap between the true parameter $\beta^\star$ and any other spurious solution in $\mathcal{B}_S \setminus \{\beta^\star\}$.
2. **Vanishing gap scenario:** The simplicity measure $R(\cdot)$, when evaluated on points in $\mathcal{B}_S \setminus \{\beta^\star\}$, can be made arbitrarily close to $R(\beta^\star)$.

We begin by stating several assumptions that apply to both scenarios considered in this paper.

**Assumption A.** *We make the following assumptions:*

*A.1 (Concentration inequalities): There exist $B_0$, $B_1$, $B_2$, absolute constants $c, \gamma$, and a threshold $N$ such that for any fixed matrix $A \in \mathbb{R}^{d \times d}$ and any $n > N$, the following inequalities hold simultaneously with probability at least $1 - n^{-20}$:*

$$|\ell_n(\beta) - \mathbb{E}[\ell_n(\beta)]| \le B_0 \sqrt{\frac{\log n}{n}}, \quad \forall \beta \in \mathbb{R}^d,$$

$$\|A(\nabla \ell_n(\beta^\star) - \mathbb{E}[\nabla \ell_n(\beta^\star)])\|_2 \le c\sqrt{\frac{V \log n}{n}} + B_1 \|A\|_2 \log^\gamma \left( \frac{B_1 \|A\|_2}{\sqrt{V}} \right) \frac{\log n}{n},$$

$$\left\| \nabla^2 \ell_n(\beta^\star) - \mathbb{E}[\nabla^2 \ell_n(\beta^\star)] \right\|_2 \le B_2 \sqrt{\frac{\log n}{n}},$$

*where $V = n \cdot \mathbb{E} \|A(\nabla \ell_n(\beta^\star) - \mathbb{E}[\nabla \ell_n(\beta^\star)])\|_2^2$ denotes the variance term.*

*A.2 (Hessian Lipschitz): There exists a constant $B_3 \ge 0$ such that for all $x \in \mathcal{X}_S \cup \mathcal{X}_T$, $y \in \mathcal{Y}$, and $\beta \in \mathbb{R}^d$, $\|\nabla^3 \ell(x, y, \beta)\|_2 \le B_3$, where $\mathcal{X}_S$ and $\mathcal{X}_T$ denote the supports of $\mathbb{P}_S(X)$ and $\mathbb{P}_T(X)$, respectively.*

*A.3 (Gap between minima): There exists a constant gap $G > 0$ separating the global minimum from all other local minima of $\mathbb{E}_S[\ell(x, y, \beta)]$. Specifically, for any local minimum $\beta' \in \mathbb{R}^d \setminus \mathcal{B}_S$, it holds that $\mathbb{E}_S[\ell(x, y, \beta')] \ge \mathbb{E}_S[\ell(x, y, \beta^\star)] + G$. Furthermore, there exists a constant $B > 0$ such that for all $\beta \in \mathbb{R}^d$ with $\|\beta\|_2 \ge B$, $\mathbb{E}_S[\ell(x, y, \beta)] \ge \mathbb{E}_S[\ell(x, y, \beta^\star)] + G$.*

*A.4 (Properties of $R(\beta)$): The simplicity measure $R(\beta)$ satisfies: (1) $R(0) = 0$ and $R(\beta) \ge 0$ for all $\beta$; (2) $R(\beta)$ is convex; (3) $R(\beta)$ is $L$-smooth.*

We now provide several remarks on Assumption A:

Assumption A.1 imposes standard concentration conditions, which are commonly satisfied when the loss function, its gradient, and its Hessian are uniformly bounded. In particular, the second inequality is a generalized version of the Bernstein inequality, which reduces to the classical form when $\gamma = 0$. Notably, the second and third inequalities require concentration only at $\beta^\star$, rather than uniformly over all $\beta$.

Assumption A.2 is a mild regularity condition requiring Lipschitz continuity of the Hessian. In general, if the loss function is differentiable up to the third order and the input distribution is supported on a compact set or has light tails, this assumption is easily satisfied.

Assumption A.4 specifies basic conditions on the simplicity measure $R(\beta)$. A canonical example satisfying all three conditions is the squared $\ell_2$-norm (also known as weight decay), $R(\beta) = \|\beta\|_2^2$, which is widely used in ridge regression and neural network training. Other valid examples include the squared group $\ell_{2,1}$ norm, $R(\beta) = \sum_{g \in \mathcal{G}} \|\beta_g\|_2^2$, where $\mathcal{G}$ is a partition of features and $\beta_g$ denotes the corresponding subvector, commonly used in multitask learning; and Huberized $\ell_1$ penalties, which smoothly transition between squared $\ell_2$ near zero and $\ell_1$ for larger values, often used to promote sparsity while preserving smoothness.

A key structural assumption is Assumption A.3. In essence, Assumption A.3 states that all local—but non-global—minima, including those at large distances, are at least $G$ worse than the global minimum. Importantly, our theoretical results do not depend on the specific choice of the constant $B$ in the second part of the assumption. This means $B$ can be chosen to be sufficiently large, so the second inequality can be interpreted as ruling out spurious local minima at infinity.

## 4.2 Constant gap scenario

We begin by analyzing the constant gap scenario. In addition to Assumption A, we introduce the following assumptions:

**Assumption B.** *B.1 (Strong convexity) There exists a constant $\alpha > 0$ such that for all $\beta_0 \in \mathcal{B}_S$,*

$$\mathbb{E}_S \left[ \nabla^2 \ell(x, y, \beta_0) \right] \succeq \alpha I_d.$$

*B.2 (Constant simplicity gap) There exists a constant gap in the simplicity measure, defined as*

$$\Delta := \min_{\beta \in \mathcal{B}_S \setminus \{\beta^\star\}} \left\{ R(\beta) - R(\beta^\star) \right\} > 0.$$

Assumption B.1 ensures sufficient local curvature of the population loss around each $\beta_0 \in \mathcal{B}_S$, while Assumption B.2 guarantees that the true model is the simpler than all source-compatible candidates by at least a gap $\Delta$ measured by $R$. This separation can be leveraged to facilitate effective learning.

We now state the main result for this setting:

**Theorem 4.1.** *Let $\lambda = \frac{8B_0}{\Delta}\sqrt{\frac{\log n}{n}}$, $\mathcal{I}_S := \mathcal{I}_S(\beta^\star)$, and $\mathcal{I}_T := \mathcal{I}_T(\beta^\star)$. Under Assumptions A and B, if $n \geq c \max\{N^\star, N\}$, then with probability at least $1 - n^{-10}$, the excess risk of the regularized estimator defined in (1) satisfies*

$$\mathcal{E}(\hat{\beta}_\lambda) \leq c \left( \frac{\mathrm{Tr}\left(\mathcal{I}_T \mathcal{I}_S^{-1}\right) \log n}{n} + \frac{B_0^2 \left\|\mathcal{I}_T^{1/2} \mathcal{I}_S^{-1} \nabla R(\beta^\star)\right\|_2^2 \log n}{\Delta^2 n} \right)$$

*for an absolute constant $c$. Here $N^\star := Poly\left(\Delta^{-1}, \alpha^{-1}, G^{-1}, L, B_0, B_1, B_2, B_3, B_s, R(\beta^\star), \|\mathcal{I}_S^{-1}\nabla R(\beta^\star)\|_2, \|\mathcal{I}_T^{\frac{1}{2}}\mathcal{I}_S^{-1}\nabla R(\beta^\star)\|_2^{-1}, \|\mathcal{I}_S^{-1}\|_2, \|\mathcal{I}_T^{\frac{1}{2}}\mathcal{I}_S^{-1}\mathcal{I}_T^{\frac{1}{2}}\|_2^{-1}\right).$*

For an exact characterization of the threshold $N^\star$, one can refer to (22) in the Appendix.

Theorem 4.1 provides a non-asymptotic upper bound on the excess risk of regularized maximum likelihood estimation under a simplicity gap. The theorem states that, when the true model is strictly simpler than all competing source-compatible solutions, the regularized estimator successfully learns it with excess risk achieving a fast convergence rate of order $\tilde{O}(1/n)$. The bound consists of two main terms:

- **Statistical difficulty under covariate shift.** The first term, $\mathrm{Tr}(\mathcal{I}_T \mathcal{I}_S^{-1})/n$, captures the intrinsic challenge of generalization under covariate shift. The matrix $\mathcal{I}_S^{-1}$ reflects the variance of the

parameter estimation, while $\mathcal{I}_T$ quantifies how the parameter estimation accuracy impacts performance on the target domain. If the directions emphasized by $\mathcal{I}_T$ are poorly captured by $\mathcal{I}_S$, generalization becomes harder—reflected by a larger trace term. In the special case where there is no distribution shift (i.e., $\mathcal{I}_S = \mathcal{I}_T$), the trace reduces to $d$, and the bound matches the classical $d/n$ rate for well-specified linear models.

- **Regularization and simplicity bias.** The second term reflects the influence of regularization and the role of simplicity. It depends on the alignment between the regularization gradient $\nabla R(\beta^\star)$ and the Fisher geometry, and it scales inversely with the square of the simplicity gap $\Delta$. This highlights the advantage of a larger simplicity gap: the more clearly the true model is separated in simplicity from competing models, the more confidently the estimator can distinguish it from spurious alternatives.

In the special case where the source-compatible model is unique (i.e., $\mathcal{B}_S = \{\beta^\star\}$ and $\Delta = \infty$), the second term vanishes, and Theorem 4.1 recovers Theorem 3.1 from Ge et al. (2023). As shown in their work, this rate is minimax optimal, meaning that our bound is tight in the worst case.

### 4.3 VANISHING GAP SCENARIO

We now turn to the vanishing-gap scenario, where the simplicity gap between the true model and competing alternatives can become arbitrarily small.

In this regime, the global minimizers of the population loss on the source domain (i.e., $\mathcal{B}_S$) may not be isolated from $\beta^\star$; instead, they may form a continuum, such as a low-dimensional surface in the neighborhood of $\beta^\star$. This necessitates additional assumptions to capture the geometric structure of $\mathcal{B}_S$ more precisely. Formally, we impose the following additional assumptions:

**Assumption C.** *C.1 The solution set $\mathcal{B}_S \subseteq \mathbb{R}^d$ is a compact $C^1$ differentiable submanifold of dimension $d_S$.*

*C.2 There exists a constant $\alpha > 0$ such that for all $\beta_0 \in \mathcal{B}_S$,*

$$\lambda_{\min}\left(\mathbb{E}_S\left[\nabla^2 \ell(x, y, \beta_0)\right]\right) \geq \alpha, \quad \text{and} \quad rank\left(\mathbb{E}_S\left[\nabla^2 \ell(x, y, \beta_0)\right]\right) = d - d_S,$$

*where $\lambda_{\min}(A)$ denotes the smallest non-zero eigenvalue of matrix $A$.*

*C.3 There exists $\tau \geq 9$ and $\Delta_{\max} < 1$ such that for all $\Delta \leq \Delta_{\max}$ and all $\beta_0 \in \mathcal{B}_S$ satisfying $R(\beta_0) - R(\beta^\star) = \Delta$, we have $\|\beta_0 - \beta^\star\|_2 \leq \Delta^\tau$.*

We note that the constant-gap scenario satisfies Assumption C with $d_S = 0$ and $\tau = \infty$.

Assumption C.2 mirrors the strong convexity condition in Assumption B.1 but adapts it to the case where $\mathcal{B}_S$ has positive dimension. It guarantees sufficient curvature in directions orthogonal to the solution manifold.

The key structural condition is Assumption C.3, which plays the role of a "soft" simplicity gap. It ensures that any model with simplicity value close to that of the true model must also be close to it in parameter space.

Finally, we remark that Assumption C.1 is introduced primarily for simplicity of presentation. It can be naturally extended to the more general setting where $\mathcal{B}_S \subseteq \mathbb{R}^d$ is a finite union of compact $C^1$ submanifolds that are separated by a constant distance.

We then state the main result for this setting:

**Theorem 4.2.** *Let $\lambda = \frac{8B_0}{\Delta_{\max}}\sqrt{\frac{\log n}{n^{1-\frac{2}{3\tau}}}}$, $\mathcal{I}_S := \mathcal{I}_S(\beta^\star)$, and $\mathcal{I}_T := \mathcal{I}_T(\beta^\star)$. Under Assumptions A and C, if $n \geq c \max\{N', N\}$, then with probability at least $1 - n^{-10}$, the excess risk of the regularized estimator defined in (1) satisfies*

$$\mathcal{E}(\hat{\beta}_\lambda) \leq c\left(\frac{\mathrm{Tr}\left(\mathcal{I}_T \mathcal{I}_S^\dagger\right)\log n}{n} + \frac{B_0^2 \left\|\mathcal{I}_T^{1/2}\mathcal{I}_S^\dagger \nabla R(\beta^\star)\right\|_2^2 \log n}{\Delta_{\max}^2 n^{1-\frac{2}{3\tau}}}\right)$$

*for an absolute constant $c$. Here $A^\dagger$ denotes the pseudoinverse of $A$ and $N' = Poly\,(\Delta_{\max}^{-1}, \alpha^{-1},$ $G^{-1}, L, B_0, B_1, B_2, B_3, B_s, \|\mathcal{I}_S\|_2, \|\mathcal{I}_S\|_2^{-1}, \mathrm{Tr}(\mathcal{I}_S), \|\mathcal{I}_S^\dagger\|_2, \|\mathcal{I}_S^\dagger\|_2^{-1}, R(\beta^\star), \|\mathcal{I}_S^\dagger \nabla R(\beta^\star)\|_2,$ $\|\mathcal{I}_T\|_2, \|\mathcal{I}_T\|_2^{-1}, \|\mathcal{I}_T^{\frac{1}{2}}\mathcal{I}_S^\dagger \nabla R(\beta^\star)\|_2^{-1}, \|\mathcal{I}_T^{\frac{1}{2}}\mathcal{I}_S^\dagger \mathcal{I}_T^{\frac{1}{2}}\|_2^{-1}).$*

For an exact characterization of the threshold $N'$, one can refer to (41) in the Appendix.

Theorem 4.2 shows that even in the absence of a fixed simplicity gap, the regularized estimator still achieves a meaningful excess risk bound of order $\tilde{O}(n^{-1+\frac{2}{3\tau}})$, provided that models with similar simplicity to the true model also produce similar predictions. While the structure of the bound resembles that of Theorem 4.1, it differs in two key ways:

- **Use of the pseudoinverse.** When $\mathcal{I}_S$ is singular, the inverse in Theorem 4.1 is replaced by the Moore–Penrose pseudoinverse $\mathcal{I}_S^\dagger$. Recall that for a positive semidefinite matrix like $\mathcal{I}_S$, the pseudoinverse $\mathcal{I}_S^\dagger$ acts like the true inverse on the subspace where $\mathcal{I}_S$ is invertible (its column space), and returns zero in directions where $\mathcal{I}_S$ is degenerate (its null space). In other words, $\mathcal{I}_S^\dagger$ projects onto the effective subspace where the source distribution provides information for estimation, and inverts only within that subspace. To illustrate, consider the case where $R(\beta) = \|\beta\|_2^2$. Any parameter $\beta \in \mathbb{R}^d$ can be decomposed into two orthogonal components: $\beta = \beta_{\text{null}} + \beta_{\text{col}}$, where $\beta_{\text{null}}$ lies in the null space of $\mathcal{I}_S$ and $\beta_{\text{col}}$ lies in its column space. Since the population loss is flat in the null space, the source domain provides no information about $\beta_{\text{null}}^\star$. Consequently, only $\beta_{\text{col}}^\star$ can be estimated, and its estimation variance is governed by $\mathcal{I}_S^\dagger$. In our setting, the simplicity bias selects the globally simplest solution—implying $\beta_{\text{null}}^\star = 0$. The regularization term thus ensures that the learned parameter remains in the estimable subspace, allowing for meaningful recovery even when $\mathcal{I}_S$ is degenerate.

- **Role of $\tau$.** Assumption C.3 introduces a smoothness condition linking simplicity and proximity to the true model. As $\tau$ increases, a gap in simplicity corresponds to a smaller deviation in parameter space, i.e., $\|\beta_0 - \beta^\star\|_2 \leq (R(\beta_0) - R(\beta^\star))^\tau$, enabling tighter generalization guarantees. This is reflected in the convergence rate $\tilde{O}(n^{-1+\frac{2}{3\tau}})$, which improves with larger $\tau$ and approaches the optimal rate $\tilde{O}(n^{-1})$ as $\tau \to \infty$. In this limit, Theorem 4.2 reduces to Theorem 4.1, thereby generalizing the constant-gap analysis and providing a smooth transition between the idealized setting of a uniquely simplest model and more realistic scenarios in which simplicity varies continuously.

## 5 CONCLUSION

This paper presents a theoretical framework for understanding OOD generalization through the lens of simplicity. By focusing on diffusion models and their compositional generalization behavior, we show that despite the expressiveness of neural architectures, models that generalize well often align with the simplest explanation consistent with the data. We formalize this insight through two regimes—the constant-gap and vanishing-gap settings—and provide sharp sample complexity guarantees for learning the true model via regularized maximum likelihood.

**Discussion on equivalence classes.** We conclude with a discussion of a potential limitation of our current analysis and outline a promising direction for extending our results to address it.

Consider a simple scenario where the response variable is given by $y = (\beta^{\star\top}x)^2$, for some true parameter $\beta^\star = (0, \beta_{-1}^\star) \neq 0$. Let the source domain be $\mathcal{X}_S := \{(0, x_{-1}) \mid x_{-1} \in \mathbb{R}^{d-1}\}$, the target domain be $\mathcal{X}_T := \mathbb{R}^d$, and the regularizer be $R(\beta) = \|\beta\|_2^2$. In this setting, both $\beta^\star$ and $-\beta^\star$ yield identical predictions on all inputs and have the same regularization value, i.e., $R(-\beta^\star) = R(\beta^\star)$. As such, these parameters should be considered equivalent, and Theorem 4.2 is expected to remain valid in this setting. However, a direct application of Theorem 4.2 is not possible because Assumption C.3 is violated: $-\beta^\star \in \mathcal{B}_S$ and $R(-\beta^\star) - R(\beta^\star) = 0$, yet we have $\|\beta^\star - (-\beta^\star)\|_2 = 2\|\beta^\star\|_2 \neq 0$.

We believe this limitation can be addressed through a modest extension of our framework. Specifically, rather than working directly in $\mathbb{R}^d$, it is more natural to consider the quotient space $\mathbb{R}^d/\sim$, where parameters are grouped into equivalence classes based on predictive and regularization equivalence. Specifically, we define the equivalence class of a parameter $\beta$ as

$$[\beta] := \left\{ \tilde{\beta} \mid \ell(x, y, \tilde{\beta}) = \ell(x, y, \beta) \text{ for all } x \in \mathcal{X}_S \cup \mathcal{X}_T, \ y \in \mathcal{Y}, \text{ and } R(\tilde{\beta}) = R(\beta) \right\},$$

and denote the associated equivalence relation by $\sim$. We believe that our theoretical results can be naturally extended to this quotient space, with $\mathcal{B}_S$ redefined accordingly. In particular, variants of Theorems 4.1 and 4.2 should hold when interpreted over equivalence classes, with bounds computed using appropriate representatives from $[\beta^\star]$. We leave a formal development of this extension to future work.

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
