# A DIFFUSION EXPERIMENTAL DETAILS

## A.1 SYNTHETIC DATASET

We begin by presenting the image-generating process for the diffusion setting, followed by the diffusion model architecture and training pipeline used in Section 3.1.

Recall that the three attributes of interest are background color, foreground color, and size, which are represented in the class label as (`[bg-color]`, `[fg-color]`, `[size]`). Every $28 \times 28$ image depicts a centered circle with two possible configurations for each attribute. For the background color, the RGB values for training images with `[bg-color] = 0` and `[bg-color] = 1` are sampled from $\mathcal{U}[0,1] \cdot [0.2, 0.2, 0.2]$ and $[0.8, 0.8, 0.8] + \mathcal{U}[0,1] \cdot [0.2, 0.2, 0.2]$, corresponding to a light and dark gray. For the foreground color, the RGB values in training images with `[fg-color] = 0` and `[fg-color] = 1` are sampled from $[0.0, 0.0, 0.8] + \mathcal{U}[0,1] \cdot [0.2, 0.2, 0.2]$ and $[0.8, 0.0, 0.0] + \mathcal{U}[0,1] \cdot [0.2, 0.2, 0.2]$, corresponding to blue and red. Finally, the radii for training images with `[size] = 0` and `[size] = 1` are sampled from $0.55 + \mathcal{U}[0,1] \cdot 0.1$ and $0.35 + \mathcal{U}[0,1] \cdot 0.1$, corresponding to large and small. For each image class in $S = \{(0,0,0), (0,0,1), (0,1,0), (1,0,0)\}$, we sample 50,000 values for each of the three attributes specified by the class label. The training images are then constructed from each of the 200,000 sampled attribute values over the four training classes, after the addition of some i.i.d. Gaussian noise with standard deviation $0.01$.

The test images are generated in a similar manner, except we now use the image classes in $T = \{(1,1,0), (1,0,1), (0,1,1), (1,1,1)\}$ and also use the mean of each attribute configuration distribution, instead of sampling attribute values. So for background color, the RGB values in test images with labels `[bg-color] = 0` and `[bg-color] = 1` are exactly $(0.1, 0.1, 0.1)$ and $(0.9, 0.9, 0.9)$. Similarly, for the foreground color, the RGB values in training images with labels `[fg-color] = 0` and `[fg-color] = 1` are exactly $(0.1, 0.1, 0.9)$ and $(0.9, 0.1, 0.1)$, and the radii for training images with labels `[size] = 0` and `[size] = 1` are $0.6$ and $0.4$. For each image class in $T$, we generate 2,000 images with the specified attribute values, with the addition of some i.i.d. Gaussian noise with standard deviation $0.01$.

## A.2 ARCHITECTURE

Our experiment uses a text-conditioned diffusion model with U-Net denoisers, as seen in Dhariwal & Nichol (2021a); Ho et al. (2020b). At a high level, diffusion models work by learning to transform Gaussian noise into samples from the data distribution through an iterative denoising process. The sampling process begins with a noisy input $x_T$, and repeatedly applies a denoiser to recover $x_{T-1}, \ldots, x_0$, where $x_0$ denotes the original image. So given some $x_t$, the U-Net denoiser aims to predict the noise $\epsilon$ that was incorporated at timestep $t$ in the forward diffusion process that resulted in $x_t$. Text-conditioning allows for its prediction $\epsilon_\theta(t, x_t, c)$ to depend also on some conditioning information $c$. The denoiser is then optimized with respect to the mean square error (MSE) between the predicted noise and the true noise: $\mathcal{L} = \mathbb{E}_{t,x_0,\epsilon}[\|\epsilon - \epsilon_\theta(t, x_t(x_0, \epsilon), c)\|^2]$.

We borrow the architecture from Okawa et al. (2023). Our conditional U-Nets comprise of two down-sampling and up-sampling convolutional blocks involving $3 \times 3$ convolutional layers, GeLU activation, global attention, and pooling layers. The conditioning information is then embedded and concatenated during up-sampling. We use a total of 500 denoising steps.

## A.3 OPTIMIZATION

Our diffusion model is trained using the Adam optimizer (Kingma & Ba (2017)) with learning rate 1e-4, batch size 64, and 400 training epochs. We also compute the test loss achieved by the model after each training epoch, for all four test classes. Similar to the training loss, we compute test loss using the mean square error between $\epsilon$ and the $\epsilon_\theta(t, x_t(x_0, \epsilon), c)$ for $(x_0, c)$ drawn from the test set. The training loss and test loss are depicted in Figure 3.

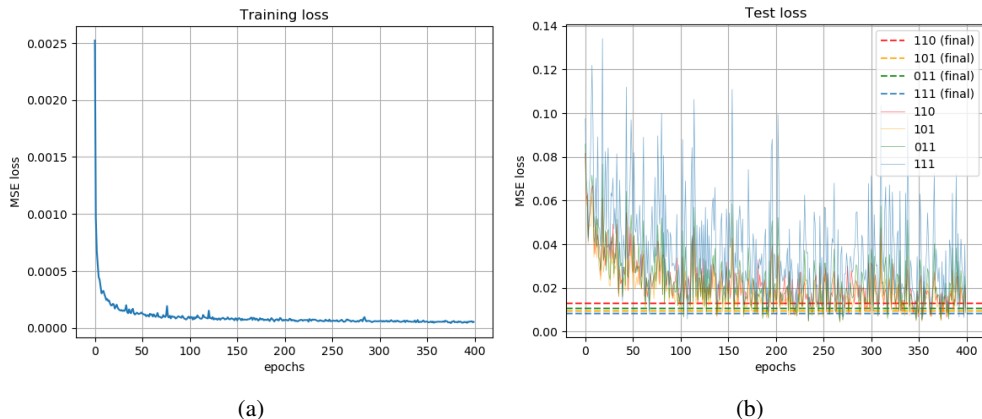

(a)  (b)

Figure 3: **Diffusion Training.** (a) Training loss (MSE) per epoch, averaged over all 200,000 training examples. (b) Test loss (MSE) per epoch for each test class, averaged over all 2,000 test examples per class. The final test loss after all 400 epochs is plotted in dashed lines, with numerical values 2.00398e-4, 1.46671e-4, 1.67488e-4, and 1.32801e-4.

### A.4 COMPUTATION RESOURCES

The experiments are conducted on a server using an NVIDIA RTX A6000 GPU. Each experiment can be completed in a few hours.

## B SIMPLIFIED SETTING EXPERIMENTAL DETAILS

### B.1 EXPERIMENTAL SETUP

Throughout all experiments in Section 3.2, we use a 2-layer MLP with biases, ReLU activations, hidden dimension 128, and the PyTorch default initialization (He initialization, He et al. (2015)). Every model is trained using the Adam optimizer (Kingma & Ba (2017)), with mean square error (MSE) loss between the true label and the label predicted by the MLP. We use a learning rate of $5 \times 10^{-5}$ and $40,000$ training epochs for all experiments.

The covariates in the training dataset are sampled from multivariate Gaussians of the form $N(s, 0.01I_3)$, where $s \in \mathcal{D}_{\text{train}} \subset \{0,1\}^3$. The choice of $\mathcal{D}_{\text{train}}$ varies across different experimental settings. For each $s \in \mathcal{D}_{\text{train}}$, we generate 100 covariates from the corresponding Gaussian. For evaluation, we define $\mathcal{D}_{\text{test}} := \{0,1\}^3 \setminus \mathcal{D}_{\text{train}}$ and sample 20 test covariates from $N(t, 0.001I_3)$ for each $t \in \mathcal{D}_{\text{test}}$.

In each experimental trial, we conduct $k = 10$ independent runs. For each run, a new training dataset is sampled, and a separate model is trained on it. Covariates are sampled independently, and labels are assigned according to the mapping defined for that trial. All training losses and model weight norms reported in the subsequent sections are averaged over the 10 independently trained models.

### B.2 IDENTITY MAPPING

We first train a collection of ten models on covariates sampled from $\mathcal{D}_{\text{train}} = \{(0,0,0), (0,0,1), (0,1,0), (1,0,0)\}$, with labels given by the identity map. The resulting fitted model closely approximates the identity map on all of $\{0,1\}^3$. The test set is generated by sampling covariates from $\mathcal{D}_{\text{test}} = \{0,1\}^3 \setminus \mathcal{D}_{\text{train}}$ and assigning labels corresponding to the identity map.

### B.3 UNIFORM MAPPING SCHEME

For the uniform mapping scheme, we sample training covariates from all eight centers in $\{0,1\}^3$ and define training labels in the following way. For covariates $x_i$ sampled in a Gaussian ball around $\{(0,0,0), (0,0,1), (0,1,0), (1,0,0)\}$, we assign the label $y_i = x_i$ given by the identity map.

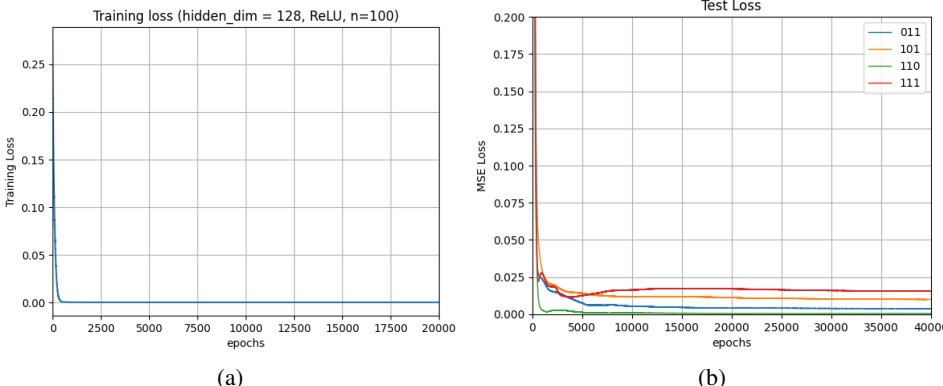

(a)  (b)

Figure 4: **Identity Mapping Training:** (a) Training loss (MSE) per epoch averaged over all ten runs, plotted for the first $20,000$ of $40,000$ total epochs; final training loss: 6.02663e-4. (b) Test loss (MSE) per epoch for each test class, averaged over all ten models. We restrict the y-axis (loss) of the plot to make the differences between the test losses for different classes visible. The final test losses after all $20,000$ epochs are 3.60000e-3, 9.82313e-3, 5.24610e-3, 1.55543e-2.

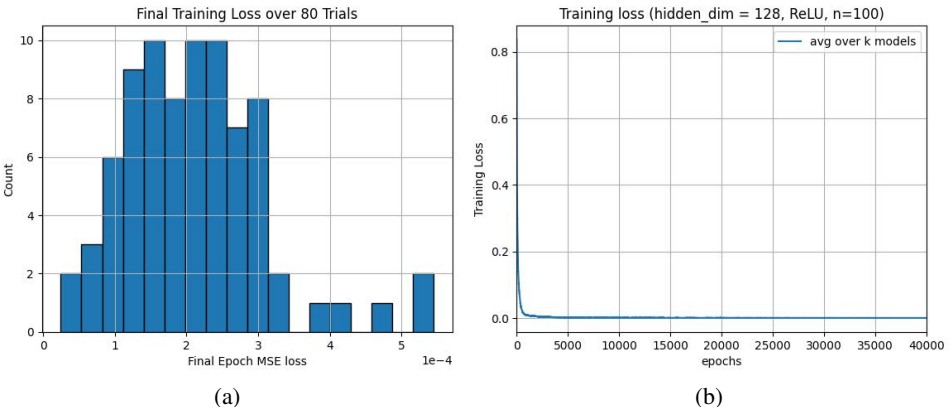

(a)  (b)

Figure 5: **Uniform Mapping Training:** (a) Histogram of training loss (MSE) in final epoch for all eighty trials. Training losses are averaged over ten independent models trained per trial. (b) An example of the training loss curve for a randomly selected trial averaged over all 10 runs.

For covariates $x_i$ sampled around $t_j \in \{(1,1,0), (1,0,1), (0,1,1), (1,1,1)\}$, we assign the label $y_i = r_j + x_i - t_j$, where $r_j \sim U([0,2]^3)$. We conduct a total of eighty trials using this scheme, where each trial resamples the random points $R = \{r_j\}_{j=1}^4$. Figure 5 depicts training losses for this setting.

## B.4 Permutation Mapping Scheme

For the permutation mapping scheme, we sample training covariates from all eight centers in $\{0,1\}^3$ and define training labels in the following way. For covariates $x_i$ sampled in a Gaussian ball around $\{(0,0,0), (0,0,1), (0,1,0), (1,0,0)\}$, we assign the label $y_i = x_i$ given by the identity map. For covariates $x_i$ sampled around $t_j \in \{(1,1,0), (1,0,1), (0,1,1), (1,1,1)\}$, we assign the label $y_i = r_j + x_i - t_j$, where $r_j \sim U(\{0,1\}^3)$. We conduct a total of twenty trials using this scheme, where each trial resamples the random points $R = \{r_j\}_{j=1}^4$. Figure 6 depicts training losses for this setting.

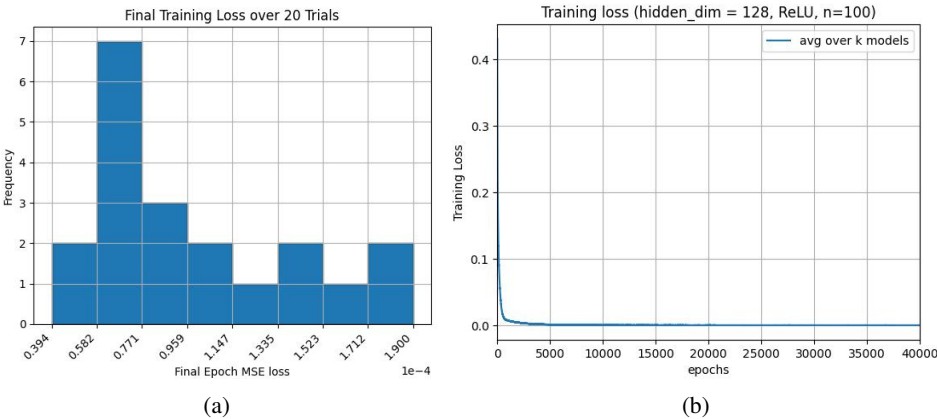

(a)          (b)

Figure 6: **Permutation Mapping Training:** (a) Histogram of training loss (MSE) in final epoch for all twenty trials. Training losses are averaged over ten independent models trained per trial. (b) An example of an average training loss curve for a randomly selected trial.

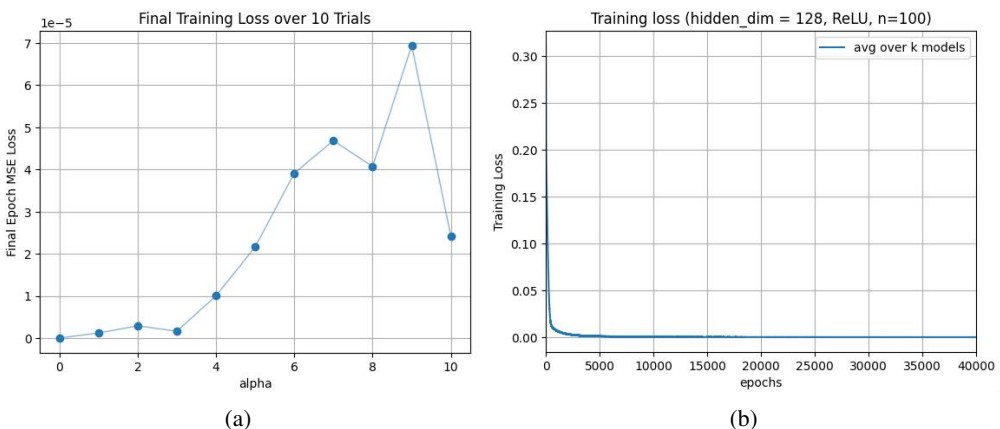

(a)          (b)

Figure 7: **Interpolation Mapping Training:** (a) Plot of training loss (MSE) in final epoch for all eleven $\alpha$. Training losses are averaged over ten independent models. Note that $\alpha = 0$ corresponds to the identity map on $\{0, 1\}^3$, while $\alpha = 1$ corresponds to the flipped map. (b) An example of an average training loss curve for $\alpha = 1$ (corresponding to the flipped map).

### B.5   INTERPOLATING BETWEEN IDENTITY AND FLIPPED MAPPINGS

For the flipped mapping, we sample training covariates from all eight centers in $\{0, 1\}^3$ and assign labels as follows: for covariates $x_i$ sampled from Gaussians centered at $\{(0, 0, 0), (0, 0, 1), (0, 1, 0), (1, 0, 0)\}$, we use the identity map and set $y_i = x_i$; for covariates sampled around $t_j \in \{(1, 1, 0), (1, 0, 1), (0, 1, 1), (1, 1, 1)\}$, we apply the flipped map and set $y_i = (1, 1, 1) - x_i$.

To study a smooth transition between these two mappings, we define an interpolation scheme over eleven choices of $\alpha \in \{0, 0.1, \dots, 1\}$. For each $\alpha$, labels for covariates sampled around $t_j$ are defined as

$$y_i = \alpha((1, 1, 1) - x_i) + (1 - \alpha)x_i.$$

For each $\alpha$, we again train 10 independent models. All reported results, including training losses shown in Figure 7, are averaged over these 10 independent models.

### B.6   COMPUTATION RESOURCES

The experiments are conducted on a personal computer with 8 CPUs. Each experiment can be completed within a few hours.

## C PROOFS FOR SECTION 4

Throughout this section, we use c to denote universal constants, which may vary from line to line.

In this section, we first present the proof of Theorem 4.1 in Section C.1, followed by the proof of Theorem 4.2 in Section C.2. We begin with a simple observation that will be used in both proofs.

In the following analysis, we work under the event that the concentration inequalities stated in Assumption A.1 hold. For notational simplicity, we define

$$\hat{L}(\beta) := \ell_n(\beta) + \lambda R(\beta) = \frac{1}{n}\sum_{i=1}^{n}\ell(x_i, y_i, \beta) + \lambda R(\beta),$$

which is the objective function minimized in (1).

Under Assumption A.1, we have for any $\beta$

$$\hat{L}(\beta) = \ell_n(\beta) + \lambda R(\beta) \geq \mathbb{E}_S[\ell(x, y, \beta)] - B_0\sqrt{\frac{\log n}{n}}$$

and

$$\hat{L}(\beta^\star) = \ell_n(\beta^\star) + \lambda R(\beta^\star) \leq \mathbb{E}_S[\ell(x, y, \beta^\star)] + \lambda R(\beta^\star) + B_0\sqrt{\frac{\log n}{n}}.$$

Thus, as long as

$$\mathbb{E}_S[\ell(x, y, \beta)] > \mathbb{E}_S[\ell(x, y, \beta^\star)] + \lambda R(\beta^\star) + 2B_0\sqrt{\frac{\log n}{n}} \equiv A(n), \qquad (2)$$

we have $\hat{L}(\beta) > \hat{L}(\beta^\star)$. In other words, it holds that

$$\hat{\beta}_\lambda \in \left\{\beta \in \mathbb{R}^d \mid \mathbb{E}_S[\ell(x, y, \beta)] \leq A(n)\right\}. \qquad (3)$$

### C.1 PROOFS OF THEOREM 4.1

In this section, we prove Theorem 4.1. Let

$$\lambda = \frac{c_\lambda}{\Delta}\sqrt{\frac{\log n}{n}}, \quad \text{where } c_\lambda = 8B_0. \qquad (4)$$

In the sequel, we define

$$D := \min\left\{\frac{\Delta}{4LB_S}, \frac{\alpha}{2B_3}\right\}, \qquad (5)$$

and let $\mathcal{B}(\beta, B)$ denote the Euclidean ball in $\mathbb{R}^d$ centered at $\beta$ with radius $B$.

We begin by proving the following proposition:

**Proposition C.1.** *Suppose $n \geq N_1(logN_1)^2$, where $N_1 = \frac{128}{\alpha^2 D^4}\max\{\frac{R(\beta^\star)^2 c_\lambda^2}{\Delta^2}, 4B_0^2\}$. Then, for all $\beta_0 \in \mathcal{B}_S$, it holds that*

$$\mathbb{E}_S[\ell(x, y, \beta)] > A(n), \quad \forall \beta \in \partial\mathcal{B}(\beta_0, D).$$

*Proof of Proposition C.1.* Fix $\beta_0 \in \mathcal{B}_S$. By Assumption A.2, we have for all $\beta \in \mathcal{B}(\beta_0, D)$

$$\left\|\mathbb{E}_S[\nabla^2\ell(x, y, \beta)] - \mathbb{E}_S[\nabla^2\ell(x, y, \beta_0)]\right\|_2 \leq B_3\|\beta - \beta_0\|_2 \leq B_3 D \leq \frac{\alpha}{2}.$$

Thus, by Weyl's inequality, we have

$$\left|\lambda_{\min}\left(\mathbb{E}_S[\nabla^2\ell(x, y, \beta)]\right) - \lambda_{\min}\left(\mathbb{E}_S[\nabla^2\ell(x, y, \beta_0)]\right)\right|$$
$$\leq \left\|\mathbb{E}_S[\nabla^2\ell(x, y, \beta)] - \mathbb{E}_S[\nabla^2\ell(x, y, \beta_0)]\right\|_2 \leq \frac{\alpha}{2},$$

which by Assumption B.1, then gives

$$\lambda_{\min}\left(\mathbb{E}_S[\nabla^2\ell(x,y,\beta)]\right) \geq \lambda_{\min}\left(\mathbb{E}_S[\nabla^2\ell(x,y,\beta_0))\right) - \frac{\alpha}{2} \geq \frac{\alpha}{2}.$$

In other words, $\mathbb{E}_S[\ell(x,y,\beta)]$ is $\frac{\alpha}{2}$-strongly convex on $\mathcal{B}(\beta_0, D)$. Thus, for any $\beta \in \partial\mathcal{B}(\beta_0, D)$, we have

$$\mathbb{E}_S[\ell(x,y,\beta)] \geq \mathbb{E}_S[\ell(x,y,\beta_0)] + \mathbb{E}_S[\nabla\ell(x,y,\beta_0)]^\top(\beta - \beta_0) + \frac{\alpha}{4}\|\beta - \beta_0\|_2^2$$

$$= \mathbb{E}_S[\ell(x,y,\beta_0)] + \frac{\alpha}{4}D^2$$

$$= \mathbb{E}_S[\ell(x,y,\beta^\star)] + \frac{\alpha}{4}D^2.$$

Consequently, as long as $n \geq N_1$, we have for any $\beta \in \partial\mathcal{B}(\beta_0, D)$

$$\mathbb{E}_S[\ell(x,y,\beta)] \geq \mathbb{E}_S[\ell(x,y,\beta^\star)] + \frac{\alpha}{4}D^2 > \mathbb{E}_S[\ell(x,y,\beta^\star)] + \lambda R(\beta^\star) + 2B_0\sqrt{\frac{\log n}{n}} = A(n),$$

which then finishes the proofs. $\qquad\square$

With the proposition in hand, we are able to establish the following lemma.

**Lemma C.2.** *Suppose that* $n \geq N_2(\log N_2)^2$, *where* $N_2 = \max\{\frac{128}{\alpha^2 D^4}, \frac{16}{G^2}\}$ · $\max\{\frac{R(\beta^\star)^2 c_\lambda^2}{\Delta^2}, 4B_0^2\} \geq N_1$. *Then, for all* $\beta \notin \cup_{\beta_0 \in \mathcal{B}_S}\mathcal{B}(\beta_0, D)$, *we have* $\mathbb{E}_S[\ell(x,y,\beta)] > A(n)$.

*Proof of Lemma C.2.* We prove the lemma by contradiction. Suppose that there exists some $\beta \notin \bigcup_{\beta_0 \in \mathcal{B}_S}\mathcal{B}(\beta_0, D)$ such that

$$\mathbb{E}_S[\ell(x,y,\beta)] \leq A(n).$$

Recall Assumption A.3. Let $\Omega := \mathcal{B}(0, B) \setminus \bigcup_{\beta_0 \in \mathcal{B}_S}\mathcal{B}(\beta_0, D)$. Note that by Assumption A.3, for all $\|\beta\|_2 \geq B$, we have

$$\mathbb{E}_S[\ell(x,y,\beta)] \geq \mathbb{E}_S[\ell(x,y,\beta^\star)] + G > A(n),$$

where the last inequality holds as long as $n \geq N_2(\log N_2)^2$. This means there exists $\beta \in \Omega$ such that $\mathbb{E}_S[\ell(x,y,\beta)] \leq A(n)$.

By Proposition C.1, we know that for all $\beta \in \partial\Omega$, it holds that

$$\mathbb{E}_S[\ell(x,y,\beta)] > A(n).$$

This implies the existence of a local minimum of $\mathbb{E}_S[\ell(x,y,\beta)]$ in $\mathring{\Omega}$. Denote this local minimum by $\beta'$.

Observe that

$$\mathbb{E}_S[\ell(x,y,\beta')] \leq A(n) = \mathbb{E}_S[\ell(x,y,\beta^\star)] + \lambda R(\beta^\star) + 2B_0\sqrt{\frac{\log n}{n}}$$

$$< \mathbb{E}_S[\ell(x,y,\beta^\star)] + G,$$

where the last inequality holds as long as $n \geq N_2(\log N_2)^2$. Therefore, $\beta' \in \Omega$ must be a global minimizer, which contradicts the definition of $\Omega$.

$\qquad\square$

By, Lemma C.2 and (3), we then have

$$\hat{\beta}_\lambda \in \left\{\beta \in \mathbb{R}^d \mid \mathbb{E}_S[\ell(x,y,\beta)] \leq A(n)\right\} \subset \cup_{\beta_0 \in \mathcal{B}_S}\mathcal{B}(\beta_0, D). \tag{6}$$

The following lemma further refines this result by showing that $\hat{\beta}_\lambda$ actually lies within the ball centered at $\beta^\star$.

**Lemma C.3.** *For all* $\beta \in \cup_{\beta_0 \in \mathcal{B}_S \setminus \{\beta^\star\}}\mathcal{B}(\beta_0, D)$, *we have:*

$$\mathbb{E}_S[\ell(x,y,\beta^\star)] + \lambda R(\beta^\star) + \frac{\lambda}{2}\Delta < \mathbb{E}_S[\ell(x,y,\beta)] + \lambda R(\beta).$$

*Proof of Lemma C.3.* Let $\beta \in \mathcal{B}(\beta_0, D)$ for some $\beta_0 \in \mathcal{B}_S \setminus \{\beta^\star\}$. By Assumption A.4, we then have

$$
\begin{aligned}
R(\beta) &\geq R(\beta_0) + \nabla R(\beta_0)^\top (\beta - \beta_0) \\
&\geq R(\beta_0) - \|\nabla R(\beta_0)\|_2 \|\beta - \beta_0\|_2 \\
&\geq R(\beta_0) - L B_S \|\beta - \beta_0\|_2 \\
&\geq R(\beta_0) - L B_S D,
\end{aligned}
$$

where the first inequality follows from the convexity of $R(\beta)$ and the third inequality follows from the fact that $\nabla R(0) = 0$ and thus $\|\nabla R(\beta_0)\|_2 \leq L\|\beta_0\|_2 \leq L B_S$.

Thus, we have

$$
\begin{aligned}
R(\beta) - R(\beta^\star) &\geq R(\beta_0) - R(\beta^\star) - L B_S D \\
&\geq \Delta - L B_S D \\
&> \frac{\Delta}{2},
\end{aligned}
$$

where the last inequality follows from the choice of $D$ given in (5). Further, notice that

$$
\mathbb{E}_S[\ell(x, y, \beta)] \geq \mathbb{E}_S[\ell(x, y, \beta^\star)].
$$

We then finish the proofs.

$\square$

By Lemma C.3 and Assumption A.1, for all $\beta \in \cup_{\beta_0 \in \mathcal{B}_S \setminus \{\beta^\star\}} \mathcal{B}(\beta_0, D)$, it holds that

$$
\begin{aligned}
\hat{L}(\beta) &= \ell_n(\beta) + \lambda R(\beta) \\
&\geq \mathbb{E}_S[\ell(x, y, \beta)] + \lambda R(\beta) - B_0 \sqrt{\frac{\log n}{n}} \\
&\geq \mathbb{E}_S[\ell(x, y, \beta^\star)] + \lambda R(\beta^\star) + \frac{\lambda}{2}\Delta - B_0 \sqrt{\frac{\log n}{n}} \\
&\geq \hat{L}(\beta^\star) + \frac{\lambda}{2}\Delta - 2B_0 \sqrt{\frac{\log n}{n}} \\
&> \hat{L}(\beta^\star),
\end{aligned}
$$

where the last inequality follows from the choice of $\lambda$ given in (4). Consequently, we have

$$
\hat{\beta}_\lambda \notin \cup_{\beta_0 \in \mathcal{B}_S \setminus \{\beta^\star\}} \mathcal{B}(\beta_0, D).
$$

Combine with (6), we have

$$
\hat{\beta}_\lambda \in \mathcal{B}(\beta^\star, D).
$$

As shown in the proofs of Proposition C.1, $\mathbb{E}_S[\ell(x, y, \beta)]$ is $\frac{\alpha}{2}$-strongly convex within the ball $\mathcal{B}(\beta^\star, D)$. As a result, we restrict our analysis to the following optimization problem:

$$
\min_{\beta \in \mathcal{B}(\beta^\star, D)} \ell_n(\beta) + \lambda R(\beta), \tag{7}
$$

$$
\text{where} \quad \mathbb{E}_S\left[\nabla^2 \ell(x, y, \beta)\right] \succeq \frac{\alpha}{2} I_d \quad \text{for all } \beta \in \mathcal{B}(\beta^\star, D).
$$

For notational simplicity, we denote the gradient concentration bound from Assumption A.1 by

$$
C(n, A) := c\sqrt{\frac{V \log n}{n}} + B_1 \|A\|_2 \log^\gamma \left(\frac{B_1 \|A\|_2}{\sqrt{V}}\right) \cdot \frac{\log n}{n},
$$

where

$$
\begin{aligned}
V &= n \cdot \mathbb{E}\left\|A\left(\nabla \ell_n(\beta^\star) - \mathbb{E}[\nabla \ell_n(\beta^\star)]\right)\right\|_2^2 \\
&= n \cdot \mathbb{E}[\nabla \ell_n(\beta^\star)^T A^T A \nabla \ell_n(\beta^\star)]
\end{aligned}
$$

$$= n \cdot \mathbb{E}[\mathsf{Tr}(A\nabla\ell_n(\beta^\star)\nabla\ell_n(\beta^\star)^T A^T)]$$
$$= \mathsf{Tr}(A\mathcal{I}_S(\beta^\star)A^T).$$

We further denote $\mathcal{I}_S := \mathcal{I}_S(\beta^\star)$ in the following.

We start by proving a useful proposition.

**Proposition C.4.** *For all $\beta$, it holds that*

$$|(\mathbb{E}_S[\ell(x,y,\beta)] - \ell_n(\beta)) - (\mathbb{E}_S[\ell(x,y,\beta^\star)] - \ell_n(\beta^\star))|$$

$$\leq \min\left\{2B_0\sqrt{\frac{\log n}{n}}, C(n,I_d)\|\beta-\beta^\star\|_2 + B_2\sqrt{\frac{\log n}{n}}\|\beta-\beta^\star\|_2^2 + B_3\|\beta-\beta^\star\|_2^3\right\}.$$

*Here*

$$C(n,I_d) = c\sqrt{\frac{\mathsf{Tr}(\mathcal{I}_S)\log n}{n}} + B_1\log^\gamma\left(\frac{B_1}{\sqrt{\mathsf{Tr}(\mathcal{I}_S)}}\right) \cdot \frac{\log n}{n}.$$

*Proof of Proposition C.4.* Note that by Assumption A.1 and A.2, for all $\beta$:

$$|(\mathbb{E}_S[\ell(x,y,\beta)] - \ell_n(\beta)) - (\mathbb{E}_S[\ell(x,y,\beta^\star)] - \ell_n(\beta^\star))|$$

$$\leq |(\beta-\beta^\star)^\top \nabla(\mathbb{E}_S[\ell(x,y,\beta^\star)] - \ell_n(\beta^\star))|$$

$$+ \frac{1}{2}|(\beta-\beta^\star)^\top\nabla^2(\mathbb{E}_S[\ell(x,y,\beta^\star)] - \ell_n(\beta^\star))(\beta-\beta^\star)| + \frac{B_3}{3}\|\beta-\beta^\star\|_2^3$$

$$\leq C(n,I)\|\beta-\beta^\star\|_2 + B_2\sqrt{\frac{\log n}{n}}\|\beta-\beta^\star\|_2^2 + B_3\|\beta-\beta^\star\|_2^3.$$

Moreover, we have

$$|(\mathbb{E}_S[\ell(x,y,\beta)] - \ell_n(\beta)) - (\mathbb{E}_S[\ell(x,y,\beta^\star)] - \ell_n(\beta^\star))|$$
$$\leq |\mathbb{E}_S[\ell(x,y,\beta)] - \ell_n(\beta)| + |\mathbb{E}_S[\ell(x,y,\beta^\star)] - \ell_n(\beta^\star)|$$
$$\leq 2B_0\sqrt{\frac{\log n}{n}}.$$

Thus, we finish the proofs. $\qquad\square$

The following lemma further restricts (7) to a smaller ball with radius $O(n^{-1/2})$.

**Lemma C.5.** *Suppose $n \geq N_3(\log N_3)^2$ where*

$$N_3 = c\max\left\{\frac{B_2^2}{\alpha^2}, \frac{c_\lambda^2 L^2\|\beta^\star\|_2^2 B_3^2}{\alpha^4\Delta^2}, \frac{B_0^2 B_3^4}{\alpha^6}\right\}.$$

*Then, for all $\beta \in \mathcal{B}(\beta^\star, D) \setminus \mathcal{B}(\beta^\star, D'')$, we have $\hat{L}(\beta) > \hat{L}(\beta^\star)$. Here*

$$D'' := \frac{8}{\alpha}(C(n,I_d) + \lambda L\|\beta^\star\|_2). \tag{8}$$

*Proof of Lemma C.5.* For any $\beta \in \mathcal{B}(\beta^\star, D)$, we have

$$\hat{L}(\beta) = \ell_n(\beta) + \lambda R(\beta)$$
$$= \mathbb{E}_S[\ell(x,y,\beta)] + \ell_n(\beta) - \mathbb{E}_S[\ell(x,y,\beta)] + \lambda R(\beta)$$
$$\geq \mathbb{E}_S[\ell(x,y,\beta^\star)] + \frac{\alpha}{4}\|\beta-\beta^\star\|_2^2 + \ell_n(\beta) - \mathbb{E}_S[\ell(x,y,\beta)] + \lambda R(\beta)$$
$$= \ell_n(\beta^\star) + \lambda R(\beta^\star) + \frac{\alpha}{4}\|\beta-\beta^\star\|_2^2$$
$$+ (\mathbb{E}_S[\ell(x,y,\beta^\star)] - \ell_n(\beta^\star)) - (\mathbb{E}_S[\ell(x,y,\beta)] - \ell_n(\beta)) + \lambda(R(\beta) - R(\beta^\star))$$
$$= \hat{L}(\beta^\star) + \frac{\alpha}{4}\|\beta-\beta^\star\|_2^2$$

$$+ \left(\mathbb{E}_S[\ell(x, y, \beta^\star)] - \ell_n(\beta^\star)\right) - \left(\mathbb{E}_S[\ell(x, y, \beta)] - \ell_n(\beta)\right) + \lambda \left(R(\beta) - R(\beta^\star)\right),$$

where the inequality follows from the strong convexity of $\mathbb{E}_S[\ell(x, y, \beta)]$ within the ball $\mathcal{B}(\beta^\star, D)$. Note that by Assumption A.4, we have

$$R(\beta) - R(\beta^\star) \geq \nabla R(\beta^\star)^\top (\beta - \beta^\star) \geq -\|\nabla R(\beta^\star)\|_2 \|\beta - \beta^\star\|_2 \geq -L\|\beta^\star\|_2 \|\beta - \beta^\star\|_2.$$

Thus, we obtain for all $\beta \in \mathcal{B}(\beta^\star, D)$ that

$$\hat{L}(\beta) \geq \hat{L}(\beta^\star) + \frac{\alpha}{4} \|\beta - \beta^\star\|_2^2$$
$$- \left|\left(\mathbb{E}_S[\ell(x, y, \beta^\star)] - \ell_n(\beta^\star)\right) - \left(\mathbb{E}_S[\ell(x, y, \beta)] - \ell_n(\beta)\right)\right| - \lambda L\|\beta^\star\|_2 \|\beta - \beta^\star\|_2. \quad (9)$$

By Proposition C.4, we then have

$$\hat{L}(\beta) \geq \hat{L}(\beta^\star) + \frac{\alpha}{4} \|\beta - \beta^\star\|_2^2 - 2B_0 \sqrt{\frac{\log n}{n}} - \lambda L\|\beta^\star\|_2 \|\beta - \beta^\star\|_2.$$

Thus, as long as

$$\|\beta - \beta^\star\|_2 > \frac{2\lambda L\|\beta^\star\|_2 + 2\sqrt{\lambda^2 L^2 \|\beta^\star\|_2^2 + 2\alpha B_0 \sqrt{\frac{\log n}{n}}}}{\alpha} \equiv D' = \tilde{O}(n^{-1/4}),$$

we have

$$\frac{\alpha}{4} \|\beta - \beta^\star\|_2^2 - 2B_0 \sqrt{\frac{\log n}{n}} - \lambda L\|\beta^\star\|_2 \|\beta - \beta^\star\|_2 > 0$$

and thus $\hat{L}(\beta) > \hat{L}(\beta^\star)$. In other words, for all $\beta \in \mathcal{B}(\beta^\star, D) \setminus \mathcal{B}(\beta^\star, D')$, we have $\hat{L}(\beta) > \hat{L}(\beta^\star)$. Next, we deal with $\mathcal{B}(\beta^\star, D')$. Note that for all $\beta \in \mathcal{B}(\beta^\star, D')$, by (9) and Proposition C.4, we have

$$\hat{L}(\beta) \geq \hat{L}(\beta^\star) + \frac{\alpha}{4} \|\beta - \beta^\star\|_2^2 - \left(C(n, I_d) \|\beta - \beta^\star\|_2 + B_2 \sqrt{\frac{\log n}{n}} \|\beta - \beta^\star\|_2^2 + B_3 \|\beta - \beta^\star\|_2^3\right)$$
$$- \lambda L\|\beta^\star\|_2 \|\beta - \beta^\star\|_2$$
$$\geq \hat{L}(\beta^\star) + \frac{\alpha}{4} \|\beta - \beta^\star\|_2^2 - \left(C(n, I_d) \|\beta - \beta^\star\|_2 + B_2 \sqrt{\frac{\log n}{n}} \|\beta - \beta^\star\|_2^2 + B_3 D' \|\beta - \beta^\star\|_2^2\right)$$
$$- \lambda L\|\beta^\star\|_2 \|\beta - \beta^\star\|_2$$

As long as $n \geq N_3(\log N_3)^2$, we have

$$\frac{\alpha}{4} - B_2 \sqrt{\frac{\log n}{n}} - B_3 D' \geq \frac{\alpha}{8}.$$

Thus, we have

$$\hat{L}(\beta) \geq \hat{L}(\beta^\star) + \frac{\alpha}{8} \|\beta - \beta^\star\|_2^2 - C(n, I_d) \|\beta - \beta^\star\|_2 - \lambda L\|\beta^\star\|_2 \|\beta - \beta^\star\|_2.$$

Consequently, for $\beta \in \mathcal{B}(\beta^\star, D')$, as long as

$$\|\beta - \beta^\star\|_2 > \frac{8}{\alpha} \left(C(n, I_d) + \lambda L\|\beta^\star\|_2\right) = D'' = \tilde{O}(n^{-1/2}),$$

we have $\hat{L}(\beta) > \hat{L}(\beta^\star)$. In other words, for all $\beta \in \mathcal{B}(\beta^\star, D') \setminus \mathcal{B}(\beta^\star, D'')$, we have $\hat{L}(\beta) > \hat{L}(\beta^\star)$. Thus, we conclude that for all $\beta \in \mathcal{B}(\beta^\star, D) \setminus \mathcal{B}(\beta^\star, D'')$, we have $\hat{L}(\beta) > \hat{L}(\beta^\star)$.

$$\square$$

We are now ready to prove Theorem 4.1. For notational simplicity, we denote $\mathcal{I}_S := \mathcal{I}_S(\beta^\star)$, $\mathcal{I}_T := \mathcal{I}_T(\beta^\star)$, $\alpha_1 := B_1 \|\mathcal{I}_S^{-1}\|_2^{1/2}$, $\alpha_2 := B_2 \|\mathcal{I}_S^{-1}\|_2$, $\alpha_3 := B_3 \|\mathcal{I}_S^{-1}\|_2^{3/2}$,

$$\kappa := \frac{\mathsf{Tr}(\mathcal{I}_T \mathcal{I}_S^{-1})}{\|\mathcal{I}_T^{\frac{1}{2}} \mathcal{I}_S^{-1} \mathcal{I}_T^{\frac{1}{2}}\|_2}, \quad \tilde{\kappa} := \frac{\mathsf{Tr}(\mathcal{I}_S^{-1})}{\|\mathcal{I}_S^{-1}\|_2}.$$

*Proof of Theorem 4.1.* We denote $g := \nabla \ell_n(\beta^\star) - \mathbb{E}[\nabla \ell_n(\beta^\star)]$. By taking $A = \mathcal{I}_S^{-1}$ in Assumption A.1, we have:

$$\|\mathcal{I}_S^{-1} g\|_2 \leq c \sqrt{\frac{\mathsf{Tr}(\mathcal{I}_S^{-1}) \log n}{n}} + B_1 \|\mathcal{I}_S^{-1}\|_2 \log^\gamma \left( \frac{B_1 \|\mathcal{I}_S^{-1}\|_2}{\sqrt{\mathsf{Tr}(\mathcal{I}_S^{-1})}} \right) \frac{\log n}{n}$$

$$= c \sqrt{\frac{\mathsf{Tr}(\mathcal{I}_S^{-1}) \log n}{n}} + B_1 \|\mathcal{I}_S^{-1}\|_2 \log^\gamma(\tilde{\kappa}^{-1/2} \alpha_1) \frac{\log n}{n}. \tag{10}$$

By Assumption A.1, A.2 and A.4, we have

$$\hat{L}(\beta) - \hat{L}(\beta^\star)$$
$$= \ell_n(\beta) - \ell_n(\beta^\star) + \lambda \left( R(\beta) - R(\beta^\star) \right)$$
$$\leq (\beta - \beta^\star)^T \nabla \ell_n(\beta^\star) + \frac{1}{2}(\beta - \beta^\star)^T \nabla^2 \ell_n(\beta^\star)(\beta - \beta^\star) + \frac{B_3}{6} \|\beta - \beta^\star\|_2^3 + \lambda \left( R(\beta) - R(\beta^\star) \right)$$
$$= (\beta - \beta^\star)^T g + \frac{1}{2}(\beta - \beta^\star)^T \nabla^2 \ell_n(\beta^\star)(\beta - \beta^\star) + \frac{B_3}{6} \|\beta - \beta^\star\|_2^3 + \lambda \left( R(\beta) - R(\beta^\star) \right)$$
$$\leq (\beta - \beta^\star)^T g + \frac{1}{2}(\beta - \beta^\star)^T \mathcal{I}_S (\beta - \beta^\star) + B_2 \sqrt{\frac{\log n}{n}} \|\beta - \beta^\star\|_2^2 + \frac{B_3}{6} \|\beta - \beta^\star\|_2^3$$
$$\quad + \lambda \left( R(\beta) - R(\beta^\star) \right)$$
$$\leq (\beta - \beta^\star)^T g + \frac{1}{2}(\beta - \beta^\star)^T \mathcal{I}_S (\beta - \beta^\star) + B_2 \sqrt{\frac{\log n}{n}} \|\beta - \beta^\star\|_2^2 + \frac{B_3}{6} \|\beta - \beta^\star\|_2^3$$
$$\quad + \lambda \left( \nabla R(\beta^\star)^\top (\beta - \beta^\star) + \frac{L}{2} \|\beta - \beta^\star\|_2^2 \right)$$
$$= \frac{1}{2}(\Delta_\beta - z)^T \mathcal{I}_S (\Delta_\beta - z) - \frac{1}{2} z^T \mathcal{I}_S z + \left( B_2 \sqrt{\frac{\log n}{n}} + \frac{\lambda L}{2} \right) \|\Delta_\beta\|_2^2 + \frac{B_3}{6} \|\Delta_\beta\|_2^3, \tag{11}$$

where $\Delta_\beta := \beta - \beta^\star$ and $z := -\mathcal{I}_S^{-1} g - \lambda \mathcal{I}_S^{-1} \nabla R(\beta^\star)$. Notice that $\Delta_{\beta^\star + z} = z$, by (10) and (11), we have

$$\hat{L}(\beta^\star + z) - \hat{L}(\beta^\star)$$
$$\leq -\frac{1}{2} z^T \mathcal{I}_S z$$
$$\quad + \left( B_2 \sqrt{\frac{\log n}{n}} + \frac{\lambda L}{2} \right) \left( c \sqrt{\frac{\mathsf{Tr}(\mathcal{I}_S^{-1}) \log n}{n}} + B_1 \|\mathcal{I}_S^{-1}\|_2 \log^\gamma(\tilde{\kappa}^{-1/2} \alpha_1) \frac{\log n}{n} + \lambda \left\| \mathcal{I}_S^{-1} \nabla R(\beta^\star) \right\|_2 \right)^2$$
$$\quad + \frac{B_3}{6} \left( c \sqrt{\frac{\mathsf{Tr}(\mathcal{I}_S^{-1}) \log n}{n}} + B_1 \|\mathcal{I}_S^{-1}\|_2 \log^\gamma(\tilde{\kappa}^{-1/2} \alpha_1) \frac{\log n}{n} + \lambda \left\| \mathcal{I}_S^{-1} \nabla R(\beta^\star) \right\|_2 \right)^3$$
$$\leq -\frac{1}{2} z^T \mathcal{I}_S z + \left( B_2 \sqrt{\frac{\log n}{n}} + \frac{\lambda L}{2} \right) \left( c \sqrt{\frac{\mathsf{Tr}(\mathcal{I}_S^{-1}) \log n}{n}} + \lambda \left\| \mathcal{I}_S^{-1} \nabla R(\beta^\star) \right\|_2 \right)^2$$
$$\quad + \frac{B_3}{6} \left( c \sqrt{\frac{\mathsf{Tr}(\mathcal{I}_S^{-1}) \log n}{n}} + \lambda \left\| \mathcal{I}_S^{-1} \nabla R(\beta^\star) \right\|_2 \right)^3$$
$$\leq -\frac{1}{2} z^T \mathcal{I}_S z + \left( 2 B_2 \sqrt{\frac{\log n}{n}} + \lambda L \right) \left( c \frac{\mathsf{Tr}(\mathcal{I}_S^{-1}) \log n}{n} + \lambda^2 \left\| \mathcal{I}_S^{-1} \nabla R(\beta^\star) \right\|_2^2 \right)$$
$$\quad + \frac{2 B_3}{3} \left( c \left( \frac{\mathsf{Tr}(\mathcal{I}_S^{-1}) \log n}{n} \right)^{3/2} + \lambda^3 \left\| \mathcal{I}_S^{-1} \nabla R(\beta^\star) \right\|_2^3 \right). \tag{12}$$

Here the second inequality holds as long as $n \geq N_4 (\log N_4)^2$ where

$$N_4 = c\tilde{\kappa}^{-1} B_1^2 \|\mathcal{I}_S^{-1}\|_2 \log^{2\gamma}(\tilde{\kappa}^{-1/2}\alpha_1),$$

and the last inequality follows from the fact that $(a+b)^n \leq 2^{n-1}(a^n + b^n)$.

Similarly, we have

$$\hat{L}(\beta) - \hat{L}(\beta^\star) \geq \frac{1}{2}(\Delta_\beta - z)^T \mathcal{I}_S (\Delta_\beta - z) - \frac{1}{2}z^T \mathcal{I}_S z - B_2\sqrt{\frac{\log n}{n}}\|\Delta_\beta\|_2^2 - \frac{B_3}{6}\|\Delta_\beta\|_2^3. \quad (13)$$

Thus, for any $\beta \in \mathcal{B}(\beta^\star, n^{-3/8})$, we have

$$\hat{L}(\beta) - \hat{L}(\beta^\star) \geq \frac{1}{2}(\Delta_\beta - z)^T \mathcal{I}_S (\Delta_\beta - z) - \frac{1}{2}z^T \mathcal{I}_S z - B_2\sqrt{\frac{\log n}{n}}n^{-\frac{3}{4}} - \frac{B_3}{6}n^{-\frac{9}{8}}. \quad (14)$$

(14) - (12) gives

$$\begin{aligned}
\hat{L}(\beta) &- \hat{L}(\beta^\star + z) \\
&\geq \frac{1}{2}(\Delta_\beta - z)^T \mathcal{I}_S (\Delta_\beta - z) \\
&\quad - \left(2B_2\sqrt{\frac{\log n}{n}} + \lambda L\right)\left(c\frac{\mathsf{Tr}(\mathcal{I}_S^{-1})\log n}{n} + \lambda^2 \left\|\mathcal{I}_S^{-1}\nabla R(\beta^\star)\right\|_2^2\right) \\
&\quad - \frac{2B_3}{3}\left(c\left(\frac{\mathsf{Tr}(\mathcal{I}_S^{-1})\log n}{n}\right)^{3/2} + \lambda^3 \left\|\mathcal{I}_S^{-1}\nabla R(\beta^\star)\right\|_2^3\right) \\
&\quad - B_2\sqrt{\frac{\log n}{n}}n^{-\frac{3}{4}} - \frac{B_3}{6}n^{-\frac{9}{8}} \\
&> \frac{1}{2}(\Delta_\beta - z)^T \mathcal{I}_S (\Delta_\beta - z) - B_3 n^{-\frac{9}{8}}. \quad (15)
\end{aligned}$$

Here the last inequality holds as long as $n \geq N_5$, where

$$N_5 = c \cdot \max\left\{B_2^9 B_3^{-9},\ \mathsf{Tr}(\mathcal{I}_S^{-1})^{9/2},\ c_\lambda^9 \Delta^{-9}\left\|\mathcal{I}_S^{-1}\nabla R(\beta^\star)\right\|_2^9,\ B_2^3 B_3^{-3}\mathsf{Tr}(\mathcal{I}_S^{-1})^3, \right.$$
$$\left. B_2^3 B_3^{-3}c_\lambda^6 \Delta^{-6}\left\|\mathcal{I}_S^{-1}\nabla R(\beta^\star)\right\|_2^6,\ L^3 c_\lambda^3 \Delta^{-3}\mathsf{Tr}(\mathcal{I}_S^{-1})^3,\ L^3 c_\lambda^9 \Delta^{-9}\left\|\mathcal{I}_S^{-1}\nabla R(\beta^\star)\right\|_2^6\right\}.$$

Consider the ellipsoid

$$\mathcal{D} := \left\{\beta \in \mathbb{R}^d \,\middle|\, \frac{1}{2}(\Delta_\beta - z)^T \mathcal{I}_S (\Delta_\beta - z) \leq B_3 n^{-\frac{9}{8}}\right\}.$$

Then by (15), for any $\beta \in \mathcal{B}(\beta^\star, n^{-3/8}) \cap \mathcal{D}^C$,

$$\hat{L}(\beta) - \hat{L}(\beta^\star + z) > 0. \quad (16)$$

Notice that by the definition of $\mathcal{D}$, using $\lambda_{\min}^{-1}(\mathcal{I}_S) = \|\mathcal{I}_S^{-1}\|_2$, we have for any $\beta \in \mathcal{D}$,

$$\|\Delta_\beta - z\|_2^2 \leq 2B_3 \|\mathcal{I}_S^{-1}\|_2 n^{-\frac{9}{8}}.$$

Thus for any $\beta \in \mathcal{D}$, we have

$$\begin{aligned}
\|\Delta_\beta\|_2^2 &\leq 2(\|\Delta_\beta - z\|_2^2 + \|z\|_2^2) \\
&\leq 4B_3 \|\mathcal{I}_S^{-1}\|_2 n^{-\frac{9}{8}} + 2\left(c\sqrt{\frac{\mathsf{Tr}(\mathcal{I}_S^{-1})\log n}{n}} + \lambda \left\|\mathcal{I}_S^{-1}\nabla R(\beta^\star)\right\|_2\right)^2 \\
&\leq 3\left(c\sqrt{\frac{\mathsf{Tr}(\mathcal{I}_S^{-1})\log n}{n}} + \lambda \left\|\mathcal{I}_S^{-1}\nabla R(\beta^\star)\right\|_2\right)^2, \quad (17)
\end{aligned}$$

where the last inequality holds as long as $n \geq N_6 = c(B_3 \tilde{\kappa}^{-1})^8$. It then holds that

$$\|\Delta_\beta\|_2 \leq 2c\sqrt{\frac{\mathsf{Tr}(\mathcal{I}_S^{-1})\log n}{n}} + 2\lambda \left\|\mathcal{I}_S^{-1}\nabla R(\beta^\star)\right\|_2 \leq n^{-3/8},$$

where the last inequality holds as long as $n \geq N_5$. In other words, we show that $\mathcal{D} \subset \mathcal{B}(\beta^\star, n^{-3/8})$.

Recall that by Lemma C.5, we have

$$\hat{\beta}_\lambda \in \mathcal{B}(\beta^\star, D'') \subset \mathcal{B}(\beta^\star, n^{-3/8}).$$

Also, for any $\beta \in \mathcal{B}(\beta^\star, n^{-3/8}) \cap \mathcal{D}^C$, we have

$$\hat{L}(\beta) - \hat{L}(\beta^\star + z) > 0.$$

Consequently, we conclude

$$\hat{\beta}_\lambda \in \mathcal{B}(\beta^\star, D'') \cap \mathcal{D}.$$

By the definition of $\mathcal{D}$, we have

$$\left\|\mathcal{I}_S^{1/2}(\Delta_{\hat{\beta}_\lambda} - z)\right\|_2^2 \leq 2B_3 n^{-\frac{9}{8}} \tag{18}$$

By (17), we further have

$$\|\hat{\beta}_\lambda - \beta^\star\|_2 \leq 2c\sqrt{\frac{\mathsf{Tr}(\mathcal{I}_S^{-1})\log n}{n}} + 2\lambda \left\|\mathcal{I}_S^{-1}\nabla R(\beta^\star)\right\|_2. \tag{19}$$

Note that by taking $A = \mathcal{I}_T^{\frac{1}{2}}\mathcal{I}_S^{-1}$ in Assumption A.1, we have

$$\|\mathcal{I}_T^{\frac{1}{2}}\mathcal{I}_S^{-1}g\|_2 \leq c\sqrt{\frac{\mathsf{Tr}(\mathcal{I}_S^{-1}\mathcal{I}_T)\log n}{n}} + B_1\|\mathcal{I}_T^{\frac{1}{2}}\mathcal{I}_S^{-1}\|_2 \log^\gamma\left(\frac{B_1\|\mathcal{I}_T^{\frac{1}{2}}\mathcal{I}_S^{-1}\|_2}{\sqrt{\mathsf{Tr}(\mathcal{I}_S^{-1}\mathcal{I}_T)}}\right)\frac{\log n}{n}$$

$$\leq c\sqrt{\frac{\mathsf{Tr}(\mathcal{I}_S^{-1}\mathcal{I}_T)\log n}{n}} + B_1\|\mathcal{I}_T^{\frac{1}{2}}\mathcal{I}_S^{-1}\|_2 \log^\gamma(\kappa^{-1/2}\alpha_1)\frac{\log n}{n}. \tag{20}$$

Thus, we have

$\|\mathcal{I}_T^{\frac{1}{2}}(\hat{\beta}_\lambda - \beta^\star)\|_2^2$

$= \|\mathcal{I}_T^{\frac{1}{2}}\Delta_{\hat{\beta}_\lambda}\|_2^2$

$= \|\mathcal{I}_T^{\frac{1}{2}}(\Delta_{\hat{\beta}_\lambda} - z) + \mathcal{I}_T^{\frac{1}{2}}z\|_2^2$

$\leq 2\|\mathcal{I}_T^{\frac{1}{2}}(\Delta_{\hat{\beta}_\lambda} - z)\|_2^2 + 2\|\mathcal{I}_T^{\frac{1}{2}}z\|_2^2$

$= 2\|\mathcal{I}_T^{\frac{1}{2}}\mathcal{I}_S^{-\frac{1}{2}}(\mathcal{I}_S^{\frac{1}{2}}(\Delta_{\hat{\beta}_\lambda} - z))\|_2^2 + 4\|\mathcal{I}_T^{\frac{1}{2}}\mathcal{I}_S^{-1}g\|_2^2 + 4\lambda^2\|\mathcal{I}_T^{\frac{1}{2}}\mathcal{I}_S^{-1}\nabla R(\beta^\star)\|_2^2$

$\leq 2\|\mathcal{I}_T^{\frac{1}{2}}\mathcal{I}_S^{-\frac{1}{2}}\|_2^2\|\mathcal{I}_S^{\frac{1}{2}}(\Delta_{\hat{\beta}_\lambda} - z)\|_2^2 + 4\|\mathcal{I}_T^{\frac{1}{2}}\mathcal{I}_S^{-1}g\|_2^2 + 4\lambda^2\|\mathcal{I}_T^{\frac{1}{2}}\mathcal{I}_S^{-1}\nabla R(\beta^\star)\|_2^2$

$\leq 4\left(c\sqrt{\frac{\mathsf{Tr}(\mathcal{I}_S^{-1}\mathcal{I}_T)\log n}{n}} + B_1\|\mathcal{I}_T^{\frac{1}{2}}\mathcal{I}_S^{-1}\|_2\log^\gamma(\kappa^{-1/2}\alpha_1)\frac{\log n}{n}\right)^2 + 4\lambda^2\|\mathcal{I}_T^{\frac{1}{2}}\mathcal{I}_S^{-1}\nabla R(\beta^\star)\|_2^2$

$\quad + 4B_3\|\mathcal{I}_T^{\frac{1}{2}}\mathcal{I}_S^{-\frac{1}{2}}\|_2^2 n^{-\frac{9}{8}}$

$\leq c\left(\frac{\mathsf{Tr}(\mathcal{I}_S^{-1}\mathcal{I}_T)\log n}{n} + \lambda^2\|\mathcal{I}_T^{\frac{1}{2}}\mathcal{I}_S^{-1}\nabla R(\beta^\star)\|_2^2\right). \tag{21}$

Here the last inequality holds as long as $n \geq \max\{N_7(\log N_7)^2, N_8\}$, where

$$N_7 = cB_1^2\|\mathcal{I}_S^{-1}\|_2\kappa^{-1}\log^{2\gamma}(\kappa^{-1/2}\alpha_1), \quad N_8 = cB_3^9\kappa^{-9}.$$

Finally, we have

$$
\mathcal{E}(\hat{\beta}_\lambda) = \mathbb{E}_T \left[ \ell(x, y, \hat{\beta}_\lambda) - \ell(x, y, \beta^\star) \right]
$$

$$
\leq \mathbb{E}_T[\nabla \ell(x, y, \beta^\star)]^T (\hat{\beta}_\lambda - \beta^\star) + \frac{1}{2}(\hat{\beta}_\lambda - \beta^\star)^T \mathcal{I}_T (\hat{\beta}_\lambda - \beta^\star) + \frac{B_3}{6}\|\hat{\beta}_\lambda - \beta^\star\|_2^3
$$

$$
= \frac{1}{2}(\hat{\beta}_\lambda - \beta^\star)^T \mathcal{I}_T (\hat{\beta}_\lambda - \beta^\star) + \frac{B_3}{6}\|\hat{\beta}_\lambda - \beta^\star\|_2^3
$$

$$
\leq c \left( \frac{\mathsf{Tr}(\mathcal{I}_S^{-1}\mathcal{I}_T)\log n}{n} + \lambda^2 \|\mathcal{I}_T^{\frac{1}{2}}\mathcal{I}_S^{-1}\nabla R(\beta^\star)\|_2^2 \right)
$$

$$
+ \frac{B_3}{6} \left( 2c\sqrt{\frac{\mathsf{Tr}(\mathcal{I}_S^{-1})\log n}{n}} + 2\lambda \left\|\mathcal{I}_S^{-1}\nabla R(\beta^\star)\right\|_2 \right)^3
$$

$$
\leq c \left( \frac{\mathsf{Tr}(\mathcal{I}_S^{-1}\mathcal{I}_T)\log n}{n} + \lambda^2 \|\mathcal{I}_T^{\frac{1}{2}}\mathcal{I}_S^{-1}\nabla R(\beta^\star)\|_2^2 \right).
$$

Here the last inequality holds as long as $n \geq N_9(\log N_9)^2$, where

$$
N_9 = c \max \left\{ B_3^2 \mathsf{Tr}(\mathcal{I}_S^{-1})^3 \mathsf{Tr}(\mathcal{I}_S^{-1}\mathcal{I}_T)^{-2}, \ B_3^2 c_\lambda^2 \Delta^{-2} \left\|\mathcal{I}_S^{-1}\nabla R(\beta^\star)\right\|_2^6 \left\|\mathcal{I}_T^{\frac{1}{2}}\mathcal{I}_S^{-1}\nabla R(\beta^\star)\right\|_2^{-4} \right\}.
$$

We then finish the proofs.

In the end, we summarize the threshold of $n$. We require $n \geq cN^\star$, where

$$
N^\star = \max \left\{ N_1^\star (\log N_1^\star)^2, \ N_2^\star \right\}, \tag{22}
$$

$$
N_1^\star = \max \Bigg\{ \left( \alpha^{-2}D^{-4} + G^{-2} \right)\left( R(\beta^\star)^2 c_\lambda^2 \Delta^{-2} + B_0^2 \right), \ \alpha^{-2}B_2^2, \alpha^{-4}c_\lambda^2 L^2 \|\beta^\star\|_2^2 B_3^2 \Delta^{-2},
$$

$$
\alpha^{-6}B_0^2 B_3^4, \tilde{\kappa}^{-1}B_1^2\|\mathcal{I}_S^{-1}\|_2 \log^{2\gamma}(\tilde{\kappa}^{-1/2}\alpha_1), \ B_1^2\|\mathcal{I}_S^{-1}\|_2 \kappa^{-1} \log^{2\gamma}(\kappa^{-1/2}\alpha_1),
$$

$$
B_3^2 \mathsf{Tr}(\mathcal{I}_S^{-1})^3 \mathsf{Tr}(\mathcal{I}_S^{-1}\mathcal{I}_T)^{-2}, \ B_3^2 c_\lambda^2 \Delta^{-2} \left\|\mathcal{I}_S^{-1}\nabla R(\beta^\star)\right\|_2^6 \left\|\mathcal{I}_T^{\frac{1}{2}}\mathcal{I}_S^{-1}\nabla R(\beta^\star)\right\|_2^{-4} \Bigg\}
$$

$$
= \max \Bigg\{ \alpha^{-2}B_0^2 B_S^4 L^4 \Delta^{-6}R(\beta^\star)^2, \ \alpha^{-6}B_0^2 B_3^4 \Delta^{-2}R(\beta^\star)^2, \ G^{-2}B_0^2 \Delta^{-2}R(\beta^\star)^2, \ \alpha^{-2}B_2^2,
$$

$$
\alpha^{-4}B_0^2 B_3^2 L^2 \Delta^{-2}\|\beta^\star\|_2^2, \ \alpha^{-6}B_0^2 B_3^4, \ \tilde{\kappa}^{-1}\alpha_1^2 \log^{2\gamma}(\tilde{\kappa}^{-1/2}\alpha_1), \ \alpha_1^2 \kappa^{-1} \log^{2\gamma}(\kappa^{-1/2}\alpha_1),
$$

$$
B_3^2 \kappa^{-2}\tilde{\kappa}^3 \|\mathcal{I}_S^{-1}\|_2^3 \|\mathcal{I}_T \mathcal{I}_S^{-1}\|_2^{-2}, \ B_0^2 B_3^2 \Delta^{-2} \left\|\mathcal{I}_S^{-1}\nabla R(\beta^\star)\right\|_2^6 \left\|\mathcal{I}_T^{\frac{1}{2}}\mathcal{I}_S^{-1}\nabla R(\beta^\star)\right\|_2^{-4} \Bigg\},
$$

$$
N_2^\star = \max \Bigg\{ B_2^9 B_3^{-9}, \ \tilde{\kappa}^{9/2}\|\mathcal{I}_S^{-1}\|_2^{9/2}, \ B_0^9 \Delta^{-9} \left\|\mathcal{I}_S^{-1}\nabla R(\beta^\star)\right\|_2^9, \ B_2^3 B_3^{-3}\tilde{\kappa}^3\|\mathcal{I}_S^{-1}\|_2^3,
$$

$$
B_0^6 B_2^3 B_3^{-3}\Delta^{-6} \left\|\mathcal{I}_S^{-1}\nabla R(\beta^\star)\right\|_2^6, B_0^3 L^3 \Delta^{-3}\tilde{\kappa}^3\|\mathcal{I}_S^{-1}\|_2^3, \ B_0^9 L^3 \Delta^{-9} \left\|\mathcal{I}_S^{-1}\nabla R(\beta^\star)\right\|_2^6,
$$

$$
(B_3\tilde{\kappa}^{-1})^8, \ B_3^9 \kappa^{-9} \Bigg\}.
$$

Here we denote $\mathcal{I}_S := \mathcal{I}_S(\beta^\star)$, $\mathcal{I}_T := \mathcal{I}_T(\beta^\star)$, $\alpha_1 := B_1\|\mathcal{I}_S^{-1}\|_2^{1/2}$, $\alpha_2 := B_2\|\mathcal{I}_S^{-1}\|_2$, $\alpha_3 := B_3\|\mathcal{I}_S^{-1}\|_2^{3/2}$,

$$
\kappa := \frac{\mathsf{Tr}(\mathcal{I}_T \mathcal{I}_S^{-1})}{\|\mathcal{I}_T^{\frac{1}{2}}\mathcal{I}_S^{-1}\mathcal{I}_T^{\frac{1}{2}}\|_2}, \ \tilde{\kappa} := \frac{\mathsf{Tr}(\mathcal{I}_S^{-1})}{\|\mathcal{I}_S^{-1}\|_2}.
$$

$\square$

## C.2 PROOFS OF THEOREM 4.2

Before proceeding to the proof of Theorem 4.2, we first recall several definitions, clarify notation, and establish a few useful propositions that will be used in the proof.

We adopt the same notation as in the proof of Theorem 4.1; specifically, we denote: $\mathcal{I}_S := \mathcal{I}_S(\beta^\star)$, $\mathcal{I}_T := \mathcal{I}_T(\beta^\star)$, $\alpha_1 := B_1 \|\mathcal{I}_S^\dagger\|_2^{1/2}$, $\alpha_2 := B_2 \|\mathcal{I}_S^\dagger\|_2$, $\alpha_3 := B_3 \|\mathcal{I}_S^\dagger\|_2^{3/2}$,

$$\kappa := \frac{\mathsf{Tr}(\mathcal{I}_T \mathcal{I}_S^\dagger)}{\|\mathcal{I}_T^{\frac{1}{2}} \mathcal{I}_S^\dagger \mathcal{I}_T^{\frac{1}{2}}\|_2}, \quad \tilde{\kappa} := \frac{\mathsf{Tr}(\mathcal{I}_S^\dagger)}{\|\mathcal{I}_S^\dagger\|_2}.$$

Let

$$\lambda = c_\lambda n^{-2\delta} (\log n)^{\frac{1}{2}}, \quad \text{where } c_\lambda = \frac{8 B_0}{\Delta_{\max}}, \tag{23}$$

$$\Delta_1 = c_1 n^{-\delta} (\log n)^{\frac{1}{4}}, \quad \text{where } c_1 = 4\sqrt{\alpha^{-1} \max\{c_\lambda R(\beta^\star), B_0\}}, \tag{24}$$

$$\Delta_2 = n^{-\delta + \frac{1}{12}} (\log n)^{\frac{1}{4}}, \tag{25}$$

$$\delta = \frac{1}{4} - \frac{1}{6\tau} < \frac{1}{4}. \tag{26}$$

For any $\beta_0 \in \mathcal{B}_S$, the tangent space at $\beta_0$ is defined as

$$\mathcal{T}(\beta_0) := \{r'(0) \mid r(t) : (-1, 1) \to \mathcal{B}_S, \ r(0) = \beta_0\}.$$

The following proposition establishes a connection between the tangent space $\mathcal{T}(\beta_0)$ and the null space of the Hessian $\mathbb{E}_S[\nabla^2 \ell(x, y, \beta_0)]$, denoted by $\mathrm{null}(\mathbb{E}_S[\nabla^2 \ell(x, y, \beta_0)])$.

**Proposition C.6.** *Under Assumptions C.1 and C.2, we have that for any $\beta_0 \in \mathcal{B}_S$,*

$$\mathcal{T}(\beta_0) = \mathrm{null}\left(\mathbb{E}_S[\nabla^2 \ell(x, y, \beta_0)]\right).$$

*Proof of Proposition C.6.* Fix $\beta_0 \in \mathcal{B}_S$. We begin by showing that

$$\mathcal{T}(\beta_0) \subseteq \mathrm{null}\left(\mathbb{E}_S[\nabla^2 \ell(x, y, \beta_0)]\right).$$

Let $v \in \mathcal{T}(\beta_0)$. By the definition of the tangent space, there exists a smooth curve $r(t) : (-1, 1) \to \mathcal{B}_S$ such that

$$r(0) = \beta_0, \quad r'(0) = v. \tag{27}$$

Since $r(t) \in \mathcal{B}_S$ for all $t \in (-1, 1)$, and by the definition of $\mathcal{B}_S$, we have

$$\mathbb{E}_S[\ell(x, y, r(t))] = \mathbb{E}_S[\ell(x, y, \beta^\star)], \quad \forall t \in (-1, 1).$$

Differentiating both sides with respect to $t$, we obtain

$$0 = \frac{d}{dt} \mathbb{E}_S[\ell(x, y, r(t))] = \langle \mathbb{E}_S[\nabla \ell(x, y, r(t))], r'(t) \rangle, \quad \forall t \in (-1, 1).$$

Differentiating once more, we get

$$0 = \frac{d}{dt} \langle \mathbb{E}_S[\nabla \ell(x, y, r(t))], r'(t) \rangle$$
$$= r'(t)^\top \mathbb{E}_S[\nabla^2 \ell(x, y, r(t))] r'(t) + \langle \mathbb{E}_S[\nabla \ell(x, y, r(t))], r''(t) \rangle.$$

Evaluating at $t = 0$ and using $r'(0) = v$ and $\mathbb{E}_S[\nabla \ell(x, y, \beta_0)] = 0$, we obtain

$$v^\top \mathbb{E}_S[\nabla^2 \ell(x, y, \beta_0)] v = 0,$$

which implies $v \in \mathrm{null}(\mathbb{E}_S[\nabla^2 \ell(x, y, \beta_0)])$. Therefore,

$$\mathcal{T}(\beta_0) \subseteq \mathrm{null}\left(\mathbb{E}_S[\nabla^2 \ell(x, y, \beta_0)]\right).$$

By Assumptions C.1 and C.2, we have

$$\dim\left(\mathrm{null}(\mathbb{E}_S[\nabla^2 \ell(x, y, \beta_0)])\right) = d_S = \dim(\mathcal{B}_S) = \dim(\mathcal{T}(\beta_0)).$$

Hence, the inclusion is actually an equality:

$$\mathcal{T}(\beta_0) = \mathrm{null}\left(\mathbb{E}_S[\nabla^2 \ell(x, y, \beta_0)]\right),$$

which completes the proof. $\square$

We denote the column space of the Hessian $\mathbb{E}_S[\nabla^2 \ell(x, y, \beta_0)]$ by $\mathsf{col}(\mathbb{E}_S[\nabla^2 \ell(x, y, \beta_0)])$. The following proposition characterizes the strong convexity of the population loss along directions within the column space at each point $\beta_0 \in \mathcal{B}_S$.

**Proposition C.7.** *For any $\beta_0 \in \mathcal{B}_S$ and any unit vector $v \in \mathsf{col}(\mathbb{E}_S[\nabla^2 \ell(x, y, \beta_0)])$, we have*

$$\mathbb{E}_S\left[\ell(x, y, \beta_0 + tv)\right] \geq \mathbb{E}_S\left[\ell(x, y, \beta_0)\right] + \frac{\alpha}{4}t^2, \quad \forall t \in \left[-\frac{\alpha}{2B_3}, \frac{\alpha}{2B_3}\right].$$

*Proof of Proposition C.7.* Fix $\beta_0 \in \mathcal{B}_S$ and a unit vector $v \in \mathsf{col}(\mathbb{E}_S[\nabla^2 \ell(x, y, \beta_0)])$. By Assumption A.2, for all $t \in \left[-\frac{\alpha}{2B_3}, \frac{\alpha}{2B_3}\right]$, we have

$$\left\|\mathbb{E}_S[\nabla^2 \ell(x, y, \beta_0 + tv)] - \mathbb{E}_S[\nabla^2 \ell(x, y, \beta_0)]\right\| \leq B_3 \cdot |t| \leq B_3 \cdot \frac{\alpha}{2B_3} = \frac{\alpha}{2}.$$

Moreover, by Assumption C.2, we know that

$$\lambda_{\min}\left(\mathbb{E}_S[\nabla^2 \ell(x, y, \beta_0)]\right) \geq \alpha.$$

Combining these two facts, we conclude that for all $t \in \left[-\frac{\alpha}{2B_3}, \frac{\alpha}{2B_3}\right]$, the Hessian along direction $v$ satisfies

$$v^\top \mathbb{E}_S[\nabla^2 \ell(x, y, \beta_0 + tv)]v \geq \alpha - \frac{\alpha}{2} = \frac{\alpha}{2}.$$

Therefore, the function $t \mapsto \mathbb{E}_S[\ell(x, y, \beta_0 + tv)]$ is $\frac{\alpha}{2}$-strongly convex in $t$, and standard properties of strongly convex functions yield:

$$\mathbb{E}_S[\ell(x, y, \beta_0 + tv)] \geq \mathbb{E}_S[\ell(x, y, \beta_0)] + \frac{\alpha}{4}t^2.$$

This completes the proof. $\qquad\square$

It is worth noting that Proposition C.4 continues to hold in this setting.

For any $\beta \in \mathbb{R}^d$, we define the distance from $\beta$ to the set $\mathcal{B}_S$ as

$$\mathrm{dist}(\beta, \mathcal{B}_S) := \min_{\beta_0 \in \mathcal{B}_S} \|\beta - \beta_0\|_2.$$

We then define the set

$$\mathcal{A}_S := \left\{\beta \in \mathbb{R}^d \mid \mathrm{dist}(\beta, \mathcal{B}_S) \leq \Delta_1\right\} \supset \mathcal{B}_S.$$

Since $\mathcal{B}_S$ is compact and $\mathrm{dist}(\cdot, \mathcal{B}_S)$ is continuous, it follows that $\mathcal{A}_S$ is also compact. The following claim characterizes the boundary of $\mathcal{A}_S$.

**Claim C.8.** *The boundary of $\mathcal{A}_S$ satisfies*

$$\partial \mathcal{A}_S \subset \left\{\beta \in \mathbb{R}^d \mid \mathrm{dist}(\beta, \mathcal{B}_S) = \Delta_1\right\}.$$

*Proof of Claim C.8.* Since $\mathcal{A}_S$ is closed, we have $\partial \mathcal{A}_S = \mathcal{A}_S \setminus \mathrm{int}(\mathcal{A}_S)$.

If $\beta \in \partial \mathcal{A}_S$, then $\beta \in \mathcal{A}_S$ but $\beta \notin \mathrm{int}(\mathcal{A}_S)$. Hence $\mathrm{dist}(\beta, \mathcal{B}_S) \leq \Delta_1$, but for any $\varepsilon > 0$, the ball $B(\beta, \varepsilon)$ is not fully contained in $\mathcal{A}_S$, so $\mathrm{dist}(\beta, \mathcal{B}_S)$ cannot be strictly less than $\Delta_1$. Thus $\mathrm{dist}(\beta, \mathcal{B}_S) = \Delta_1$.

$\qquad\square$

Combining Proposition C.6, Proposition C.7, and Claim C.8, we obtain the following lemma.

**Lemma C.9.** *Suppose that $n \geq cN_1'$, where*

$$N_1' := \left(\frac{c_1 B_3}{\alpha}\right)^{\frac{12\tau}{3\tau-2}} = \max\left\{\left(\frac{B_0 B_3^2}{\alpha^3}\right)^{\frac{6\tau}{3\tau-2}}, \left(\frac{B_0 B_3^2 R(\beta^\star)}{\alpha^3 \Delta_{\max}}\right)^{\frac{6\tau}{3\tau-2}}\right\}. \tag{28}$$

*Then, for all $\beta \in \partial \mathcal{A}_S$, it holds that*

$$\mathbb{E}_S[\ell(x, y, \beta)] \geq \mathbb{E}_S[\ell(x, y, \beta^\star)] + \frac{\alpha}{4}\Delta_1^2.$$

*Proof of Lemma C.9.* Fix $\beta \in \partial \mathcal{A}_S$, and define

$$\beta' := \arg\min_{\beta_0 \in \mathcal{B}_S} \|\beta - \beta_0\|_2.$$

Since $\mathcal{B}_S$ is closed, we have $\beta' \in \mathcal{B}_S$ and by definition $\|\beta - \beta'\|_2 = \Delta_1$.

We now show that $\beta - \beta' \in \mathcal{T}(\beta')^\perp$, where $\mathcal{T}(\beta')^\perp$ denotes the orthogonal complement of the tangent space at $\beta'$, defined as

$$\mathcal{T}(\beta')^\perp := \left\{ u \in \mathbb{R}^d \mid u^\top v = 0 \quad \forall v \in \mathcal{T}(\beta') \right\}.$$

Let $f(x) := \|x - \beta\|_2^2$. Then $\beta'$ minimizes $f$ over $\mathcal{B}_S$, i.e.,

$$\beta' = \arg\min_{x \in \mathcal{B}_S} f(x).$$

Since $f$ is smooth, the first-order optimality condition implies that its directional derivative vanishes along directions in the tangent space:

$$2\langle \beta - \beta', v \rangle = 0, \quad \forall v \in \mathcal{T}(\beta').$$

Therefore,

$$\beta - \beta' \in \mathcal{T}(\beta')^\perp.$$

By Proposition C.6, we know

$$\mathcal{T}(\beta') = \text{null}\left( \mathbb{E}_S[\nabla^2 \ell(x, y, \beta')] \right),$$

and thus

$$\mathcal{T}(\beta')^\perp = \text{col}\left( \mathbb{E}_S[\nabla^2 \ell(x, y, \beta')] \right).$$

It follows that

$$\beta - \beta' \in \text{col}\left( \mathbb{E}_S[\nabla^2 \ell(x, y, \beta')] \right).$$

Note that as long as $n \geq cN_1'$, we have $0 < \Delta_1 \leq \frac{\alpha}{2B_3}$.

Now apply Proposition C.7 to the direction $v := \frac{\beta - \beta'}{\|\beta - \beta'\|_2}$ (which is a unit vector in the column space):

$$\begin{aligned}
\mathbb{E}_S[\ell(x, y, \beta)] = \mathbb{E}_S\left[\ell\left(x, y, \beta' + \Delta_1 \cdot v\right)\right] \\
\geq \mathbb{E}_S[\ell(x, y, \beta')] + \frac{\alpha}{4}\Delta_1^2 \\
= \mathbb{E}_S[\ell(x, y, \beta^\star)] + \frac{\alpha}{4}\Delta_1^2,
\end{aligned}$$

where the last equality uses the fact that $\beta' \in \mathcal{B}_S$, and all elements in $\mathcal{B}_S$ achieve the same population loss as $\beta^\star$.

This completes the proof. $\qquad\square$

Recall the definition of $A(n)$ from (2). By the choice of $\lambda, \Delta_1$ given in (23) and (24), we have

$$\mathbb{E}_S[\ell(x, y, \beta^\star)] + \frac{\alpha}{4}\Delta_1^2 > \mathbb{E}_S[\ell(x, y, \beta^\star)] + \lambda R(\beta^\star) + 2B_0\sqrt{\frac{\log n}{n}} = A(n).$$

As a result, by Lemma C.9, for all $\beta \in \partial \mathcal{A}_S$, it holds that

$$\mathbb{E}_S[\ell(x, y, \beta)] > A(n).$$

Similar to Lemma C.2, we can establish the following result:

**Lemma C.10.** *Suppose that* $n \geq c \cdot \max\left\{N_1', \ N_2'\right\}$, *where*

$$N_2' = \max\left\{ \left(\frac{c_\lambda R(\beta^\star)}{G}\right)^{\frac{12\tau}{5\tau - 4}}, \frac{B_0^3}{G^3} \right\} = \max\left\{ \left(\frac{B_0 R(\beta^\star)}{\Delta_{\max} G}\right)^{\frac{12\tau}{5\tau - 4}}, \frac{B_0^3}{G^3} \right\}. \qquad (29)$$

*Then, for all* $\beta \notin \mathcal{A}_S$, *it holds that* $\mathbb{E}_S[\ell(x, y, \beta)] > A(n)$.

*Proof of Lemma C.10.* We prove the result by contradiction. Suppose there exists some $\beta \notin \mathcal{A}_S$ such that

$$\mathbb{E}_S[\ell(x, y, \beta)] \leq A(n).$$

Recall Assumption A.3. Define the set $\Omega := \mathcal{B}(0, B) \setminus \mathcal{A}_S$. Note that by Assumption A.3, for all $\|\beta\|_2 \geq B$, we have

$$\mathbb{E}_S[\ell(x, y, \beta)] \geq \mathbb{E}_S[\ell(x, y, \beta^\star)] + G > A(n),$$

where the last inequality holds as long as $n \geq c \cdot N_2'$. This means there exists $\beta \in \Omega$ such that $\mathbb{E}_S[\ell(x, y, \beta)] \leq A(n)$.

From Lemma C.9, we know that for all $\beta \in \partial\Omega$,

$$\mathbb{E}_S[\ell(x, y, \beta)] > A(n).$$

This implies the existence of a local minimum of $\mathbb{E}_S[\ell(x, y, \beta)]$ in $\Omega$. Let $\beta' \in \Omega$ be such a local minimizer.

We then observe:

$$\mathbb{E}_S[\ell(x, y, \beta')] \leq A(n) = \mathbb{E}_S[\ell(x, y, \beta^\star)] + \lambda R(\beta^\star) + 2B_0\sqrt{\frac{\log n}{n}}$$

$$< \mathbb{E}_S[\ell(x, y, \beta^\star)] + G,$$

where the last inequality holds when $n \geq c \cdot N_2'$.

Therefore, $\beta'$ must be a global minimizer of the population loss, implying $\beta' \in \mathcal{B}_S \subset \mathcal{A}_S$, which contradicts the fact that $\beta' \in \Omega$.

This contradiction completes the proof. $\qquad\square$

Combining Lemma C.10 with (3), we conclude that

$$\hat{\beta}_\lambda \in \mathcal{A}_S. \tag{30}$$

Next, we establish the following lemma, which further refines the region in which $\hat{\beta}_\lambda$ must lie:

**Lemma C.11.** *Suppose that $n \geq c \cdot N_3'$, where*

$$N_3' = \max\Bigg\{ (c_1 L B_S)^{12}, \alpha_1^3 \log^{3\gamma}(\alpha_1), \left(c_1^2 \mathsf{Tr}(\mathcal{I}_S)\right)^6, \left(c_1^2 B_2\right)^4, \left(c_1^3 B_3\right)^{12}, \left(\mathsf{Tr}(\mathcal{I}_S)\right)^{3/2},$$

$$B_2^4, B_3^2 \Bigg\}. \tag{31}$$

*Then, for all*

$$\beta \in \bigcup_{\substack{\beta_0 \in \mathcal{B}_S \\ R(\beta_0) - R(\beta^\star) \geq \Delta_2}} \mathcal{B}(\beta_0, \Delta_1)$$

*it holds that*

$$\hat{L}(\beta) > \hat{L}(\beta^\star).$$

*Proof of Lemma C.11.* By the definition of $N_3'$, we have

$$N_3' \geq \max\Bigg\{ (c_1 L B_S)^{12}, \frac{B_1^3}{\mathsf{Tr}(\mathcal{I}_S)^{3/2}} \log^{3\gamma}\left(\frac{B_1}{\sqrt{\mathsf{Tr}(\mathcal{I}_S)}}\right), \left(\frac{c_1\sqrt{\mathsf{Tr}(\mathcal{I}_S)}}{c_\lambda}\right)^{12}, \left(\frac{c_1^2 B_2}{c_\lambda}\right)^4, \left(\frac{c_1^3 B_3}{c_\lambda}\right)^{12},$$

$$\left(\frac{\Delta_{\max}^{\tau-1}\sqrt{\mathsf{Tr}(\mathcal{I}_S)}}{c_\lambda}\right)^3, \left(\frac{\Delta_{\max}^{2\tau-1} B_2}{c_\lambda}\right)^4, \left(\frac{\Delta_{\max}^{3\tau-1} B_3}{c_\lambda}\right)^2 \Bigg\}.$$

We start by proving the following proposition.

**Proposition C.12.** *Suppose that $\beta_0 \in \mathcal{B}_S \setminus \{\beta^\star\}$. Then for all $\beta \in \mathcal{B}\left(\beta_0, \frac{R(\beta_0)-R(\beta^\star)}{4LB_S}\right)$, we have*

$$\mathbb{E}_S[\ell(x,y,\beta^\star)] + \lambda R(\beta^\star) + \frac{\lambda}{2}\left(R(\beta_0) - R(\beta^\star)\right) < \mathbb{E}_S[\ell(x,y,\beta)] + \lambda R(\beta).$$

*Proof of Proposition C.12.* Let $\beta \in \mathcal{B}\left(\beta_0, \frac{R(\beta_0)-R(\beta^\star)}{4LB_S}\right)$ for some $\beta_0 \in \mathcal{B}_S \setminus \{\beta^\star\}$. By Assumption A.4, we then have

$$
\begin{aligned}
R(\beta) &\geq R(\beta_0) + \nabla R(\beta_0)^\top (\beta - \beta_0) \\
&\geq R(\beta_0) - \|\nabla R(\beta_0)\|_2 \|\beta - \beta_0\|_2 \\
&\geq R(\beta_0) - LB_S \|\beta - \beta_0\|_2 \\
&\geq R(\beta_0) - \frac{R(\beta_0) - R(\beta^\star)}{4},
\end{aligned}
$$

where the first inequality follows from the convexity of $R(\beta)$ and the third inequality follows from the fact that $\nabla R(0) = 0$ and thus $\|\nabla R(\beta_0)\|_2 \leq L\|\beta_0\|_2 \leq LB_S$.

Thus, we have

$$
\begin{aligned}
R(\beta) - R(\beta^\star) &\geq \frac{3}{4}\left(R(\beta_0) - R(\beta^\star)\right) \\
&> \frac{1}{2}\left(R(\beta_0) - R(\beta^\star)\right).
\end{aligned}
$$

Further, notice that

$$\mathbb{E}_S[\ell(x,y,\beta)] \geq \mathbb{E}_S[\ell(x,y,\beta^\star)].$$

We then finish the proofs.

$\square$

In the following, we fix a $\beta_0 \in \mathcal{B}_S$ such that $R(\beta_0) - R(\beta^\star) \geq \Delta_2$. By the definition of $\Delta_1$ and $\Delta_2$, as long as $n \geq N_3'$, we have

$$\Delta_1 \leq \frac{R(\beta_0) - R(\beta^\star)}{4LB_S},$$

which implies

$$\mathcal{B}(\beta_0, \Delta_1) \subset \mathcal{B}\left(\beta_0, \frac{R(\beta_0) - R(\beta^\star)}{4LB_S}\right).$$

Thus, by Proposition C.12, for all $\beta \in \mathcal{B}(\beta_0, \Delta_1)$, we have

$$
\begin{aligned}
\hat{L}(\beta) &= \ell_n(\beta) + \lambda R(\beta) \\
&= \mathbb{E}_S[\ell(x,y,\beta)] + \lambda R(\beta) + \ell_n(\beta) - \mathbb{E}_S[\ell(x,y,\beta)] \\
&> \mathbb{E}_S[\ell(x,y,\beta^\star)] + \lambda R(\beta^\star) + \frac{\lambda}{2}\left(R(\beta_0) - R(\beta^\star)\right) + \ell_n(\beta) - \mathbb{E}_S[\ell(x,y,\beta)] \\
&= \ell_n(\beta^\star) + \lambda R(\beta^\star) + \frac{\lambda}{2}\left(R(\beta_0) - R(\beta^\star)\right) \\
&\quad + (\ell_n(\beta) - \mathbb{E}_S[\ell(x,y,\beta)]) - (\ell_n(\beta^\star) - \mathbb{E}_S[\ell(x,y,\beta^\star)]) \\
&\geq \hat{L}(\beta^\star) + \frac{\lambda}{2}\left(R(\beta_0) - R(\beta^\star)\right) - |(\ell_n(\beta) - \mathbb{E}_S[\ell(x,y,\beta)]) - (\ell_n(\beta^\star) - \mathbb{E}_S[\ell(x,y,\beta^\star)])|.
\end{aligned}
$$
(32)

**Case 1:** $R(\beta_0) - R(\beta^\star) \geq \Delta_{\max} n^{-\frac{1}{3\tau}}$

By Proposition C.4 and (32), we obtain

$$\hat{L}(\beta) > \hat{L}(\beta^\star) + \frac{\lambda}{2}\Delta_{\max}n^{-\frac{1}{3\tau}} - 2B_0\sqrt{\frac{\log n}{n}}.$$

By the choice of $\lambda$, we conclude that

$$\hat{L}(\beta) > \hat{L}(\beta^\star).$$

**Case 2:** $\Delta_2 \leq R(\beta_0) - R(\beta^\star) \leq \Delta_{\max} n^{-\frac{1}{3\tau}}$

Suppose that $R(\beta_0) - R(\beta^\star) = n^{-\epsilon}$ for some $\epsilon$. By Assumption C.3, we have

$$\|\beta_0 - \beta^\star\|_2 \leq n^{-\epsilon\tau}.$$

As a result, for all $\beta \in \mathcal{B}(\beta_0, \Delta_1)$, we have

$$
\begin{aligned}
\|\beta - \beta^\star\|_2 &\leq \|\beta - \beta_0\|_2 + \|\beta_0 - \beta^\star\|_2 \\
&\leq \Delta_1 + n^{-\epsilon\tau} \\
&= c_1 n^{-\delta} (\log n)^{\frac{1}{4}} + n^{-\epsilon\tau}.
\end{aligned}
\tag{33}
$$

By Proposition C.4 and (32), we have

$$
\hat{L}(\beta) > \hat{L}(\beta^\star) + \frac{\lambda}{2} n^{-\epsilon} - \left( C(n, I_d) \|\beta - \beta^\star\|_2 + B_2 \sqrt{\frac{\log n}{n}} \|\beta - \beta^\star\|_2^2 + B_3 \|\beta - \beta^\star\|_2^3 \right).
\tag{34}
$$

Here

$$
\begin{aligned}
C(n, I_d) &= c\sqrt{\frac{\mathsf{Tr}(\mathcal{I}_S) \log n}{n}} + B_1 \log^\gamma \left( \frac{B_1}{\sqrt{\mathsf{Tr}(\mathcal{I}_S)}} \right) \cdot \frac{\log n}{n} \\
&\leq c\sqrt{\frac{\mathsf{Tr}(\mathcal{I}_S) \log n}{n}},
\end{aligned}
$$

where the inequality holds as long as $n \geq cN_3'$. Combining (33) and (34), we have

$$
\begin{aligned}
\hat{L}(\beta) >\ & \hat{L}(\beta^\star) + \frac{\lambda}{2} n^{-\epsilon} - c\sqrt{\frac{\mathsf{Tr}(\mathcal{I}_S) \log n}{n}} \left( c_1 n^{-\delta} (\log n)^{\frac{1}{4}} + n^{-\epsilon\tau} \right) \\
& - B_2 \sqrt{\frac{\log n}{n}} \left( c_1 n^{-\delta} (\log n)^{\frac{1}{4}} + n^{-\epsilon\tau} \right)^2 - B_3 \left( c_1 n^{-\delta} (\log n)^{\frac{1}{4}} + n^{-\epsilon\tau} \right)^3 \\
=\ & \hat{L}(\beta^\star) + \frac{c_\lambda}{2} n^{-2\delta-\epsilon} \sqrt{\log n} - c\sqrt{\frac{\mathsf{Tr}(\mathcal{I}_S) \log n}{n}} \left( c_1 n^{-\delta} (\log n)^{\frac{1}{4}} + n^{-\epsilon\tau} \right) \\
& - B_2 \sqrt{\frac{\log n}{n}} \left( c_1 n^{-\delta} (\log n)^{\frac{1}{4}} + n^{-\epsilon\tau} \right)^2 - B_3 \left( c_1 n^{-\delta} (\log n)^{\frac{1}{4}} + n^{-\epsilon\tau} \right)^3 \\
\geq\ & \hat{L}(\beta^\star) + \frac{c_\lambda}{2} n^{-2\delta-\epsilon} \sqrt{\log n} - c\sqrt{\frac{\mathsf{Tr}(\mathcal{I}_S) \log n}{n}} \left( c_1 n^{-\delta} (\log n)^{\frac{1}{4}} + n^{-\epsilon\tau} \right) \\
& - 2B_2 \sqrt{\frac{\log n}{n}} \left( c_1^2 n^{-2\delta} (\log n)^{\frac{1}{2}} + n^{-2\epsilon\tau} \right) - 4B_3 \left( c_1^3 n^{-3\delta} (\log n)^{\frac{3}{4}} + n^{-3\epsilon\tau} \right) \\
=\ & \hat{L}(\beta^\star) + \frac{1}{2} n^{-2\delta-\epsilon} \sqrt{\log n} \bigg( c_\lambda - c\sqrt{\mathsf{Tr}(\mathcal{I}_S)} \left( c_1 n^{\delta+\epsilon-\frac{1}{2}} (\log n)^{\frac{1}{4}} + n^{2\delta-(\tau-1)\epsilon-\frac{1}{2}} \right) \\
& - 4B_2 \left( c_1^2 n^{\epsilon-\frac{1}{2}} (\log n)^{\frac{1}{2}} + n^{2\delta-(2\tau-1)\epsilon-\frac{1}{2}} \right) - 8\frac{B_3}{\sqrt{\log n}} \left( c_1^3 n^{\epsilon-\delta} (\log n)^{\frac{3}{4}} + n^{2\delta-(3\tau-1)\epsilon} \right) \bigg) \\
\geq\ & \hat{L}(\beta^\star) + \frac{1}{2} n^{-2\delta-\epsilon} \sqrt{\log n} \bigg( c_\lambda - c\sqrt{\mathsf{Tr}(\mathcal{I}_S)} \left( \frac{c_1}{\Delta_2} n^{\delta-\frac{1}{2}} (\log n)^{\frac{1}{4}} + \Delta_{\max}^{\tau-1} \cdot n^{2\delta-\frac{5}{6}+\frac{1}{3\tau}} \right) \\
& - 4B_2 \left( \frac{c_1^2}{\Delta_2} n^{-\frac{1}{2}} (\log n)^{\frac{1}{2}} + \Delta_{\max}^{2\tau-1} n^{2\delta-\frac{7}{6}+\frac{1}{3\tau}} \right) \\
& - 8\frac{B_3}{\sqrt{\log n}} \left( \frac{c_1^3}{\Delta_2} n^{-\delta} (\log n)^{\frac{3}{4}} + \Delta_{\max}^{3\tau-1} n^{2\delta-1+\frac{1}{3\tau}} \right) \bigg) \\
=\ & \hat{L}(\beta^\star) + \frac{1}{2} n^{-2\delta-\epsilon} \sqrt{\log n} \bigg( c_\lambda - c\sqrt{\mathsf{Tr}(\mathcal{I}_S)} \left( c_1 n^{2\delta-\frac{7}{12}} + \Delta_{\max}^{\tau-1} \cdot n^{2\delta-\frac{5}{6}+\frac{1}{3\tau}} \right)
\end{aligned}
$$

$$- 4B_2 \left( c_1^2 n^{\delta - \frac{7}{12}} (\log n)^{\frac{1}{4}} + \Delta_{\max}^{2\tau - 1} n^{2\delta - \frac{7}{6} + \frac{1}{3\tau}} \right) - 8B_3 \left( c_1^3 n^{-\frac{1}{12}} + \Delta_{\max}^{3\tau - 1} n^{2\delta - 1 + \frac{1}{3\tau}} \right) \right)$$

$$\geq \hat{L}(\beta^\star) + \frac{1}{2} n^{-2\delta - \epsilon} \sqrt{\log n} \left( c_\lambda - c\sqrt{\mathsf{Tr}(\mathcal{I}_S)} \left( c_1 n^{-\frac{1}{12}} + \Delta_{\max}^{\tau - 1} \cdot n^{-\frac{1}{3}} \right) \right.$$

$$\left. - 4B_2 \left( c_1^2 n^{-\frac{1}{4}} + \Delta_{\max}^{2\tau - 1} n^{-\frac{2}{3}} \right) - 8B_3 \left( c_1^3 n^{-\frac{1}{12}} + \Delta_{\max}^{3\tau - 1} n^{-\frac{1}{2}} \right) \right).$$

As a result, as long as $n \geq cN_3'$, we have

$$\hat{L}(\beta) \geq \hat{L}(\beta^\star).$$

Combining Case 1 and Case 2, we finish the proofs.

$\square$

We denote

$$\mathcal{D}_S^0 := \left\{ \beta \in \beta_0 + \mathrm{col}\left( \mathbb{E}_S[\nabla^2 \ell(x, y, \beta_0)] \right) \mid \|\beta - \beta_0\|_2 \leq \Delta_1 \right\}.$$

Recall from (30) that

$$\hat{\beta}_\lambda \in \mathcal{A}_S \subset \bigcup_{\beta_0 \in \mathcal{B}_S} \mathcal{D}_S^0.$$

Combining this with Lemma C.11, we conclude that

$$\hat{\beta}_\lambda \in \bigcup_{\substack{\beta_0 \in \mathcal{B}_S \\ R(\beta_0) - R(\beta^\star) \leq \Delta_2}} \mathcal{D}_S^0.$$

Further, by Assumption C.3, we conclude that

$$\hat{\beta}_\lambda \in \bigcup_{\substack{\beta_0 \in \mathcal{B}_S \\ \|\beta_0 - \beta^\star\|_2 \leq \Delta_2^\tau}} \mathcal{D}_S^0 \equiv \mathcal{D}_S.$$

As a result, we restrict our analysis to the following optimization problem:

$$\min_{\beta \in \mathcal{D}_S} \ell_n(\beta) + \lambda R(\beta). \tag{35}$$

It is worth noting that the strong convexity result stated in Proposition C.7 holds over the region $\mathcal{D}_S$ by our choice of $\Delta_1$.

Note that the optimization problem in (35) is equivalent to the following:

$$\min_{\substack{\beta_0 \in \mathcal{B}_S \\ \|\beta_0 - \beta^\star\|_2 \leq \Delta_2^\tau}} \min_{\beta \in \mathcal{D}_S^0} \ell_n(\beta) + \lambda R(\beta), \tag{36}$$

where

$$\mathcal{D}_S^0 = \left\{ \beta \in \beta_0 + \mathrm{col}\left( \mathbb{E}_S[\nabla^2 \ell(x, y, \beta_0)] \right) \mid \|\beta - \beta_0\|_2 \leq \Delta_1 \right\}.$$

Thus, we begin by fixing some $\beta_0 \in \mathcal{B}_S$ and analyzing the following local optimization problem:

$$\min_{\beta \in \mathcal{D}_S^0} \ell_n(\beta) + \lambda R(\beta). \tag{37}$$

We emphasize two key properties:

1. $\beta_0$ is the minimizer of the population loss over $\mathcal{D}_S^0$, i.e., $\beta_0 = \arg\min_{\beta \in \mathcal{D}_S^0} \mathbb{E}_S[\ell(x, y, \beta)]$;

2. The function $\mathbb{E}_S[\ell(x, y, \beta)]$ is $\frac{\alpha}{2}$-strongly convex over $\mathcal{D}_S^0$.

In the sequel, we denote

$$\hat{\beta}_\lambda^0 := \arg\min_{\beta \in \mathcal{D}_S^0} \ell_n(\beta) + \lambda R(\beta).$$

Recall: $\mathcal{I}_S := \mathcal{I}_S(\beta^\star)$, $\mathcal{I}_T := \mathcal{I}_T(\beta^\star)$, $\alpha_1 := B_1 \|\mathcal{I}_S^\dagger\|_2^{1/2}$, $\alpha_2 := B_2 \|\mathcal{I}_S^\dagger\|_2$, $\alpha_3 := B_3 \|\mathcal{I}_S^\dagger\|_2^{3/2}$,

$$\kappa := \frac{\mathsf{Tr}(\mathcal{I}_T \mathcal{I}_S^\dagger)}{\|\mathcal{I}_T^{\frac{1}{2}} \mathcal{I}_S^\dagger \mathcal{I}_T^{\frac{1}{2}}\|_2}, \quad \tilde{\kappa} := \frac{\mathsf{Tr}(\mathcal{I}_S^\dagger)}{\|\mathcal{I}_S^\dagger\|_2}.$$

Following the same reasoning as in (19) and (21) from the proof of Theorem 4.1, we obtain the following result:

**Lemma C.13.** *Suppose $\tau \geq 9$ and $n \geq c \cdot N_4'$, where*

$$
\begin{aligned}
N_4' := \max\Bigg\{ & \left(B_2 \|\mathcal{I}_S\|_2^{-1}\right)^4, \left(B_3 \|\mathcal{I}_S\|_2^{-1}\right)^4, \|\mathcal{I}_S\|_2^{\frac{12}{3\tau-16}}, B_3^{\frac{24}{3\tau-16}}, \left(\alpha^{-1} B_2\right)^3, \left(\alpha^{-3} B_0 B_3^2\right)^3, \\
& \left(c_\lambda \alpha^{-2} L B_S B_3\right)^{\frac{12\tau}{5\tau-4}}, \left(\tilde{\kappa}^{-1} \|\mathcal{I}_S^\dagger\|_2 \|\mathcal{I}_S\|_2^2\right)^{\frac{12}{3\tau-16}}, \tilde{\kappa}^{-1} B_1^2 \|\mathcal{I}_S^\dagger\|_2 \log^{2\gamma}(\tilde{\kappa}^{-1/2} \alpha_1), \\
& (c_\lambda L)^{\frac{24\tau}{\tau-8}}, B_3^{24}, (\tilde{\kappa} \|\mathcal{I}_S^\dagger\|_2)^{12}, \left(c_\lambda \left\|\mathcal{I}_S^\dagger \nabla R(\beta^\star)\right\|_2\right)^{\frac{24\tau}{\tau-8}}, (B_2 \tilde{\kappa} \|\mathcal{I}_S^\dagger\|_2)^3, \\
& \left(B_2 c_\lambda^2 \left\|\mathcal{I}_S^\dagger \nabla R(\beta^\star)\right\|_2^2\right)^{\frac{24\tau}{3\tau-16}}, \left(L c_\lambda \tilde{\kappa} \|\mathcal{I}_S^\dagger\|_2\right)^{\frac{24\tau}{\tau-8}}, \left(L c_\lambda^3 \left\|\mathcal{I}_S^\dagger \nabla R(\beta^\star)\right\|_2^2\right)^{\frac{8\tau}{\tau-8}}, \\
& \left(\alpha^{-1} B_3\right)^{\frac{24}{3\tau-4}}, (B_3 \tilde{\kappa}^{-1} \alpha^{-1} \|\mathcal{I}_S^\dagger\|_2^{-1})^8, \left(B_3 \kappa^{-1} \|\mathcal{I}_T\|_2 \|\mathcal{I}_S^\dagger\|_2 \|\mathcal{I}_T^{\frac{1}{2}} \mathcal{I}_S^\dagger \mathcal{I}_T^{\frac{1}{2}}\|_2^{-1}\right)^8 \\
& \left(c_\lambda^2 \left\|\mathcal{I}_S^\dagger \nabla R(\beta^\star)\right\|_2^2\right)^{\frac{12\tau}{\tau-8}}, \left(\|\mathcal{I}_S\|_2^{-1} B_3\right)^{\frac{24}{3\tau-4}}, B_1^2 \|\mathcal{I}_S^\dagger\|_2 \kappa^{-1} \log^{2\gamma}(\kappa^{-1/2} \alpha_1) \Bigg\}.
\end{aligned}
\tag{38}
$$

*Then the following bounds hold:*

$$\|\hat{\beta}_\lambda^0 - \beta_0\|_2 \leq 2\left(c\sqrt{\frac{\mathsf{Tr}(\mathcal{I}_S^\dagger)\log n}{n}} + \lambda \left\|\mathcal{I}_S^\dagger \nabla R(\beta^\star)\right\|_2\right)$$

$$\|\mathcal{I}_T^{\frac{1}{2}}(\hat{\beta}_\lambda^0 - \beta_0)\|_2^2 \leq c\left(\frac{\mathsf{Tr}(\mathcal{I}_S^\dagger \mathcal{I}_T)\log n}{n} + \lambda^2 \|\mathcal{I}_T^{\frac{1}{2}} \mathcal{I}_S^\dagger \nabla R(\beta^\star)\|_2^2\right).$$

*Proof.* The proof follows a similar procedure as in the derivations of (19) and (21). For completeness, we defer the detailed argument to Section C.2.1. $\square$

With Lemma C.13 in place, we are now ready to complete the proof of Theorem 4.2.

*Proof of Theorem 4.2.* By Lemma C.13 and (36), we have

$$
\begin{aligned}
\|\hat{\beta}_\lambda^0 - \beta^\star\|_2 &\leq \|\hat{\beta}_\lambda^0 - \beta_0\|_2 + \|\beta_0 - \beta^\star\|_2 \\
&\leq 2\left(c\sqrt{\frac{\mathsf{Tr}(\mathcal{I}_S^\dagger)\log n}{n}} + \lambda \left\|\mathcal{I}_S^\dagger \nabla R(\beta^\star)\right\|_2\right) + \Delta_2^\tau,
\end{aligned}
$$

and

$$
\begin{aligned}
\|\mathcal{I}_T^{\frac{1}{2}}(\hat{\beta}_\lambda^0 - \beta^\star)\|_2^2 &\leq 2\|\mathcal{I}_T^{\frac{1}{2}}(\hat{\beta}_\lambda^0 - \beta_0)\|_2^2 + 2\|\mathcal{I}_T^{\frac{1}{2}}(\beta_0 - \beta^\star)\|_2^2 \\
&\leq 2\|\mathcal{I}_T^{\frac{1}{2}}(\hat{\beta}_\lambda^0 - \beta_0)\|_2^2 + 2\|\mathcal{I}_T\|_2 \|\beta_0 - \beta^\star\|_2^2 \\
&\leq c\left(\frac{\mathsf{Tr}(\mathcal{I}_S^\dagger \mathcal{I}_T)\log n}{n} + \lambda^2 \|\mathcal{I}_T^{\frac{1}{2}} \mathcal{I}_S^\dagger \nabla R(\beta^\star)\|_2^2 + \|\mathcal{I}_T\|_2 \Delta_2^{2\tau}\right).
\end{aligned}
$$

Then, by Taylor's expansion, we have

$$\mathcal{E}(\hat{\beta}_\lambda^0) = \mathbb{E}_T \left[ \ell(x, y, \hat{\beta}_\lambda^0) - \ell(x, y, \beta^\star) \right]$$

$$\leq \mathbb{E}_T[\nabla\ell(x, y, \beta^\star)]^T(\hat{\beta}_\lambda^0 - \beta^\star) + \frac{1}{2}(\hat{\beta}_\lambda^0 - \beta^\star)^T \mathcal{I}_T(\hat{\beta}_\lambda^0 - \beta^\star) + \frac{B_3}{6}\|\hat{\beta}_\lambda^0 - \beta^\star\|_2^3$$

$$= \frac{1}{2}(\hat{\beta}_\lambda^0 - \beta^\star)^T \mathcal{I}_T(\hat{\beta}_\lambda^0 - \beta^\star) + \frac{B_3}{6}\|\hat{\beta}_\lambda^0 - \beta^\star\|_2^3$$

$$\leq c \left( \frac{\mathsf{Tr}(\mathcal{I}_S^\dagger \mathcal{I}_T)\log n}{n} + \lambda^2\|\mathcal{I}_T^{\frac{1}{2}}\mathcal{I}_S^\dagger \nabla R(\beta^\star)\|_2^2 + \|\mathcal{I}_T\|_2 \Delta_2^{2\tau} \right)$$

$$+ cB_3 \left( \left( \sqrt{\frac{\mathsf{Tr}(\mathcal{I}_S^\dagger)\log n}{n}} \right)^3 + \lambda^3 \left\|\mathcal{I}_S^\dagger \nabla R(\beta^\star)\right\|_2^3 + \Delta_2^{3\tau} \right)$$

$$\leq c \left( \frac{\mathsf{Tr}(\mathcal{I}_S^\dagger \mathcal{I}_T)\log n}{n} + \lambda^2\|\mathcal{I}_T^{\frac{1}{2}}\mathcal{I}_S^\dagger \nabla R(\beta^\star)\|_2^2 \right) \tag{39}$$

Here the last inequality holds as long as $n \geq N_5'$, where

$$N_5' = \max\Bigg\{ B_3^4(\mathsf{Tr}(\mathcal{I}_S^\dagger))^6(\mathsf{Tr}(\mathcal{I}_S^\dagger \mathcal{I}_T))^{-4},$$

$$\left( B_3 c_\lambda \left\|\mathcal{I}_S^\dagger \nabla R(\beta^\star)\right\|_2^3 \|\mathcal{I}_T^{\frac{1}{2}}\mathcal{I}_S^\dagger \nabla R(\beta^\star)\|_2^{-2} \right)^{\frac{12\tau}{5\tau-4}},$$

$$\left( B_3\|\mathcal{I}_T\|_2^{-1} \right)^{\frac{24}{3\tau-4}}, \left( \|\mathcal{I}_T\|_2(\mathsf{Tr}(\mathcal{I}_S^\dagger \mathcal{I}_T))^{-1} \right)^{\frac{12}{3\tau-16}} \Bigg\}$$

$$= \max\Bigg\{ B_3^4\kappa^{-4}\tilde{\kappa}^6\|\mathcal{I}_S^\dagger\|_2^6\|\mathcal{I}_T^{\frac{1}{2}}\mathcal{I}_S^\dagger \mathcal{I}_T^{\frac{1}{2}}\|_2^{-4},$$

$$\left( B_3 c_\lambda \left\|\mathcal{I}_S^\dagger \nabla R(\beta^\star)\right\|_2^3 \|\mathcal{I}_T^{\frac{1}{2}}\mathcal{I}_S^\dagger \nabla R(\beta^\star)\|_2^{-2} \right)^{\frac{12\tau}{5\tau-4}},$$

$$\left( B_3\|\mathcal{I}_T\|_2^{-1} \right)^{\frac{24}{3\tau-4}}, \left( \kappa^{-1}\|\mathcal{I}_T\|_2\|\mathcal{I}_T^{\frac{1}{2}}\mathcal{I}_S^\dagger \mathcal{I}_T^{\frac{1}{2}}\|_2^{-1} \right)^{\frac{12}{3\tau-16}} \Bigg\}. \tag{40}$$

Since (39) holds for any fixed $\beta_0$ under consideration, we conclude that

$$\mathcal{E}(\hat{\beta}_\lambda) \leq c \left( \frac{\mathsf{Tr}(\mathcal{I}_S^\dagger \mathcal{I}_T)\log n}{n} + \lambda^2\|\mathcal{I}_T^{\frac{1}{2}}\mathcal{I}_S^\dagger \nabla R(\beta^\star)\|_2^2 \right)$$

$$= c \left( \frac{\mathsf{Tr}(\mathcal{I}_S^\dagger \mathcal{I}_T)\log n}{n} + \frac{c_\lambda^2\|\mathcal{I}_T^{\frac{1}{2}}\mathcal{I}_S^\dagger \nabla R(\beta^\star)\|_2^2}{n^{1-\frac{2}{3\tau}}} \right).$$

In the end, we summarize the threshold of $n$. We require $n \geq cN'$, where

$$N' := \max\{N_1', N_2', N_3', N_4', N_5'\}. \tag{41}$$

Here $N_1', N_2', N_3', N_4', N_5'$ are defined in (28), (29), (31), (38), (40), respectively.

$\square$

### C.2.1 PROOF OF LEMMA C.13

In this section, we present the proof of Lemma C.13. Recall the definition of $N_4'$ in Lemma C.13, we have

$$N_4' \geq \max\left\{ \left( \frac{B_2 + B_3}{\|\mathcal{I}_S\|_2} \right)^4, \left( \frac{\|\mathcal{I}_S\|_2}{\sqrt{\mathsf{Tr}(\mathcal{I}_S)}} \right)^{\frac{24}{3\tau-16}}, \left( \frac{B_3}{B_2} \right)^{\frac{24}{3\tau-16}}, \left( \frac{B_2}{\alpha} \right)^3, \left( \frac{B_0 B_3^2}{\alpha^3} \right)^3, \right.$$

$$\left(\frac{c_\lambda LB_S B_3}{\alpha^2}\right)^{\frac{12\tau}{5\tau-4}}, \left(\frac{\|\mathcal{I}_S^\dagger\|_2 \|\mathcal{I}_S\|_2}{\sqrt{\mathsf{Tr}(\mathcal{I}_S^\dagger)}}\right)^{\frac{24}{3\tau-16}}, \tilde{\kappa}^{-1} B_1^2 \|\mathcal{I}_S^\dagger\|_2 \log^{2\gamma}(\tilde{\kappa}^{-1/2}\alpha_1),$$

$$(c_\lambda LB_3^{-1})^{\frac{24\tau}{\tau-8}}, \left(\frac{B_3}{B_2}\right)^{24}, \mathsf{Tr}(\mathcal{I}_S^\dagger)^{12}, \left(c_\lambda \left\|\mathcal{I}_S^\dagger \nabla R(\beta^\star)\right\|_2\right)^{\frac{24\tau}{\tau-8}}, B_2^3 B_3^{-3} \mathsf{Tr}(\mathcal{I}_S^\dagger)^3,$$

$$\left(B_2 B_3^{-1} c_\lambda^2 \left\|\mathcal{I}_S^\dagger \nabla R(\beta^\star)\right\|_2^2\right)^{\frac{24\tau}{3\tau-16}}, \left(B_3^{-1} L c_\lambda \mathsf{Tr}(\mathcal{I}_S^\dagger)\right)^{\frac{24\tau}{3\tau-8}}, \left(B_3^{-1} L c_\lambda^3 \left\|\mathcal{I}_S^\dagger \nabla R(\beta^\star)\right\|_2^2\right)^{\frac{8\tau}{\tau-8}},$$

$$\left(\frac{B_3}{\alpha}\right)^{\frac{24}{3\tau-4}}, (B_3 \alpha^{-1} \mathsf{Tr}(\mathcal{I}_S^\dagger)^{-1})^8, \left(B_3 \|\mathcal{I}_T\|_2 \|\mathcal{I}_S^\dagger\|_2 \mathsf{Tr}(\mathcal{I}_S^\dagger \mathcal{I}_T)^{-1}\right)^8,$$

$$\left(c_\lambda^2 \left\|\mathcal{I}_S^\dagger \nabla R(\beta^\star)\right\|_2^2\right)^{\frac{12\tau}{\tau-8}}, \left(\|\mathcal{I}_S\|_2^{-1} B_3\right)^{\frac{24}{3\tau-4}}, B_1^2 \|\mathcal{I}_S^\dagger\|_2 \kappa^{-1} \log^{2\gamma}(\kappa^{-1/2}\alpha_1)\right\}.$$

The proof of Lemma C.13 follows the same reasoning used to derive inequalities (19) and (21) in the proof of Theorem 4.1. Recall that establishing those bounds required applying concentration inequalities at the ground truth parameter $\beta^\star$. In the current setting, we apply the same concentration tools at $\beta_0$ instead.

Note that

$$\|\beta_0 - \beta^\star\|_2 \le \Delta_2^\tau = n^{-\frac{\tau-1}{6}}(\log n)^{\frac{\tau}{4}} \le n^{-\frac{\tau}{8}+\frac{1}{6}},$$

which implies that $\beta_0$ lies sufficiently close to $\beta^\star$ if $\tau$ is sufficiently large. This proximity is small enough to ensure that both $\nabla \ell_n(\beta_0)$ and $\nabla^2 \ell_n(\beta_0)$ remain close to their expectations at $\beta^\star$—namely, $\mathbb{E}[\nabla \ell_n(\beta^\star)]$ and $\mathbb{E}[\nabla^2 \ell_n(\beta^\star)]$, respectively.

We formalize this intuition in the following proposition.

**Proposition C.14.** *Under Assumption A.1 and A.2, we have for any fixed matrix $A \in \mathbb{R}^{d \times d}$ and any $n \ge \max\{(B_2 + B_3)^4 \|\mathcal{I}_S\|_2^{-4}, N\}$, the following inequalities hold simultaneously with probability at least $1 - n^{-20}$:*

$$\|A(\nabla \ell_n(\beta_0) - \mathbb{E}[\nabla \ell_n(\beta^\star)])\|_2 \le c\sqrt{\frac{V \log n}{n}} + B_1 \|A\|_2 \log^\gamma \left(\frac{B_1 \|A\|_2}{\sqrt{V}}\right) \frac{\log n}{n} + c\|A\|_2 \|\mathcal{I}_S\|_2 \Delta_2^\tau,$$

$$\max\left\{\left\|\nabla^2 \ell_n(\beta_0) - \mathbb{E}[\nabla^2 \ell_n(\beta^\star)]\right\|_2, \left\|\nabla^2 \ell_n(\beta_0) - \mathbb{E}[\nabla^2 \ell_n(\beta_0)]\right\|_2\right\} \le B_2 \sqrt{\frac{\log n}{n}} + 2B_3 \Delta_2^\tau,$$

*where $V = n \cdot \mathbb{E}\|A(\nabla \ell_n(\beta^\star) - \mathbb{E}[\nabla \ell_n(\beta^\star)])\|_2^2$ denotes the variance term.*

*Proof of Proposition C.14.* Note that

$$\|A(\nabla \ell_n(\beta_0) - \nabla \ell_n(\beta^\star))\|_2$$
$$\le \|A\|_2 \|\nabla \ell_n(\beta_0) - \nabla \ell_n(\beta^\star)\|_2$$
$$\le \|A\|_2 \left(\left\|\nabla^2 \ell_n(\beta^\star)(\beta_0 - \beta^\star)\right\|_2 + B_3 \|\beta_0 - \beta^\star\|_2^2\right)$$
$$\le \|A\|_2 \left(\left\|\nabla^2 \ell_n(\beta^\star) - \mathcal{I}_S\right\|_2 \|\beta_0 - \beta^\star\|_2 + \|\mathcal{I}_S\|_2 \|\beta_0 - \beta^\star\|_2 + B_3 \|\beta_0 - \beta^\star\|_2^2\right)$$
$$\le \|A\|_2 \left(\|\mathcal{I}_S\|_2 + B_2 \sqrt{\frac{\log n}{n}} + B_3 \Delta_2^\tau\right) \Delta_2^\tau$$
$$\le \|A\|_2 \left(\|\mathcal{I}_S\|_2 + B_2 n^{-1/4} + B_3 n^{-1/4}\right) \Delta_2^\tau$$
$$\le \|A\|_2 \left(\|\mathcal{I}_S\|_2 + (B_2 + B_3) n^{-1/4}\right) \Delta_2^\tau$$
$$\le c\|A\|_2 \|\mathcal{I}_S\|_2 \Delta_2^\tau,$$

where the last inequality holds as long as $n \ge (B_2 + B_3)^4 \|\mathcal{I}_S\|_2^{-4}$. Thus, by Assumption A.1, we have

$$\|A(\nabla \ell_n(\beta_0) - \mathbb{E}[\nabla \ell_n(\beta^\star)])\|_2$$

$$\leq \|A \left( \nabla \ell_n(\beta^\star) - \mathbb{E}[\nabla \ell_n(\beta^\star)]\right)\|_2 + \|A \left(\nabla \ell_n(\beta_0) - \nabla \ell_n(\beta^\star)\right)\|_2$$

$$\leq c\sqrt{\frac{V \log n}{n}} + B_1 \|A\|_2 \log^\gamma \left(\frac{B_1 \|A\|_2}{\sqrt{V}}\right) \frac{\log n}{n} + c\|A\|_2 \|\mathcal{I}_S\|_2 \Delta_2^\tau.$$

By Assumption A.2, we have

$$\left\|\nabla^2 \ell_n(\beta_0) - \nabla^2 \ell_n(\beta^\star)\right\|_2 \leq B_3 \|\beta_0 - \beta^\star\|_2 \leq B_3 \Delta_2^\tau,$$

$$\left\|\mathbb{E}[\nabla^2 \ell_n(\beta_0)] - \mathbb{E}[\nabla^2 \ell_n(\beta^\star)]\right\|_2 \leq B_3 \|\beta_0 - \beta^\star\|_2 \leq B_3 \Delta_2^\tau.$$

Consequently, by Assumption A.1, we have

$$\left\|\nabla^2 \ell_n(\beta_0) - \mathbb{E}[\nabla^2 \ell_n(\beta^\star)]\right\|_2$$

$$\leq \left\|\nabla^2 \ell_n(\beta_0) - \nabla^2 \ell_n(\beta^\star)\right\|_2 + \left\|\nabla^2 \ell_n(\beta^\star) - \mathbb{E}[\nabla^2 \ell_n(\beta^\star)]\right\|_2$$

$$\leq B_2 \sqrt{\frac{\log n}{n}} + B_3 \Delta_2^\tau,$$

and

$$\left\|\nabla^2 \ell_n(\beta_0) - \mathbb{E}[\nabla^2 \ell_n(\beta_0)]\right\|_2$$

$$\leq \left\|\nabla^2 \ell_n(\beta_0) - \nabla^2 \ell_n(\beta^\star)\right\|_2 + \left\|\mathbb{E}[\nabla^2 \ell_n(\beta_0)] - \mathbb{E}[\nabla^2 \ell_n(\beta^\star)]\right\|_2$$

$$\leq B_2 \sqrt{\frac{\log n}{n}} + 2B_3 \Delta_2^\tau.$$

We then finish the proofs.

$\square$

With Proposition C.14 in place, we are now ready to establish Lemma C.13.

*Proof of Lemma C.13.* Recall the notations: $\mathcal{I}_S := \mathcal{I}_S(\beta^\star)$, $\mathcal{I}_T := \mathcal{I}_T(\beta^\star)$, $\alpha_1 := B_1 \|\mathcal{I}_S^\dagger\|_2^{1/2}$, $\alpha_2 := B_2 \|\mathcal{I}_S^\dagger\|_2$, $\alpha_3 := B_3 \|\mathcal{I}_S^\dagger\|_2^{3/2}$,

$$\kappa := \frac{\mathsf{Tr}(\mathcal{I}_T \mathcal{I}_S^\dagger)}{\|\mathcal{I}_T^{\frac{1}{2}} \mathcal{I}_S^\dagger \mathcal{I}_T^{\frac{1}{2}}\|_2}, \quad \tilde{\kappa} := \frac{\mathsf{Tr}(\mathcal{I}_S^\dagger)}{\|\mathcal{I}_S^\dagger\|_2}.$$

We further denote $\mathcal{I}_S^0 = \mathcal{I}_S(\beta_0)$ and $\mathcal{I}_T^0 = \mathcal{I}_T(\beta_0)$.

We start by proving a useful proposition.

**Proposition C.15.** *Suppose that $n \geq N_4'$. Then, for all $\beta$, it holds that*

$$|(\mathbb{E}_S[\ell(x, y, \beta)] - \ell_n(\beta)) - (\mathbb{E}_S[\ell(x, y, \beta_0)] - \ell_n(\beta_0))|$$

$$\leq \min\left\{2B_0 \sqrt{\frac{\log n}{n}}, C(n, I_d) \|\beta - \beta_0\|_2 + B_2 \sqrt{\frac{\log n}{n}} \|\beta - \beta_0\|_2^2 + B_3 \|\beta - \beta_0\|_2^3\right\}.$$

*Here*

$$C(n, I_d) = c\sqrt{\frac{\mathsf{Tr}(\mathcal{I}_S) \log n}{n}} + B_1 \log^\gamma \left(\frac{B_1}{\sqrt{\mathsf{Tr}(\mathcal{I}_S)}}\right) \cdot \frac{\log n}{n}.$$

*Proof of Proposition C.15.* Note that by Proposition C.14, for all $\beta$:

$$|(\mathbb{E}_S[\ell(x, y, \beta)] - \ell_n(\beta)) - (\mathbb{E}_S[\ell(x, y, \beta_0)] - \ell_n(\beta_0))|$$

$$\leq \left|(\beta - \beta_0)^\top \nabla \left(\mathbb{E}_S[\ell(x, y, \beta_0)] - \ell_n(\beta_0)\right)\right|$$

$$+ \frac{1}{2} \left|(\beta - \beta_0)^\top \nabla^2 \left(\mathbb{E}_S[\ell(x, y, \beta_0)] - \ell_n(\beta_0)\right)(\beta - \beta_0)\right| + \frac{B_3}{3} \|\beta - \beta_0\|_2^3$$

$$\leq (C(n, I_d) + c\|\mathcal{I}_S\|_2 \Delta_2^\tau) \|\beta - \beta_0\|_2 + \left(\frac{B_2}{2}\sqrt{\frac{\log n}{n}} + B_3 \Delta_2^\tau\right) \|\beta - \beta_0\|_2^2 + B_3 \|\beta - \beta_0\|_2^3$$

$$\leq C(n, I_d) \|\beta - \beta_0\|_2 + B_2 \sqrt{\frac{\log n}{n}} \|\beta - \beta_0\|_2^2 + B_3 \|\beta - \beta_0\|_2^3,$$

where the last inequality holds as long as $n \geq cN_4'$. Moreover, we have

$$|(\mathbb{E}_S[\ell(x, y, \beta)] - \ell_n(\beta)) - (\mathbb{E}_S[\ell(x, y, \beta_0)] - \ell_n(\beta_0))|$$
$$\leq |\mathbb{E}_S[\ell(x, y, \beta)] - \ell_n(\beta)| + |\mathbb{E}_S[\ell(x, y, \beta_0)] - \ell_n(\beta_0)|$$
$$\leq 2B_0 \sqrt{\frac{\log n}{n}}.$$

Thus, we finish the proofs. $\qquad\square$

We now proceed to establish the following lemma.

**Lemma C.16.** *Suppose $n \geq N_4'$. Then, for all $\beta \in \mathcal{D}_S^0 \setminus \mathcal{B}(\beta_0, D'')$, we have $\hat{L}(\beta) > \hat{L}(\beta_0)$. Here*

$$D'' := \frac{8}{\alpha} \left(C(n, I_d) + \lambda L \|\beta_0\|_2\right). \tag{42}$$

*Proof of Lemma C.16.* For any $\beta \in \mathcal{D}_S^0$, we have

$$\hat{L}(\beta) = \ell_n(\beta) + \lambda R(\beta)$$
$$= \mathbb{E}_S[\ell(x, y, \beta)] + \ell_n(\beta) - \mathbb{E}_S[\ell(x, y, \beta)] + \lambda R(\beta)$$
$$\geq \mathbb{E}_S[\ell(x, y, \beta_0)] + \frac{\alpha}{4} \|\beta - \beta_0\|_2^2 + \ell_n(\beta) - \mathbb{E}_S[\ell(x, y, \beta)] + \lambda R(\beta)$$
$$= \ell_n(\beta_0) + \lambda R(\beta_0) + \frac{\alpha}{4} \|\beta - \beta_0\|_2^2$$
$$\quad + (\mathbb{E}_S[\ell(x, y, \beta_0)] - \ell_n(\beta_0)) - (\mathbb{E}_S[\ell(x, y, \beta)] - \ell_n(\beta)) + \lambda (R(\beta) - R(\beta_0))$$
$$= \hat{L}(\beta_0) + \frac{\alpha}{4} \|\beta - \beta_0\|_2^2$$
$$\quad + (\mathbb{E}_S[\ell(x, y, \beta_0)] - \ell_n(\beta_0)) - (\mathbb{E}_S[\ell(x, y, \beta)] - \ell_n(\beta)) + \lambda (R(\beta) - R(\beta_0)),$$

where the inequality follows from the strong convexity of $\mathbb{E}_S[\ell(x, y, \beta)]$ within $\mathcal{D}_S^0$. Note that by Assumption A.4, we have

$$R(\beta) - R(\beta_0) \geq \nabla R(\beta_0)^\top (\beta - \beta_0) \geq -\|\nabla R(\beta_0)\|_2 \|\beta - \beta_0\|_2 \geq -L\|\beta_0\|_2 \|\beta - \beta_0\|_2.$$

Thus, we obtain for all $\beta \in \mathcal{D}_S^0$ that

$$\hat{L}(\beta) \geq \hat{L}(\beta_0) + \frac{\alpha}{4} \|\beta - \beta_0\|_2^2$$
$$\quad - |(\mathbb{E}_S[\ell(x, y, \beta_0)] - \ell_n(\beta_0)) - (\mathbb{E}_S[\ell(x, y, \beta)] - \ell_n(\beta))| - \lambda L\|\beta_0\|_2 \|\beta - \beta_0\|_2. \tag{43}$$

By Proposition C.15, we then have

$$\hat{L}(\beta) \geq \hat{L}(\beta_0) + \frac{\alpha}{4} \|\beta - \beta_0\|_2^2 - 2B_0 \sqrt{\frac{\log n}{n}} - \lambda L\|\beta_0\|_2 \|\beta - \beta_0\|_2.$$

Thus, as long as

$$\|\beta - \beta_0\|_2 > \frac{2\lambda L\|\beta_0\|_2 + 2\sqrt{\lambda^2 L^2 \|\beta_0\|_2^2 + 2\alpha B_0 \sqrt{\frac{\log n}{n}}}}{\alpha} \equiv D' = \tilde{O}(n^{-1/4}),$$

we have

$$\frac{\alpha}{4} \|\beta - \beta_0\|_2^2 - 2B_0 \sqrt{\frac{\log n}{n}} - \lambda L\|\beta_0\|_2 \|\beta - \beta_0\|_2 > 0$$

and thus $\hat{L}(\beta) > \hat{L}(\beta_0)$. In other words, for all $\beta \in \mathcal{D}_S^0 \setminus \mathcal{B}(\beta_0, D')$, we have $\hat{L}(\beta) > \hat{L}(\beta_0)$.

Next, we deal with $\mathcal{D}_S^0 \cap \mathcal{B}(\beta_0, D')$. Note that for all $\beta \in \mathcal{D}_S^0 \cap \mathcal{B}(\beta_0, D')$, by (43) and Proposition C.15, we have

$$
\hat{L}(\beta) \geq \hat{L}(\beta_0) + \frac{\alpha}{4} \|\beta - \beta_0\|_2^2 - \left( C(n, I_d) \|\beta - \beta_0\|_2 + B_2 \sqrt{\frac{\log n}{n}} \|\beta - \beta_0\|_2^2 + B_3 \|\beta - \beta_0\|_2^3 \right)
$$
$$
- \lambda L \|\beta_0\|_2 \|\beta - \beta_0\|_2
$$
$$
\geq \hat{L}(\beta_0) + \frac{\alpha}{4} \|\beta - \beta_0\|_2^2 - \left( C(n, I_d) \|\beta - \beta_0\|_2 + B_2 \sqrt{\frac{\log n}{n}} \|\beta - \beta_0\|_2^2 + B_3 D' \|\beta - \beta_0\|_2^2 \right)
$$
$$
- \lambda L \|\beta_0\|_2 \|\beta - \beta_0\|_2
$$

As long as $n \geq N_4'$, we have

$$
\frac{\alpha}{4} - B_2 \sqrt{\frac{\log n}{n}} - B_3 D' \geq \frac{\alpha}{8}.
$$

Thus, we have

$$
\hat{L}(\beta) \geq \hat{L}(\beta_0) + \frac{\alpha}{8} \|\beta - \beta_0\|_2^2 - C(n, I_d) \|\beta - \beta_0\|_2 - \lambda L \|\beta_0\|_2 \|\beta - \beta_0\|_2.
$$

Consequently, for $\beta \in \mathcal{D}_S^0 \cap \mathcal{B}(\beta_0, D')$, as long as

$$
\|\beta - \beta_0\|_2 > \frac{8}{\alpha} \left( C(n, I_d) + \lambda L \|\beta_0\|_2 \right) = D'' = \tilde{O}(n^{-\frac{1}{2} + \frac{1}{3\tau}}),
$$

we have $\hat{L}(\beta) > \hat{L}(\beta_0)$. In other words, for all $\beta \in (\mathcal{D}_S^0 \cap \mathcal{B}(\beta_0, D')) \setminus \mathcal{B}(\beta_0, D'')$, we have $\hat{L}(\beta) > \hat{L}(\beta_0)$. Thus, we conclude that for all $\beta \in \mathcal{D}_S^0 \setminus \mathcal{B}(\beta_0, D'')$, we have $\hat{L}(\beta) > \hat{L}(\beta_0)$. $\qquad\square$

We denote $g := \nabla \ell_n(\beta_0) - \mathbb{E}[\nabla \ell_n(\beta^\star)]$. By taking $A = \mathcal{I}_S^\dagger$ in Proposition C.14, we have:

$$
\|\mathcal{I}_S^\dagger g\|_2 \leq c \sqrt{\frac{\mathsf{Tr}(\mathcal{I}_S^\dagger) \log n}{n}} + B_1 \|\mathcal{I}_S^\dagger\|_2 \log^\gamma \left( \frac{B_1 \|\mathcal{I}_S^\dagger\|_2}{\sqrt{\mathsf{Tr}(\mathcal{I}_S^\dagger)}} \right) \frac{\log n}{n} + c \|\mathcal{I}_S^\dagger\|_2 \|\mathcal{I}_S\|_2 \Delta_2^\tau
$$
$$
\leq c \sqrt{\frac{\mathsf{Tr}(\mathcal{I}_S^\dagger) \log n}{n}} + B_1 \|\mathcal{I}_S^\dagger\|_2 \log^\gamma (\tilde{\kappa}^{-1/2} \alpha_1) \frac{\log n}{n}. \tag{44}
$$

Here the last inequality holds as long as $n \geq N_4'$.

Note that by Assumption A.2, we have

$$
\|\mathcal{I}_S^0 - \mathcal{I}_S\|_2 \leq B_3 \|\beta_0 - \beta^\star\|_2 \leq B_3 \Delta_2^\tau, \tag{45}
$$

which implies

$$
\left\| (\mathcal{I}_S^0)^\dagger - (\mathcal{I}_S)^\dagger \right\|_2 \leq c \max \left\{ \|(\mathcal{I}_S^0)^\dagger\|_2^2, \|(\mathcal{I}_S)^\dagger\|_2^2 \right\} \cdot \|\mathcal{I}_S^0 - \mathcal{I}_S\|_2 \leq c \frac{B_3}{\alpha^2} \Delta_2^\tau. \tag{46}
$$

And consequently, we have

$$
\left\| (\mathcal{I}_S^0)^\dagger \mathcal{I}_S^0 - (\mathcal{I}_S)^\dagger \mathcal{I}_S \right\|_2
$$
$$
\leq \left\| (\mathcal{I}_S^0)^\dagger - (\mathcal{I}_S)^\dagger \right\|_2 \|\mathcal{I}_S^0\|_2 + \|(\mathcal{I}_S)^\dagger\|_2 \|\mathcal{I}_S^0 - \mathcal{I}_S\|_2
$$
$$
\leq c \left( \alpha^{-1} \|\mathcal{I}_S\|_2 + 1 \right) \frac{B_3}{\alpha} \Delta_2^\tau. \tag{47}
$$

Here the last inequality holds as long as $n \geq N_4'$.

By Proposition C.14, Assumption A.2 and A.4, for all $\beta - \beta_0 \in \mathsf{col}(\mathcal{I}_S^0)$, we have

$$
\hat{L}(\beta) - \hat{L}(\beta_0)
$$

$$= \ell_n(\beta) - \ell_n(\beta_0) + \lambda\left(R(\beta) - R(\beta_0)\right)$$

$$\leq (\beta - \beta_0)^T \nabla \ell_n(\beta_0) + \frac{1}{2}(\beta - \beta_0)^T \nabla^2 \ell_n(\beta_0)(\beta - \beta_0) + \frac{B_3}{6}\|\beta - \beta_0\|_2^3 + \lambda\left(R(\beta) - R(\beta_0)\right)$$

$$= (\beta - \beta_0)^T g + \frac{1}{2}(\beta - \beta_0)^T \nabla^2 \ell_n(\beta_0)(\beta - \beta_0) + \frac{B_3}{6}\|\beta - \beta_0\|_2^3 + \lambda\left(R(\beta) - R(\beta_0)\right)$$

$$\leq (\beta - \beta_0)^T g + \frac{1}{2}(\beta - \beta_0)^T \mathcal{I}_S^0(\beta - \beta_0) + \left(\frac{B_2}{2}\sqrt{\frac{\log n}{n}} + B_3 \Delta_2^\tau\right)\|\beta - \beta_0\|_2^2 + \frac{B_3}{6}\|\beta - \beta_0\|_2^3$$

$$+ \lambda\left(R(\beta) - R(\beta_0)\right)$$

$$\leq (\beta - \beta_0)^T g + \frac{1}{2}(\beta - \beta_0)^T \mathcal{I}_S^0(\beta - \beta_0) + B_2\sqrt{\frac{\log n}{n}}\|\beta - \beta_0\|_2^2 + \frac{B_3}{6}\|\beta - \beta_0\|_2^3$$

$$+ \lambda\left(\nabla R(\beta_0)^\top(\beta - \beta_0) + \frac{L}{2}\|\beta - \beta_0\|_2^2\right)$$

$$\leq (\beta - \beta_0)^T g + \frac{1}{2}(\beta - \beta_0)^T \mathcal{I}_S^0(\beta - \beta_0) + B_2\sqrt{\frac{\log n}{n}}\|\beta - \beta_0\|_2^2 + \frac{B_3}{6}\|\beta - \beta_0\|_2^3$$

$$+ \lambda\left(\nabla R(\beta^\star)^\top(\beta - \beta_0) + \frac{3L}{2}\|\beta - \beta_0\|_2^2\right)$$

$$= \frac{1}{2}(\Delta_\beta - z)^T \mathcal{I}_S^0(\Delta_\beta - z) - \frac{1}{2}z^T \mathcal{I}_S^0 z + \left(B_2\sqrt{\frac{\log n}{n}} + \frac{3\lambda L}{2}\right)\|\Delta_\beta\|_2^2 + \frac{B_3}{6}\|\Delta_\beta\|_2^3, \qquad (48)$$

where $\Delta_\beta := \beta - \beta_0$ and $z := -(\mathcal{I}_S^0)^\dagger g - \lambda(\mathcal{I}_S^0)^\dagger \nabla R(\beta^\star)$. Notice that $\Delta_{\beta_0+z} = z$, by (44), (46), and (48), we have

$$\hat{L}(\beta_0 + z) - \hat{L}(\beta_0)$$

$$\leq -\frac{1}{2}z^T \mathcal{I}_S^0 z$$

$$+ \left(B_2\sqrt{\frac{\log n}{n}} + \frac{3\lambda L}{2}\right)\left(c\sqrt{\frac{\mathsf{Tr}(\mathcal{I}_S^\dagger)\log n}{n}} + B_1\|\mathcal{I}_S^\dagger\|_2 \log^\gamma(\tilde{\kappa}^{-1/2}\alpha_1)\frac{\log n}{n} + \lambda\left\|\mathcal{I}_S^\dagger \nabla R(\beta^\star)\right\|_2\right)^2$$

$$+ \frac{B_3}{6}\left(c\sqrt{\frac{\mathsf{Tr}(\mathcal{I}_S^\dagger)\log n}{n}} + B_1\|\mathcal{I}_S^\dagger\|_2 \log^\gamma(\tilde{\kappa}^{-1/2}\alpha_1)\frac{\log n}{n} + \lambda\left\|\mathcal{I}_S^\dagger \nabla R(\beta^\star)\right\|_2\right)^3$$

$$\leq -\frac{1}{2}z^T \mathcal{I}_S^0 z + \left(B_2\sqrt{\frac{\log n}{n}} + \frac{3\lambda L}{2}\right)\left(c\sqrt{\frac{\mathsf{Tr}(\mathcal{I}_S^\dagger)\log n}{n}} + \lambda\left\|\mathcal{I}_S^\dagger \nabla R(\beta^\star)\right\|_2\right)^2$$

$$+ \frac{B_3}{6}\left(c\sqrt{\frac{\mathsf{Tr}(\mathcal{I}_S^\dagger)\log n}{n}} + \lambda\left\|\mathcal{I}_S^\dagger \nabla R(\beta^\star)\right\|_2\right)^3$$

$$\leq -\frac{1}{2}z^T \mathcal{I}_S^0 z + \left(2B_2\sqrt{\frac{\log n}{n}} + 3\lambda L\right)\left(c\frac{\mathsf{Tr}(\mathcal{I}_S^\dagger)\log n}{n} + \lambda^2\left\|\mathcal{I}_S^\dagger \nabla R(\beta^\star)\right\|_2^2\right)$$

$$+ \frac{2B_3}{3}\left(c\left(\frac{\mathsf{Tr}(\mathcal{I}_S^\dagger)\log n}{n}\right)^{3/2} + \lambda^3\left\|\mathcal{I}_S^\dagger \nabla R(\beta^\star)\right\|_2^3\right). \qquad (49)$$

Here, the first and second inequality holds as long as $n \geq N_4'$ and the last inequality follows from the fact that $(a+b)^n \leq 2^{n-1}(a^n + b^n)$.

Similarly, we have

$$\hat{L}(\beta) - \hat{L}(\beta_0)$$

$$\geq \frac{1}{2}(\Delta_\beta - z)^T \mathcal{I}_S^0 (\Delta_\beta - z) - \frac{1}{2} z^T \mathcal{I}_S^0 z - \left( B_2 \sqrt{\frac{\log n}{n}} + \lambda L \right) \|\Delta_\beta\|_2^2 - \frac{B_3}{6} \|\Delta_\beta\|_2^3. \tag{50}$$

Thus, for any $\beta \in \mathcal{D}_S^0 \cap \mathcal{B}(\beta_0, n^{-3/8})$, we have

$$\hat{L}(\beta) - \hat{L}(\beta_0)$$
$$\geq \frac{1}{2}(\Delta_\beta - z)^T \mathcal{I}_S^0 (\Delta_\beta - z) - \frac{1}{2} z^T \mathcal{I}_S^0 z - B_2 n^{-\frac{7}{6}} - c_\lambda L n^{-\frac{7}{6} + \frac{1}{3\tau}} - \frac{B_3}{6} n^{-\frac{9}{8}}. \tag{51}$$

(51) - (49) gives

$$\hat{L}(\beta) - \hat{L}(\beta_0 + z)$$
$$\geq \frac{1}{2}(\Delta_\beta - z)^T \mathcal{I}_S^0 (\Delta_\beta - z)$$
$$- \left( 2B_2 \sqrt{\frac{\log n}{n}} + 3\lambda L \right) \left( c \frac{\mathsf{Tr}(\mathcal{I}_S^\dagger) \log n}{n} + \lambda^2 \left\| \mathcal{I}_S^\dagger \nabla R(\beta^\star) \right\|_2^2 \right)$$
$$- \frac{2B_3}{3} \left( c \left( \frac{\mathsf{Tr}(\mathcal{I}_S^\dagger) \log n}{n} \right)^{3/2} + \lambda^3 \left\| \mathcal{I}_S^\dagger \nabla R(\beta^\star) \right\|_2^3 \right)$$
$$- B_2 n^{-\frac{7}{6}} - c_\lambda L n^{-\frac{7}{6} + \frac{1}{3\tau}} - \frac{B_3}{6} n^{-\frac{9}{8}}$$
$$> \frac{1}{2}(\Delta_\beta - z)^T \mathcal{I}_S^0 (\Delta_\beta - z) - B_3 n^{-\frac{9}{8}}. \tag{52}$$

Here the last inequality holds as long as $n \geq N_4'$.

Consider the ellipsoid

$$\mathcal{D} := \left\{ \beta \in \mathcal{D}_S^0 \,\middle|\, \frac{1}{2}(\Delta_\beta - z)^T \mathcal{I}_S^0 (\Delta_\beta - z) \leq B_3 n^{-\frac{9}{8}} \right\}.$$

Then by (52), for any $\beta \in \mathcal{D}_S^0 \cap \mathcal{B}(\beta_0, n^{-3/8}) \cap \mathcal{D}^C$,

$$\hat{L}(\beta) - \hat{L}(\beta_0 + z) > 0. \tag{53}$$

Notice that by the definition of $\mathcal{D}$, we have for any $\beta \in \mathcal{D}$,

$$\left\| (\mathcal{I}_S^0)^{\frac{1}{2}} (\Delta_\beta - z) \right\|_2^2 \leq 2B_3 n^{-\frac{9}{8}}.$$

Since $\Delta_\beta - z \in \mathsf{col}(\mathcal{I}_S^0)$, we have

$$\|\Delta_\beta - z\|_2^2 \leq 2\alpha^{-1} B_3 n^{-\frac{9}{8}}$$

where the inequality follows from Assumption C.2.

As a result, we have

$$\|\Delta_\beta\|_2^2 \leq 4B_3 \alpha^{-1} n^{-\frac{9}{8}} + 2\|z\|_2^2$$
$$\leq 4B_3 \alpha^{-1} n^{-\frac{9}{8}} + 2 \left( c \sqrt{\frac{\mathsf{Tr}(\mathcal{I}_S^\dagger) \log n}{n}} + \lambda \left\| \mathcal{I}_S^\dagger \nabla R(\beta^\star) \right\|_2 \right)^2$$
$$\leq 2 \left( c \sqrt{\frac{\mathsf{Tr}(\mathcal{I}_S^\dagger) \log n}{n}} + \lambda \left\| \mathcal{I}_S^\dagger \nabla R(\beta^\star) \right\|_2 \right)^2, \tag{54}$$

where the last inequality holds as long as $n \geq N_4'$. It then holds that

$$\|\Delta_\beta\|_2 \leq 2 \left( c \sqrt{\frac{\mathsf{Tr}(\mathcal{I}_S^\dagger) \log n}{n}} + \lambda \left\| \mathcal{I}_S^\dagger \nabla R(\beta^\star) \right\|_2 \right) \leq n^{-3/8},$$

Here, the last inequality holds as long as $n \geq N_4'$. In other words, we show that $\mathcal{D} \subset \mathcal{D}_S^0 \cap \mathcal{B}(\beta_0, n^{-3/8})$.

Recall that by Lemma C.16, we have

$$\hat{\beta}_\lambda \in \mathcal{D}_S^0 \cap \mathcal{B}(\beta_0, D'') \subset \mathcal{D}_S^0 \cap \mathcal{B}(\beta_0, n^{-3/8}).$$

Also, for any $\beta \in \mathcal{D}_S^0 \cap \in \mathcal{B}(\beta_0, n^{-3/8}) \cap \mathcal{D}^C$, we have

$$\hat{L}(\beta) - \hat{L}(\beta_0 + z) > 0.$$

Consequently, we conclude

$$\hat{\beta}_\lambda^0 \in \mathcal{D}_S^0 \cap \mathcal{B}(\beta_0, D'') \cap \mathcal{D}.$$

By the definition of $\mathcal{D}$, we have

$$\left\| (\mathcal{I}_S^0)^{1/2}(\Delta_{\hat{\beta}_\lambda^0} - z) \right\|_2^2 \leq 2B_3 n^{-\frac{9}{8}}. \tag{55}$$

By (54), we further have

$$\|\hat{\beta}_\lambda^0 - \beta_0\|_2 \leq 2 \left( c\sqrt{\frac{\mathsf{Tr}(\mathcal{I}_S^\dagger) \log n}{n}} + \lambda \left\| \mathcal{I}_S^\dagger \nabla R(\beta^\star) \right\|_2 \right). \tag{56}$$

Note that by taking $A = \mathcal{I}_T^{\frac{1}{2}} \mathcal{I}_S^\dagger$ in Proposition C.14, we have

$$\|\mathcal{I}_T^{\frac{1}{2}} \mathcal{I}_S^\dagger g\|_2 \leq c\sqrt{\frac{\mathsf{Tr}(\mathcal{I}_S^\dagger \mathcal{I}_T) \log n}{n}} + B_1 \|\mathcal{I}_T^{\frac{1}{2}} \mathcal{I}_S^\dagger\|_2 \log^\gamma \left( \frac{B_1 \|\mathcal{I}_T^{\frac{1}{2}} \mathcal{I}_S^\dagger\|_2}{\sqrt{\mathsf{Tr}(\mathcal{I}_S^\dagger \mathcal{I}_T)}} \right) \frac{\log n}{n}$$

$$\leq c\sqrt{\frac{\mathsf{Tr}(\mathcal{I}_S^\dagger \mathcal{I}_T) \log n}{n}} + B_1 \|\mathcal{I}_T^{\frac{1}{2}} \mathcal{I}_S^\dagger\|_2 \log^\gamma(\kappa^{-1/2}\alpha_1) \frac{\log n}{n}. \tag{57}$$

Thus, we have

$\|\mathcal{I}_T^{\frac{1}{2}}(\hat{\beta}_\lambda^0 - \beta_0)\|_2^2$

$\leq 2\|\mathcal{I}_T^{\frac{1}{2}}((\mathcal{I}_S^0)^{\frac{1}{2}})^\dagger (\mathcal{I}_S^0)^{\frac{1}{2}}(\Delta_{\hat{\beta}_\lambda^0} - z)\|_2^2 + 2\|\mathcal{I}_T^{\frac{1}{2}} z\|_2^2$

$\leq 2\|\mathcal{I}_T^{\frac{1}{2}}((\mathcal{I}_S^0)^{\frac{1}{2}})^\dagger\|_2^2 \|(\mathcal{I}_S^0)^{\frac{1}{2}}(\Delta_{\hat{\beta}_\lambda^0} - z)\|_2^2 + 4\|\mathcal{I}_T^{\frac{1}{2}} \mathcal{I}_S^\dagger g\|_2^2 + 4\lambda^2 \|\mathcal{I}_T^{\frac{1}{2}} \mathcal{I}_S^\dagger \nabla R(\beta^\star)\|_2^2 + c\frac{B_3^2 \|\mathcal{I}_T\|}{\alpha^4} \Delta_2^{2\tau}$

$\leq 4 \left( c\sqrt{\frac{\mathsf{Tr}(\mathcal{I}_S^\dagger \mathcal{I}_T) \log n}{n}} + B_1 \|\mathcal{I}_T^{\frac{1}{2}} \mathcal{I}_S^\dagger\|_2 \log^\gamma(\kappa^{-1/2}\alpha_1) \frac{\log n}{n} \right)^2 + 4\lambda^2 \|\mathcal{I}_T^{\frac{1}{2}} \mathcal{I}_S^\dagger \nabla R(\beta^\star)\|_2^2$

$\quad + 4B_3 \|\mathcal{I}_T\|_2 (\|\mathcal{I}_S^\dagger\|_2 + \alpha^{-2} B_3 \Delta_2^\tau) n^{-\frac{9}{8}} + c\frac{B_3^2 \|\mathcal{I}_T\|}{\alpha^4} \Delta_2^{2\tau}$

$\leq c \left( \frac{\mathsf{Tr}(\mathcal{I}_S^\dagger \mathcal{I}_T) \log n}{n} + \lambda^2 \|\mathcal{I}_T^{\frac{1}{2}} \mathcal{I}_S^\dagger \nabla R(\beta^\star)\|_2^2 \right). \tag{58}$

Here the last inequality holds as long as $n \geq N_4'$.

$\square$