# OpenReview forum: "Principled Out-of-Distribution Generalization via Simplicity"
_ICLR.cc/2026/Conference — Submitted to ICLR 2026_

### Official Review · Reviewer_smRC · 2025-10-31

**Soundness:** 3
**Presentation:** 1
**Contribution:** 2
**Rating:** 2
**Confidence:** 3

**Summary:**

This paper considers generalization to out of distribution source data through looking at simplicity. The authors begin by introducing what they understand as generalization, after which they argue that best generalization arises through minimizing a "simplicity" function. They then provide some experiments as examples, highlighting the effect in some toy problems. They then proceed to introduce theorems with derivations relating excess risk to the simplicity metric for the case of a constant simplicity gap, and a vanishing simplicity gap.

**Strengths:**

- Generalization is an important and interesting topic. Especially with regards to diffusion models where there is an optimal densoiser available in closed form that would only produce images from the training set. Understanding why these models generalize nonetheless is an important question.

**Weaknesses:**

- Overall I found this paper difficult to follow, experiments are not well connected to the presented theory, and presented theory is not elaborated on very much, it is left to the reader to interpret the experimental results in light of the presented theory.
- It seems that none of the theory the authors develop is actually related to diffusion models other than in the general "generative model", and even then this seems like a stretch because the authors specifically consider "conditional" generative models, which could be called classifiers or perhaps regressors for the examples the autors consider. Doing this analysis for classifiers or conditional generative models is not necessarily wrong or uninteresting, but the text no longer matches the content.
- The diffusion experiment seems contrived and it also seems that its only purpose is to relate the work to diffusion models. Diffusion is not mentioned in all of Section 4.
- In Section 4.1, it seems as if the authors struggled to formalize their hypothesis into mathematical formulation, there are a lot of different notations: for the optimal model parameters, sets thereof, including and excluding the simplicity metric, etc. Cleaning up this crucial section would greatly help readability in the paper.
- It seems that the authors neglect to define the "simplicity metric" other than "R(.)". It appears to me that such a central concept in the framework deserves more attention and elaboration.
- After the section of motivating experiments, the main results are presented as theorems for two separate scenarios. Unfortunately the authors neglect to relate those theorems back to the earlier experiments, leaving the reader to understand what the relevance of them is with regards to the earlier experiments.

**Questions:**

In particular the exeriment in 3.2 raises some questions:
- On line 152 it states that "Our training data consists of n iid samples from the source domain", yet in 216 it satates "Each such model is trained on 400 samples from S, along with 400 samples from T". These are inconsistent, and it is important to clarify, since the example in its current form does not match the setup the paper seems to outline.
- The previous question already may clarify this, but it seems that the authors are trying to learn an identity function from partially correct and corrupted data. I am struggling to understand what the goal and setup is here. If the setup is to learn an identity function from corrupted data, this seems more like an excercise in regularization. In particular if the goal is to learn the correct mapping despite providing wrong data, this leaves open the question what the real "correct" data should be? If one changes the problem so that the "permuted" solution is arbitrarily chosen as the correct one, would this change the results or interpretation? I would very much like it if the authors could clarify what the setup is in 3.2 and why they give it the interpretation they do.
Then some clarifications regarding 4.1 would be useful:
- Can the authors elaborate what they mean in the unlabelled equation in 278? The text in 275-277 does not seem to match the formula.
- In 286, it is stated that the solutions to 1 are not unique, while in 275-277 the authors argue that the inclusion of R creates the "true" parameter (which I read to be "unique"). Can the authors elaborate?

---

### Official Review · Reviewer_6cLQ · 2025-11-01

**Soundness:** 2
**Presentation:** 2
**Contribution:** 1
**Rating:** 2
**Confidence:** 3

**Summary:**

The paper proposes a framework to explain the generalization ability of large-scale models beyond their training distributions. The authors argue, based on a simple conceptual experiment, that among all models fitting the training data equally well, the one that generalizes is the simplest model with respect to a certain complexity metric. Assuming that the model that well generalizes over the target distribution $T$ among the ones that fit the source distribution $S$ is the simplest in the sense that either of the following two cases hold:
(i) constant-gap case, where the true model is strictly simpler than spurious ones, or
(ii) vanishing-gap case, where simplicity varies smoothly among solutions,
they derive high-probability convergence bounds on the excess risk when using the regularized objective favoring simplicity.

**Strengths:**

The paper's observation that simplicity may be aligned with OOD generalization ability is interesting.
The paper seems technically strong, in the sense that their learning-theoretical analysis of OOD generalization seems solid and clearly stated.

**Weaknesses:**

1. **Weak validation of the simplicity–generalization link**

One of the paper’s central conceptual claims, arguably its most important contribution, is the proposed association between simplicity and out-of-distribution generalization. However, this connection is not validated in a convincing way. The authors first show that diffusion models can generalize on an extremely simple synthetic conditional generation task, and then abruptly pivot to a toy setting with identity-mapping learning experiment to argue that generalizable models tend to be simpler. This setup feels ad-hoc and leaves open whether the observed correlation is a genuine principle or merely an artifact of the trivial setting. In fact, I initially expected the theoretical analysis to derive or explain this association formally, but the paper lacks it.

2. **Unclear theoretical message**

The theory essentially assumes the existence of a simplicity measure that separates the truly generalizable model from spurious ones, and then proves that a regularized objective using that measure recovers the correct solution. While the technical analysis may be nontrivial, the conceptual takeaway is limited and not surprising: if one knows the right simplicity metric a priori, optimizing for it should work. This makes the result feel tautological and of limited practical significance, as it offers little guidance on how such a metric should be identified or justified in realistic settings.

**Questions:**

Could you further validate the association between model simplicity and generalization capability? What will be the simplicity measure that can be used in practice? Rather than assuming it, can you theoretically explain why a well-generalizing model should be simpler?

---

### Official Review · Reviewer_ouaz · 2025-11-01

**Soundness:** 3
**Presentation:** 3
**Contribution:** 2
**Rating:** 4
**Confidence:** 2

**Summary:**

This paper studies the out-of-distribution generalization. They proposed a complexity metric $R(\cdot)$ to measure the OOD generalizability of neural network, inspired from the empirical observations that when achieves OOD generalization, the network has a simple form (measure in terms of norm). Furthermore, they develop a theoretical framework for OOD generalization via $R(\cdot)$, study the problem under both (1) constant-gap (2) vanishing-gap regimes, establish the required sample complexity to learn generalizable simple model.

**Strengths:**

The paper introduces a simplicity metric $R$ and formalizes the intuition that simplicity aligns with generalization into a rigorous theoretical framework for out-of-distribution (OOD) generalization. It makes a clear theoretical contribution toward understanding why and how machine learning models are able to generalize beyond their training distributions.

**Weaknesses:**

The paper uses diffusion model compositional generalization as a motivating background, but there remains a substantial gap between its empirical and theoretical analyses and real diffusion model settings:

1. The paper studies the negative log-likelihood loss, which is only a lower bound of the denoising score matching objective used in diffusion models [1].

2. There is a large discrepancy between the OOD generalization behavior demonstrated in diffusion models (Section 3.1) and the simplified setting (Section 3.2). To convincingly show that “simplicity is a key factor in achieving successful OOD generalization,” it would be more compelling to evaluate this principle directly on real diffusion models.

3. Some assumptions, such as B.1 (strong convexity), are quite restrictive and may not hold for deep neural networks like diffusion models.

I am not an expert in machine learning, so I cannot fully assess the contribution of the main theoretical results in Section 4. However, I believe there exists a significant gap between the simplified conceptual framework and practical diffusion or foundation model settings, making it challenging to directly connect the theoretical findings with real-world applications.

[1] Song, Yang, Conor Durkan, Iain Murray, and Stefano Ermon. "Maximum likelihood training of score-based diffusion models." Advances in neural information processing systems 34 (2021): 1415-1428.

**Questions:**

In line 244, the paper claims that “simplicity is a key factor in achieving successful OOD generalization.” Although I generally agree with this intuition, I do not think the experiment in Section 3.2 convincingly demonstrates it. The generalizable mapping is deliberately designed to be an identity mapping, which is inherently simple. In contrast, the non-generalizable alternatives are constructed to be more complex. Therefore, the observed differences in simplicity and weight norms arise primarily from the mappings chosen for training rather than from out-of-distribution generalization behavior.

---

### Official Review · Reviewer_Dk6e · 2025-11-07

**Soundness:** 3
**Presentation:** 3
**Contribution:** 3
**Rating:** 4
**Confidence:** 3

**Summary:**

This paper studies a very important question: what's the main factor that enables modern networks generalize well to out-of-distribution (OOD) data. The authors propose a main idea: simplicity. They argue that even if many different models can fit the training data perfectly, the one that generalizes correctly is the simplest one.

The paper shows this mainly theoratically by building a framework around this idea. They study a regularized maximum likelihood estimator and provide theoretical proofs showing how many samples are needed to find this "true, simple" model. They analyze two situations: a "constant-gap" (where the true model is much simpler) and a "vanishing-gap" (where the true model is only a little simpler).

**Strengths:**

This paper tackles the important question of OOD generalization. The paper is a great read to understand what kind of model helps handle OOD best. It gives a clear principle: among all the models that can fit your data, the simplest one is the one you should trust to generalize. This is a very useful idea. This is a great way to frame the problem as simplicity or say regularization is not just for fighting noise; it is the main tool for selecting the one true model from all these perfect solutions. The theoretical part of the paper is strong. The authors are careful to define their assumptions and build their proofs on them. Analyzing both the "constant-gap" and "vanishing-gap" settings makes the analysis more complete and realistic. The paper is written very clearly.

**Weaknesses:**

The main weakness is that there are not many experiments to support the paper empirically. The MLP experiment is very clean and simple, which is good for explaining the idea. However, this is very different from the complex tasks that real foundation models face. It is hard to be sure that this "simplicity" principle will work for real, large-scale computer vision or language problems.

**Questions:**

1. Clarification on the diffusion model conditioning (Section 3.1): The paper states that a "text-conditioned diffusion model" was trained. This term is slightly ambiguous. Could the authors please clarify whether this was a class-conditioned model (e.g., using the 3-bit labels like (0,0,1) directly as conditioning) or if a separate text encoder was used to encode natural language descriptions of the attributes?

2. Need for stronger quantitative OOD benchmarking: The core empirical evidence for the paper's simplicity hypothesis rests on the 2-layer MLP experiment. While this is a clean and valuable illustration, it is a very controlled, low-dimensional task. The paper's claim would be significantly more convincing if it were supported by quantitative results on more standard, challenging OOD benchmarks. Could the authors demonstrate that models with lower parameter norms (i.e., "simpler" models, perhaps trained with stronger regularization) consistently outperform their higher-norm ("more complex") counterparts on established OOD tasks?

3. In diffusion models, discussions of "simplicity" often refer to architectural complexity (e.g., the number of parameters or layers). This paper frames simplicity as a property of the parameter norms within a fixed, high-capacity architecture. It would be highly beneficial for the authors to explicitly discuss this distinction. Since most modern foundation models already include regularization by default, an analysis or discussion on the interplay between architectural complexity and the paper's focus on parameter-norm simplicity would greatly enrich the work.

The core content is compelling. I would be happy to raise my score if more quantitative experimental results, as suggested in point 2, are included to validate these claims more broadly.

---

### Meta-Review · Area_Chair_HHaB · 2026-01-09

**Summary:**

Reviewers all point issues and no rebuttal has been submitted to discuss them.

**Reviewer Concerns:**

No rebuttal submitted.

**Reviewer Scores:**

No rebuttal submitted.

---

### Decision · Program_Chairs · 2026-01-26

Reject